# IMPLICIT BIAS OF HESSIAN APPROXIMATION IN REGULARIZED RANDOMIZED SR1 METHOD

## ABSTRACT

Quasi-Newton methods have recently been shown to demonstrate dimension-independent convergence rate outperforming vanilla gradient descent (GD) in modern high-dimensional problems. By examining the spectrum of the Hessian approximation throughout the iterative process, we analyze a regularized quasi-Newton algorithm based on the standard randomized symmetric rank-one (SR1) update. The evolution of the spectrum reveals an implicit bias introduced by the Hessian learning, which promotes a preferential reduction of certain eigenvalues. This observation precisely captures the quality of Hessian approximation. Incorporating the implicit effect of Hessian update, we show that the regularized randomized SR1 method achieves a convergence rate of $\tilde{\mathcal{O}}\left(\frac{d_{\text{eff}}^2}{k^2}\right)$ for standard self-concordant objective functions, where $d_{\text{eff}}$ is the effective dimension of Hessian. In specific high-dimensional settings, which are common in practice, this method preserves convergence speeds comparable to accelerated gradient descent (AGD) while maintaining similar computational complexity per iteration. This work highlights the impact of implicit bias and offers a new theoretical perspective on the efficiency of quasi-Newton methods.

## 1 INTRODUCTION

We are interested in using quasi-Newton methods to solve the following unconstrained convex optimization problem:
$$\min_{x \in \mathbb{R}^d} f(x). \tag{1}$$
The convergence properties have been widely studied since the 1970s in the asymptotic regime (Broyden, 1970; BROYDEN et al., 1973; Dennis & More, 1973; Khalfan et al., 1993). Recently, a series of breakthrough works such as Rodomanov & Nesterov (2022); Jin & Mokhtari (2023) have obtained explicit non-asymptotic convergence rates for quasi-Newton methods for quadratic objectives and local problems of general objectives. These initial results have been refined through further research (Krutikov et al., 2023; Rodomanov, 2024; Jin et al., 2025a;b) and have offered insights guiding the development of algorithmic design (Liu & Luo, 2022; Jin et al., 2022).

In these analyses, non-asymptotic local convergence after the $k$-th iteration is given by, for example, $\left(e^{d/k \ln \kappa} - 1\right)^{k/2}, (d\kappa/k)^k, (d \log \kappa/k)^k$, where $d$ is the problem dimension and $\kappa$ the condition number. This implies that superlinear convergence is achieved when $k = \tilde{\Omega}(d)$, but no meaningful rate is guaranteed for $k = \mathcal{O}(d)$. The requirement of $k = \Omega(d)$ arises because accurate Hessian approximation across all dimensions is necessary for superlinear convergence in regions where the Hessian remains stable. Since quasi-Newton methods update the Hessian approximation with constant-rank strategies, their convergence inherently depends on $d$. The high dependence of the convergence rate on the dimension $d$ significantly limits its interpretability in practical high-dimensional optimization problems, which is a central challenge in modern learning context.

However, quasi-Newton algorithms, served as intermediate methods between gradient descent methods, which have dimension-independent convergence properties, and Newton's methods, which are renowned for their rapid convergence rates, are expected to demonstrate dimension-independent convergence and also surpass the performance of vanilla gradient descent even in high dimensions. Empirical studies (Goldfarb et al., 2020; Berahas et al., 2022; Yousefi & Martínez, 2023) also

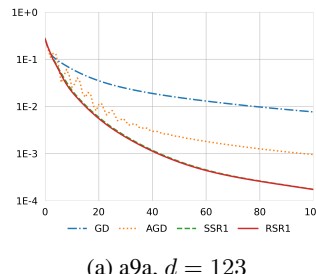 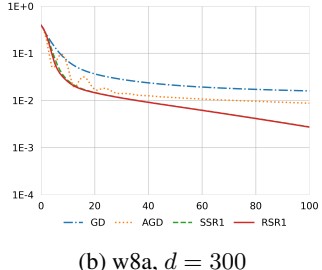

(a) a9a, $d = 123$        (b) w8a, $d = 300$

Figure 1: Early-stage convergence behavior of regularized SR1 versus GD and AGD methods in linear regression tasks. Experimental results on a9a and w8a datasets reveal that the regularized SR1 algorithm matches the rapid initial convergence speed of AGD, while outperforming standard GD.

substantiate this performance advantage, demonstrating that quasi-Newton methods and their variants (such as limited-memory versions) are more efficient than vanilla gradient descent in high-dimensional problems. Thus, it becomes important to examine the convergence behavior of quasi-Newton methods through the lens of complexity theory with the following naturally raised question: how do quasi-Newton methods differentiate from gradient descent in high-dimensional settings through the lens of complexity theory?

A key factor behind the success of modern machine learning algorithms is implicit bias, which states that distinct optimization trajectories, despite converging to the same target (e.g. minimizing the empirical loss), preferentially select certain trajectories over others. This preference can significantly impact their practical effectiveness (Gunasekar et al., 2017; Arora et al., 2019; Li et al., 2022). In particular, the condition associated with implicit bias, **low effective dimension** of Hessian, is ubiquitous and naturally arises in high-dimensional machine learning problems (Cai & Hall, 2006; Liang & Rakhlin, 2020). This condition has been empirically substantiated (Sagun et al., 2018; Ghorbani et al., 2019) and serves as a fundamental theoretical assumption (Silin & Fan, 2022). In the context of Hessian approximation, when interpreting the quasi-Newton method as online learning processes targeting Hessian matrix approximation, while the update rules eventually achieve full Hessian approximation and guarantee superlinear convergence, they inherently prioritize specific dimensional approximations before the superlinearly converging phase. This work aims to characterize such implicit preference and sheds light on how to enhance optimization efficiency in high-dimensional problems, thereby establishing a theoretically guaranteed convergence rate.

In this study, we build upon the framework of randomized symmetric rank-one (SR1) algorithms (Davidon, 1991; Lin et al., 2022), a specific quasi-Newton method chosen for its simple update rule, best local theoretical guarantees, and broad prior study in the recent non-asymptotic convergence result (Liu & Luo, 2021; Ye et al., 2023; Liu et al., 2024a). Concretely, we first quantify the implicit bias of Hessian approximation in terms of its trace. Then, utilizing this implicit preference, we demonstrate that regularization, a widely employed practical and analytical technique in Newton-type methods (Moré, 1978), enables quasi-Newton algorithms to achieve improved convergence rates. Finally, we illustrate the empirically observed global effectiveness of quasi-Newton methods in high-dimensional optimization under a specific setting. This setting is applicable across a broad spectrum of optimization problems.

**Organization.** The rest of the paper is organized as follows. Section 2 discusses our work with related literature. Section 3 provides necessary background for our algorithm to be analyzed. Section 4 presents our main result for Hessian approximation and a proof sketch. Section 5 applies the main result to regularized SR1 method and rigorously establishes global convergence rates of $\tilde{\mathcal{O}}(1/k^2)$, as well as its application in high-dimensional optimization. Section 6 gives experiments on our framework compared with other methods and Section 7 summarizes the paper and points out its limitation and future direction.

## 2 RELATED WORK

**Hessian approximation methods.** In addition to traditional quasi-Newton updates and its variants, other approximation techniques involves low-dimensional subspace Newton methods (Gower et al., 2019; Qu et al., 2016; Doikov et al., 2018; Hanzely et al., 2020; Jiang et al., 2024), and a various of stochastic QN methods (Bordes et al., 2009; Byrd et al., 2016; Gower et al., 2016). Doikov et al. (2024) noted the importance of spectral preconditioning and analyzed the influence of Hessian's spectrum to regularized QN methods. Our result fully consider the evolution of spectrum but does not rely on its specific structure.

**Cubic regularization methods.** Regularized Newtons method stabilizes iterations in singular curvatures (Li et al., 2004; Burger & Kaltenbacher, 2006), with asymptotic quadratic convergence proven by Polyak (2009). Adaptive variants, like Nesterovs cubic regularization (Nesterov & Polyak, 2006) (implicitly adjusting regularization via step size) and gradient norm regularization (Mishchenko, 2023a; Doikov & Nesterov, 2024), achieve $\tilde{\mathcal{O}}(\frac{1}{k^2})$ rates under Hessian Lipschitz conditions. However, these methods require exact Hessians, requiring high computational complexity. To reduce the computational cost, Benson & Shanno (2018); Ghadimi et al. (2017); Kamzolov et al. (2023b); Scieur (2024) proposed cubic regularized inexact Newton methods. Some of them (Kamzolov et al., 2023b; Jiang et al., 2024) also achieved global convergence rate $\mathcal{O}\left(\frac{1}{k^2}\right)$, but they were based on either assumptions on Hessian approximation quality, or Hessian's low-rank structure. Note that they still introduced cubic acceleration strategies including solving a non-trivial sub-problem in each iteration, which are beyond the scope of this paper. Our framework is much simpler, more general, and easier to implement than theirs.

**Quasi-Newton methods.** The quasi-Newton method approximates Hessian information via secant equations, with different forms including DFP (Davidon, 1991), BFGS (Shanno, 1970), and SR1 (Davidon, 1991). Recent work by Rodomanov & Nesterov (2021b) established non-asymptotic rates for greedy quasi-Newton updates, followed by analyses of classical (Rodomanov & Nesterov, 2022; 2021a; Rodomanov, 2024; Ye et al., 2023; Jin & Mokhtari, 2023) and modified methods (Lin et al., 2022; Liu et al., 2024b; Liu & Luo, 2021). Among these analyses, the greedy or randomized SR1 achieves the fastest rate $\mathcal{O}((1-\frac{1}{d})^{\frac{k(k-1)}{2}})$ (Lin et al., 2022) but requires $\mathcal{O}(d)$ iterations to enter the convergence phase. These methods have been extended to non-linear equations (Ye et al., 2021; Liu et al., 2023), saddle-point problems (Xiao et al., 2024; Liu & Luo, 2021), and other settings (Ranganath et al., 2025; Benson & Shanno, 2018; Du & You, 2024).

## 3 PRELIMINARIES

### 3.1 NOTATION AND PROBLEM SETUP

We consider the problem in Euclidean space $\mathbb{R}^d$. Denote $\lambda_{\max}(\mathbf{A}) = \lambda_1(\mathbf{A}) \geq \cdots \geq \lambda_d(\mathbf{A})$ as the eigenvalues of a real symmetric matrix $\mathbf{A} \in \mathbb{R}^{d \times d}$. For any $x, y \in \mathbb{R}^d$ and function $g$, if $\nabla^2 g(y) \succeq 0$, we denote $\|x\|_y = \sqrt{x^\top \nabla^2 g(y) x}$ and $g^*$ as the minimum of $g$. Throughout the paper, we will make assumptions on the objective function $f(x)$ as followed:

**Assumption 1** (Bounded level sets). *The diameter of the level set at the initial point $x_0$,*

$$\mathcal{L}(x_0) \stackrel{def}{=} \left\{ x \in \mathbb{R}^d : f(x) \leq f(x_0) \right\}$$

*is bounded by a constant $D$, then $\|x_0 - x^*\|_2 \leq D$.*

**Assumption 2** (Gradient Lipschitz). *There exist a constant $L$ such that for all $x, y \in \mathbb{R}^d$, we have*

$$\|\nabla f(x) - \nabla f(y)\|_2 \leq L \|x - y\|_2.$$

**Assumption 3** (Self-concordancy). *The objective function $f \in C^2(\mathbb{R}^d)$ is convex and there exists a constant $M$ such that for all $x, y \in \mathbb{R}^d$ and $\|y - x\|_x < \frac{1}{M}$, we have*

$$(1 - M\|y - x\|_x)^2 \nabla^2 f(x) \preceq \nabla^2 f(y) \preceq \frac{1}{(1 - M\|y - x\|_x)^2} \nabla^2 f(x). \tag{2}$$

Assumption 3 is a standard assumption in the convergence of Newton method (Nesterov & Nemirovskii, 1994; Nesterov, 2018) and the recent non-asymptotic convergence analysis of quasi-Newton methods (Rodomanov & Nesterov, 2021b; Lin et al., 2022).

## 3.2 METHODOLOGY

**Randomized SR1 update.** The Hessian update in quasi-Newton methods can be viewed as an online learning process which iteratively refines the approximation to the current Hessian matrix. The SR1 update can be categorized into different versions. The classical SR1 methods find the Hessian approximation $\mathbf{B}_{k+1}$ for the next iteration using the moving direction and the secant equation. The recent progress on non-asymptomatic analysis originates from other types of SR1 update:

$$\mathbf{B}_{k+1} = \mathbf{SR1}(\mathbf{A}, \mathbf{B}_k, \mathbf{s}_k) \stackrel{\text{def}}{=} \mathbf{B}_k + \frac{(\mathbf{A} - \mathbf{B}_k)\mathbf{s}_k \mathbf{s}_k^\top (\mathbf{A} - \mathbf{B}_k)}{\mathbf{s}_k^\top (\mathbf{A} - \mathbf{B}_k)\mathbf{s}_k}, \tag{3}$$

where $\mathbf{A} \in \mathbb{R}^{d \times d}$ is the target Hessian matrix at the current iteration, $\mathbf{B}_k \in \mathbb{R}^{d \times d}$ is the current Hessian approximation, and $\mathbf{s}_k \in \mathbb{R}^d$ is the update direction vector for the rank-one correction, selected by a greedy or random strategy. Rodomanov & Nesterov (2021b) established the first superlinear convergence of quasi-Newton method utilizing the greedy update, and was later extended to randomized update (Lin et al., 2022) and secant equation update (Rodomanov & Nesterov, 2021a). This paper primarily focuses on the algorithm bias of (3) from randomized update direction. To be specific, given a distribution $\mu$ defined on the real line $\mathbb{R}$ that satisfies

$$\mathbb{E}_{x \sim \mu}[x] = 0, \quad \mathbb{E}_{x \sim \mu}[x^2] = 1, \quad \mathbb{E}_{x \sim \mu}[x^4] = C_1 < +\infty, \tag{4}$$

We draw a sample vector $\mathbf{u}_k$ whose coordinates are independently generated from $\mu$.

**Regularized Newton's method.** Regularization techniques, also referred to as the Levenberg-Marquardt regularization (Moré, 1978), are commonly used to stabilize Newton-type algorithms. They interpolate between Newton's method and gradient descent through a damping factor $\alpha_k > 0$ and perform the update:

$$x_{k+1} = x_k - (\mathbf{G}_k + \alpha_k \mathbf{I}_d)^{-1} \nabla f(x_k), \tag{5}$$

where $\mathbf{G}_k$ represents the second order information (typically Hessian matrix (Mishchenko, 2023b) or its quasi Newton approximation (Kamzolov et al., 2023a)). Large $\alpha_k$ biases the method towards the gradient descent direction, which promotes stability in singular curvature, while small $\alpha_k$ permits fast Newton-type local convergence when the local geometry is well-conditioned. As a consequence, implementations of regularized Newton's method progressively reduce $\alpha_k$, leading to a transition from the unstable global exploration to the local faster convergence.

# 4 SPECTRAL ANALYSIS FOR $\mathbf{A} - \mathbf{B}_k$

## 4.1 MOTIVATION AND MAIN RESULT

In this section, we start the analyses with the theoretical guarantees of SR1 approximating a positive semi-definite matrix $\mathbf{A} \in \mathbb{R}^{d \times d}$. Given an initial matrix $\mathbf{B}_0 \preceq \mathbf{A}$, the randomized SR1 updates the approximation as (3) where $\mathbf{s}_k$ is randomly sampled from a distribution satisfying (4).

To illustrate the implicit preference of the update rule (3), we first provide a **heuristic analysis** through a deterministic analogy of the update (3). While the simplified deterministic analogy lacks mathematical rigor, it offers valuable insight into the algorithm.

In the deterministic analogy of (3): (1) the denominator $\mathbf{s}_k^\top (\mathbf{A} - \mathbf{B}_k)\mathbf{s}_k$ is replaced by a deterministic scalar $c_{\text{den}}$, suggested by the concentration of high-dimensional vectors; (2) the rank-one $\mathbf{s}_k \mathbf{s}_k^\top$ is replaced by its expectation $\mathbf{I}_d$. Then, we can write the evolution of $\mathbf{A} - \mathbf{B}_k$ under the deterministic update as:

$$(\mathbf{A} - \mathbf{B}_{k+1}) = (\mathbf{A} - \mathbf{B}_k) - \frac{1}{c_{\text{den}}}(\mathbf{A} - \mathbf{B}_k)^2 \tag{6}$$

Without loss of generality, consider a diagonal $\mathbf{A} = \text{diag}(\lambda_1, \lambda_2, \ldots, \lambda_d)$ with $\lambda_1 \geq \lambda_2 \geq \cdots \geq \lambda_d > 0$, and initialize $\mathbf{B}_0 = \mathbf{0}$. Under these conditions, all iterates in (6) remain diagonal. For each eigenvalue $\lambda_i$ of $\mathbf{A}$, we analyze the evaluation of the corresponding error component $[\mathbf{A} - \mathbf{B}_k]_{ii}$ by its continuous-time approximation $x^{(i)}(k)$ evolving as:

$$\frac{dx^{(i)}}{dt} = -\frac{1}{c_{\text{den}}}[x^{(i)}(t)]^2, \quad x^{(i)}(0) = \lambda_i.$$

---

**Algorithm 1** Randomized SR1 Update

---

1: **Requires:** Initial matrix $\mathbf{B}_0 \in \mathbb{R}^{d \times d}$, $0 \preceq \mathbf{B}_0 \preceq \mathbf{A}$, distribution $\mathcal{D}$.
2: **for** $k = 0, 1, 2 \ldots$ **do**
3:     Sample a random vector $\mathbf{s}_k \sim \mathcal{D}$ which satisfies (4).
4:     Compute $\mathbf{B}_{k+1} = \mathbf{SR1}(\mathbf{A}, \mathbf{B}_k, \mathbf{s}_k)$.
5: **end for**

---

The flow admits the closed-form solution $x^{(i)}(t) = \frac{\lambda_i}{1 + \lambda_i t / c_{\mathrm{den}}}$. In fact, this solution exhibits distinct convergence behaviors depending on the magnitude of the initial eigenvalues:

- For $\lambda_i \gg c_{\mathrm{den}}/t$, the error decreases rapidly as $x^{(i)}(t) \approx c_{\mathrm{den}}/t$;

- For $\lambda_i \ll c_{\mathrm{den}}/t$, the error remains nearly unchanged: $x^{(i)}(t) \approx \lambda_i$.

Thus, this demonstrates that the update rule (6) prioritizes components with large initial eigenvalues.

The following theorem, originating from this implicit preference for large eigenvalues, provides an upper bound on the approximation error in terms of the $\ell_2$-norm. Minimizing the $\ell_2$-norm equivalently imposes a uniform constraint on all eigenvalues of the error matrix, which confirms the algorithms implicit bias on the spectrum: the SR1 update prioritizes error reduction in large eigenspaces while having limited impact on small eigenspaces.

**Theorem 1.** *Suppose that $0 \preceq \mathbf{B}_0 \preceq \mathbf{A}$, $\mathbf{B}_k$ is produced by Algorithm 1, then for every $r \geq 3, 0 < p < 1$, there exists $K \in \mathbb{N}^*$ satisfying $K = \mathcal{O}\big(r(\ln r)^3 + \ln \frac{1}{p}\big)$, such that with probability at least $1 - p$, we have*

$$\|\mathbf{A} - \mathbf{B}_K\|_2 \leq \frac{\mathrm{Tr}(\mathbf{A} - \mathbf{B}_0)}{r}. \tag{7}$$

Theorem 1 shows that, with high probability, $\|\mathbf{A} - \mathbf{B}_k\|_2 = \tilde{\mathcal{O}}(\mathrm{Tr}(\mathbf{A})k^{-1})$.

The uniform bound of matrix $\ell_2$-norm is more challenging in the proof technically, which is corroborated in the related fields. Previous works on SR1 approximation have primarily focused on aggregate eigenvalue measurements, such as the trace function (Lin et al., 2022), the log-determinant barrier function (Ye et al., 2023), and the Frobenius norm (Jin & Mokhtari, 2023).

### 4.2 PROOF SKETCH OF THEOREM 1

Without loss of generality, in the proof we can set $\mathrm{Tr}(\mathbf{A} - \mathbf{B}_0) = 1$ by normalization. The difficulty of proving Theorem 1 arises from the uniform bound nature of the $\ell_2$- norm and the online algorithm nature of the SR1 update. These lead to one prominent challenge: the uniform spectral bound $\|\mathbf{A} - \mathbf{B}_k\|_2$ is not guaranteed to decrease sufficiently in one single iteration, regardless of $\mathbf{s}_k$'s selection. This challenge necessitates an analysis of the decrease at each individual eigenvalue. In fact, our proof investigates this fine-grained spectral analysis, and moreover, considers two different phases in the iteration with different decrease patterns.

To illustrate this, Figure 2 shows how the largest 5 eigenvalues of the approximation error matrix $\mathbf{A} - \mathbf{B}_k$ evolve empirically over iterations of SR1. This example considers the specific case where the initial matrix has multiple identical largest eigenvalues, as presented in 2 (a).

In this case, the largest eigenvalue does not decrease during the earlier steps. Besides, the spectrum distribution exhibits two stages during the iteration:
**Stage 1:**(2(b), Dispersion) Spectral gap emerges between large eigenvalues - the largest eigenvalue remains while the subsequent eigenvalues decrease;
**Stage 2:**(2(c), Normalization) The largest eigenvalue is reduced to achieve the uniform spectral decay.

This two-stage phenomenon results from that a rank-one update cannot simultaneously reduce multiple large eigenvalues in the approximation matrix. In fact, this case serves as the worst-case scenario in our analysis, and our proof also divides into two stages as the empirical results in Figure 2.

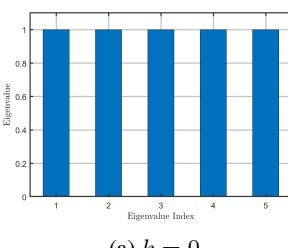 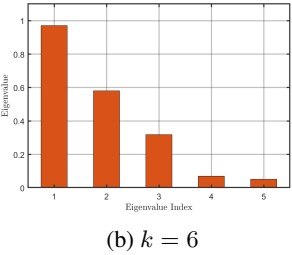 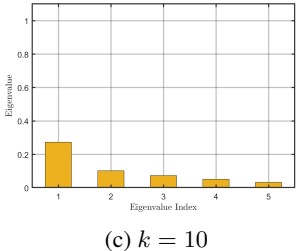

(a) $k = 0$        (b) $k = 6$        (c) $k = 10$

Figure 2: Top 5 Eigenvalues of $\mathbf{A} - \mathbf{B}_k$. X-axis: eigenvalue index, Y-axis: eigenvalue magnitude.

First, the following lemma formalizes the different decay rates of eigenvalues in Stage 1.

**Lemma 1** (Informal version of Lemma 5). *Under the conditions in Theorem 1, the $i$-th eigenvalue of the approximation error matrix enjoys a rate of $\tilde{\mathcal{O}}(1/\sqrt{ik})$ with high probability.*

In particular, focusing on the $k$-th eigenvalue demonstrates that $\lambda_k(\mathbf{A} - \mathbf{B}_t) \leq \frac{1}{4k}$ after $t = \tilde{\mathcal{O}}(k)$ iterations with high probability. However, this result does not establish the desired $\tilde{\mathcal{O}}(1/k)$ rate in Theorem 1. To improve this result, we consider two situations as below:

**Situation 1:** After $\tilde{\mathcal{O}}(k)$ iterations, the top $k$ eigenvalues are relatively close in magnitude. In other words, $\lambda_1/\lambda_k$ is bounded and the largest eigenvalue $\lambda_1$ is thereby bounded.

**Situation 2:** After $\tilde{\mathcal{O}}(k)$ iterations, the top $k$ eigenvalues are not uniformly distributed. In other words, there exist non-negligible gaps among the top $k$ eigenvalues.

Stage 2 exactly characterizes the reduction effect in the second situation: Using several iterations after stage 1, the SR1 update notably reduces large eigenvalues, while the left, which have already been reduced a lot, remains nearly the same. This stage inspires us to accelerate the decay rate of the dominant eigenvalue, based on the existence of gaps among the top eigenvalues. We prove the following lemma that quantifies the rate in detail:

**Lemma 2** (Informal version of Lemma 6). *Under the conditions corresponding to Theorem 1, for any $m \leq s \in \mathbb{N}^*, 0 < u < 1, r > 0$, and under mild conditions, there exists $K = \tilde{\mathcal{O}}(\frac{s^2 r}{m^2 u})$, such that $\lambda_m(\mathbf{A} - \mathbf{B}_K)$ will be smaller than $\max\{(1+u)\lambda_s(\mathbf{A} - \mathbf{B}_0), \frac{1}{r}\}$ iterations with high probability.*

Note that the conclusion in Lemma 2 can be applied to any starting matrix $\mathbf{B}_k$ and corresponding $\mathbf{B}_{k+K}$, because the SR1 update is an online learning process, and $\mathbf{B}_k$ satisfies $\text{tr}(\mathbf{A} - \mathbf{B}_k) \leq \text{tr}(\mathbf{A} - \mathbf{B}_0) \leq C_0$ (the condition in Theorem 1), it is only the matter of indices.

In particular, for any $r > 0$, if $m = \Theta(s), \frac{1}{u} = \tilde{\mathcal{O}}(1)$ and we have already reduced $\lambda_s(\mathbf{A} - \mathbf{B}_k)$ to the level of $\frac{1}{r(1+u)}$ for some $k \in \mathbb{N}*$, then $\lambda_m(\mathbf{A} - \mathbf{B}_{k+K}) \leq \frac{1}{r}$ where $K = \tilde{\mathcal{O}}(\frac{r}{u}) = \tilde{\mathcal{O}}(r) \cdot \tilde{\mathcal{O}}(1) = \tilde{\mathcal{O}}(r)$. Note that using Lemma 1 we need $\tilde{\mathcal{O}}(\frac{r^2}{m^2})$ iterations.

Theorem 1 is established by combining Lemma 1 and Lemma 2. First, by Lemma 1, $\lambda_{\lfloor r \rfloor}(\mathbf{A} - \mathbf{B}_{K_0}) \leq \frac{1}{4r}$ within $K_0 = \tilde{\mathcal{O}}(r)$ iterations with high probability. Then Lemma 2 enables an induction:

$$\lambda_{\lfloor r \rfloor}(\mathbf{A} - \mathbf{B}_{K_0}), \quad \lambda_{\lfloor r/2 \rfloor}(\mathbf{A} - \mathbf{B}_{K_0 + K_1}), \quad \cdots, \quad \lambda_1(\mathbf{A} - \mathbf{B}_{K_0 + K_1 + \cdots + K_t})$$

are all smaller than $\frac{1}{r}$ with high probability by induction, where $t = \lfloor \log_2 r \rfloor, K_i = \tilde{\mathcal{O}}(r)$.

The induction process is as follows. Take $u = \frac{1}{\log_2 r}, s = \lfloor r/2^i \rfloor, m = \lfloor r/2^{i+1} \rfloor$ in Lemma 2. Once $\lambda_s(\mathbf{A} - \mathbf{B}_{K_0 + \cdots + K_i}) \leq \frac{1}{4r}(1 + \frac{1}{\log_2 r})^i$, then we have $\lambda_m(\mathbf{A} - \mathbf{B}_{K_0 + \cdots + K_{i+1}}) \leq \frac{1}{4r}(1 + \frac{1}{\log_2 r})^{i+1}$ with high probability and ultimately we have $\lambda_1 \leq \frac{1}{4r}(1 + \frac{1}{\log_2 r})^t \leq \frac{1}{r}$. The total number of iterations of this process is $\sum_{i=0}^{t} K_i = (1 + t)\tilde{\mathcal{O}}(r) = \tilde{\mathcal{O}}(r)$. Then the proof of Theorem 1 is complete.

The proof of Lemma 1 is based on the observation that if the $i$-th eigenvalue remains larger than $\tilde{\mathcal{O}}(1/\sqrt{ik})$, the trace would decrease at an accelerated rate. The proof of Lemma 2 involves a delicate

---

**Algorithm 2** A General Framework of Regularized SR1 Method

---

1: **Requires:** Initial point $x_0 \in \mathbb{R}^d$, matrix $\mathbf{B}_0 \in \mathbb{R}^{d \times d}$, distribution $\mathcal{D}$, stepsize $\{r_k\}$, $l_2$ regularizer $\{\epsilon_k\} > 0$, Hessian correction term $\{\gamma_k, d_k\}$, subsequence $\{n_k\} \subseteq [N]$.

2: **for** $k = 0, 1, 2 \ldots N$ **do**

3:      $x_{k+1} = x_k - (\gamma_k \mathbf{B}_k + (\frac{1}{r_k} + \epsilon_k)\mathbf{I}_d)^{-1}(\nabla f(x_k) + \epsilon_k(x_k - x_0))$

4:      Sample a vector $\mathbf{s}_k \sim \mathcal{D}$ which satisfies (4)

5:      Compute $\mathbf{B}_{k+1} = d_k \mathbf{SR1}(\nabla^2 f(x_{n_k}), \mathbf{B}_k, \mathbf{s}_k)$

6: **end for**

---

construction of rational functions for comparison based on the eigenvalue structure of $\mathbf{A} - \mathbf{B}_k$. For formal lemmas and proof details, see Appendix B.

## 5 INSIGHT TOWARDS REGULARIZED SR1 METHOD

### 5.1 A REGULARIZED SR1 FRAMEWORK

We present the general framework of the regularized SR1 method in Algorithm 2. Our goal is to theoretically analyze its convergence behavior and establish a principled approach for parameter selection, leveraging the Hessian approximation efficiency results derived in Section 4. Algorithm 2 utilizes the following strategies common in practice:

1. Randomized SR1 and regularized Newton formulas, which forms the algorithm's basis.

2. $l_2$ regularization $\epsilon_k$ to enhance the stability of the algorithm.

3. Hessian correction term $\gamma_k, d_k$ to ensure $\gamma_k d_k \mathbf{B}_k \preceq \nabla^2 f(x_k)$.

4. Lazy Hessian strategy $n_k$ which only uses part of the exact Hessian.

The Hessian correction step aligns with the previous quasi-Newton convergence analyses (Liu & Luo, 2021; Rodomanov & Nesterov, 2021b). The $l_2$ regularization is a commonly adopted practical technique in optimization (Loshchilov & Hutter, 2019; Zhang et al., 2019). Note that Jiang & Mokhtari (2024) used extra gradient similar to $l_2$ regularization in their regularized quasi-Newton method. Sequence $\{n_k\}$ satisfied either $n_k = n_{k-1}$ or $n_k = k$, which means only changing the target Hessian in certain steps and thus reducing the Hessian computational cost (Doikov et al., 2023; Chen et al., 2025). We will show that with proper choice of parameters (could be decided in advance), Algorithm 2 enjoys an explicit global convergence rate.

### 5.2 COMPUTATIONAL COMPLEXITY

We briefly discuss the computational complexity of the inverse step $(\mathbf{B}_k + c_k \mathbf{I}_d)^{-1} \mathbf{v}_k$. In quasi-Newton regime, $\mathbf{B}_0$ is typically set to be easy to compute its inverse, usually $\mathbf{B}_0 = c\mathbf{I}_d$. Since $\mathbf{B}_k$ is constructed via a sequence of rank-one updates, it admits a factorization $\mathbf{B}_k = c\mathbf{I}_d + \mathbf{U}_k \mathbf{U}_k^\top$, where $\mathbf{U}_k \in \mathbb{R}^{d \times k}$. Applying the Sherman-Morrison-Woodbury formula, the inverse operation is reduced to inverting a $k \times k$ matrix, which costs $\mathcal{O}(k^3)$, along with matrix-vector multiplications costing $\mathcal{O}(kd)$. Thus, typically, the complexity in the $k$-th iteration is $\mathcal{O}(k^3 + kd)$, aligns with the previous research. However, as we will see, in our parameters scheme, where $c_k$ only change in a few steps, it is unnecessary to compute the inverse of a $k \times k$ matrix in every iteration. We will show that we can achieve the overall complexity of $\tilde{\mathcal{O}}(k^3 + k^2 d)$ in the first $k$ iterations instead of $\mathcal{O}(k^4 + k^2 d)$. For details, we refer the readers to Appendix C.3.

### 5.3 CONVERGENCE ANALYSIS

Let us first consider the quadratic case to get inspiration. That is, $f(x) = \frac{1}{2}x^\top \mathbf{A}x + b^\top x + c$, $\mathbf{A} \succeq 0$. The following lemma establishes an elementary one-step descent property of the iterative scheme.

**Lemma 3** (Quadratic optimization). *Suppose that $\nabla^2 f(x) \equiv \mathbf{A}$. In one step of Algorithm 2, if $0 \preceq \mathbf{B}_k \preceq \mathbf{A}$ and $0 \leq r_k \leq \frac{1}{\|\mathbf{A} - \mathbf{B}_k\|_2}$, then we have*

$$f_{\epsilon_k}(x_{k+1}) - f^*_{\epsilon_k} \leq \frac{1}{(1 + (\mu + \epsilon_k)r_k)^2}(f_{\epsilon_k}(x_k) - f^*_{\epsilon_k}), \tag{8}$$

*where $f_{\epsilon_k}(x) = f(x) + \frac{\epsilon_k}{2}\|x - x_0\|_2^2$ and $\mu = \lambda_{\min}(\mathbf{A})$.*

Given that $\|\mathbf{A} - \mathbf{B}_k\|_2$ decreases at a rate of $\tilde{\mathcal{O}}(\mathrm{Tr}(\mathbf{A})k^{-1})$, a practical choice implied by Lemma 3 is accordingly increase $r_k \sim \tilde{\Theta}(k/\mathrm{Tr}(\mathbf{A}))$. Let $x^* = \arg\min f$. If $r_k$ does so and satisfies the condition in Lemma 3, and $\epsilon_k \equiv \epsilon > 0, \gamma_k \equiv 1$, then after $k = \tilde{\Omega}(\mathrm{Tr}(\mathbf{A})^{\frac{1}{2}}\epsilon^{-\frac{1}{2}})$ iterations, $f_\epsilon(x_k) - f^*_\epsilon$ starts to contract at least in a linear rate of $1 - \Omega(\epsilon^{\frac{1}{2}}\mathrm{Tr}(\mathbf{A})^{-\frac{1}{2}})$. Hence, on one hand, for quadratic convex problem, SR1 update leads $f_\epsilon$ to achieve an $\epsilon$ approximate minimizer within $\tilde{\mathcal{O}}(\mathrm{Tr}(\mathbf{A})^{\frac{1}{2}}\epsilon^{-\frac{1}{2}})$ iterations. On the other hand, $f^*_\epsilon - f^* \leq f_\epsilon(x^*) - f^* = \frac{\epsilon}{2}\|x^*\|_2^2$. Combining these two we obtain $\tilde{\mathcal{O}}(\mathrm{Tr}(\mathbf{A})k^{-2})$ convergence rate. The proof for Lemma 3 and both local and global convergence for quadratic functions are postponed to Appendix C.1.

Now we shift our focus to general convex self-concordant functions. To clarify our motivation, for the moment let $\mathrm{Tr}(\nabla^2 f(x)) \leq 1$ by normalizing. We regard Algorithm 2 as an approximation of optimizing quadratic forms with Hessian $\nabla^2 f + \epsilon_k \mathbf{I}_d$, where $\epsilon_k \mathbf{I}_d$ is decaying $l_2$ regularized factor.

The key observation is stated as below :

1. If $f - f^* \sim \mathcal{O}(\epsilon_k)$, the Hessian perturbation will be no more than $\mathcal{O}(\sqrt{\epsilon_k})$.
2. If the Hessian's perturbation is controlled at the level of $\sqrt{\epsilon_k}$, the Hessian approximation will satisfy $\|\nabla^2 f - \mathbf{B}_k\|_2 \sim \mathcal{O}(\sqrt{\epsilon_k})$ in at most $\tilde{\mathcal{O}}(\epsilon_k^{-\frac{1}{2}})$ iterations. Then, Lemma 3 implies $f - f^* \sim \mathcal{O}(\epsilon_k)$ within at most $\tilde{\mathcal{O}}(\epsilon_k^{-\frac{1}{2}})$ iterations.

This basically answers the question why regularized QN method could exhibit faster convergence than vanilla gradient descent and inspires us to gradually decrease $\epsilon_k$ at a moderate linear rate to maintain these two conditions in practical algorithm designs.

Guided by this observation, we can design an easy-to-practice parameter scheme and prove a global convergence rate of $\tilde{\mathcal{O}}(d_{\mathrm{eff}}^2 L^2 M^2 D^4/k^2)$, where $d_{\mathrm{eff}} = \frac{\sup \mathrm{Tr}(\nabla^2 f(x))}{\sup \|\nabla^2 f(x)\|_2}$.

**Theorem 2** (General convex optimization). *Under Assumption 1, 2, 3, there exists an explicit choice of parameters in Algorithm 2 such that for every $\varepsilon > 0$, we only need at most $\tilde{\mathcal{O}}\left(d_{\mathrm{eff}}L(M+1)D^2(M+\varepsilon^{-\frac{1}{2}})\right)$ iterations in Algorithm 2 to obtain a solution $z$ such that $f(z) - f^* \leq \varepsilon$ with high probability.*

For details on how to choose parameters, we refer readers to Appendix C.2.4. The proof of Theorem 2 is postponed to Appendix C.2.

## 5.4 APPLICATION IN HIGH-DIMENSIONAL SCENARIOS

It is supported by both theoretical and empirical evidence that in many high-dimensional optimization problems, the maximal Hessian's trace $T$ is guaranteed to be small, such as general kernels (Terras, 1999; Gu & Gu, 2013; Zhang et al., 2015; Blanchard & Mücke, 2018), random feature model (Rahimi & Recht, 2007; Bach, 2017), neural tangent kernel (Bietti & Bach, 2021; Hu et al., 2021). In this situation, under Assumption 1, 3, Algorithm 2 can achieve a global convergence rate of $\tilde{\mathcal{O}}(1/k^2)$ where the constant does not explicitly depend on $d$.

A concrete example is the fundamental problem of empirical risk minimization problem over a generalized linear model, with the objective $f(x) = \frac{1}{n}\sum_{i=1}^n f_i(a_i^\top x)$. Conventional assumptions assumes that for each $i \in \{1, 2, \cdots, n\}$, the data $a_i$ is normalized to $\|a_i\|^2 \leq R^2$ (results from the common data normalization processing) and $f_i \in C^2$ is convex and $L_0$-smooth. Thus the Hessian trace of $f$ is bounded by

$$\mathrm{tr}\left(\nabla^2 f(x)\right) = \frac{1}{n}\sum_{i=1}^n f_i''(a_i^\top x)\mathrm{tr}(a_i a_i^\top) \leq \sum_{i=1}^n \frac{L_0}{n}\|a_i\|^2 \leq L_0 R^2.$$

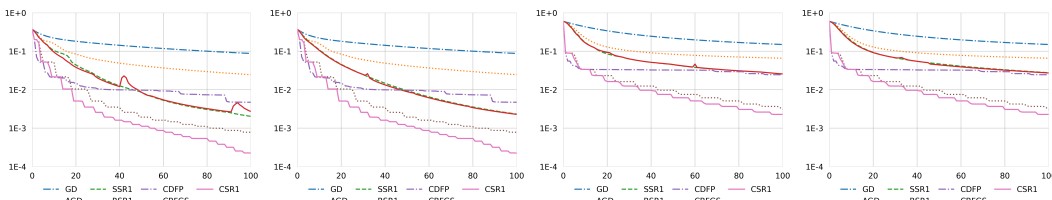

(a) a9a, $\rho = 0.3$, $c = 0.1$. (b) a9a, $\rho = 0.9$, $c = 0.01$. (c) w8a, $\rho = 0.3$, $c = 0.1$. (d) w8a, $\rho = 0.9$, $c = 0.01$.

Figure 3: Iteration numbers vs. $f(x) - f(x^*)$.

In low-precise regime, when $k \ll d$, the computational complexity of inverse step simplifies to $\mathcal{O}(k^2 d)$. The gradient computation itself usually reaches $\mathcal{O}(kd^2)$ (quadratic case) or $\mathcal{O}(knd)$ where $n$ is the number of samples. Therefore, the efficiency of regularized SR1 method is comparable to AGD in high-dimensional problems with bounded Hessian traces.

## 6 EXPERIMENTS

In this section, we present illustrations on the efficiency of the SR1 methods in the regime before superlinear convergence in the logistic regression tasks formulated by

$$\min_{x \in \mathbb{R}^d} f(x) = \frac{1}{n} \sum_{i=1}^{n} \log \left( 1 + \exp(-b_i a_i^\top x) \right),$$

where $d = 300, n = 49749$ for w8a dataset and $d = 123, n = 32561$ for a9a dataset. We conduct experiments on our regularized SR1 algorithm with randomized update (RSR1) and the secant equation update (SSR1), gradient descent (GD), accelerated gradient descent (AGD) and three classical quasi-newton methods (CSR1, CBFGS, CDFP, where 'C' refers to 'classical').

For data preprocessing, we normalize the feature vectors to improve the condition number of the optimization problem. For GD and AGD, our parameters are selected through grid search. Specifically, learning rates are chosen from the set $\{k \times 10^t : k = 1, 2, 5, t = -2, -1, 0, 1\}$, and momentum coefficients are selected from $\{0.9, 0.95, 0.99, 0.999\}$. For three classical quasi-newton methods, the learning rates are selected by exact line search. For Algorithm 2, our parameters are set as described in the phase 3 of Table 1 regarding $L = 1$, where $\varepsilon_0 = 1, S_0 = 0$, $\varepsilon_t = \rho^t \varepsilon_0, S_{t+1} - S_t = c/\sqrt{\varepsilon_t}, \alpha_t = 1, \beta_t = (1 - \sqrt{\varepsilon_t})^2, \eta_t = \frac{1}{4\sqrt{\varepsilon_t}}$ and $c, \rho$ are tuning hyper-parameters listed above. All experiments are repeated multiple times to ensure the stability of Algorithm 2. To plot $f(x) - f(x^*)$ as the vertical axis, we approximate $f(x^*)$ by the loss at iteration 500 of the best-performing algorithm.

We run simulations with 100 iterations, which is fewer than the problem dimension; as a result, the SR1 method does not enter the superlinear convergence regime. The results in Figure 3 demonstrate the following several implications: (1) SR1 methods consistently outperform vanilla gradient descent, confirming the effectiveness of Hessian approximation preconditioning even outside the superlinear convergence regime; (2) the SR1 method achieves convergence rates comparable to or better than AGD, supporting our worst-case guarantee of $\tilde{\mathcal{O}}\left(\frac{1}{k^2}\right)$.

## 7 DISCUSSION

This paper elucidates the mechanism underlying the global convergence of quasi-Newton (QN) methods in high-dimensional settings. By characterizing the algorithmic bias in the spectral evolution during Hessian approximation, we establish a favorable approximation quality under randomized SR1 updates. Besides our proposed framework, we believe that the implicit bias viewpoint could be taken into consideration in many other Hessian approximation methods, improve local superlinear convergence rate by diminishing the dependence on the dimension and be applied to a various of inexact Newton methods combining with other techniques such as cubic and acceleration.

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

# Appendix

## A NOTATION AND THEORY STRUCTURE

First, we provide the necessary notation that appears in the Appendix. Denote $\lambda_{\max}(\mathbf{A}) = \lambda_1(\mathbf{A}) \geq \cdots \geq \lambda_d(\mathbf{A})$ as the eigenvalues of a real symmetric matrix $\mathbf{A} \in \mathbb{R}^{d \times d}$. For a positive definite matrix $\mathbf{A}$, we can endow $\mathbb{R}^d$ with conjugate Euclidean norms:

$$\|x\|_{\mathbf{A}} \stackrel{\text{def}}{=} \sqrt{x^\top \mathbf{A} x}, \quad \|x\|_{\mathbf{A}}^* \stackrel{\text{def}}{=} \sqrt{x^\top \mathbf{A}^{-1} x}, x \in \mathbb{R}^d.$$

The corresponding matrix norm for a matrix $\mathbf{H}$ is

$$\|\mathbf{H}\|_{\mathbf{A}} \stackrel{\text{def}}{=} \max_{x \neq 0} \frac{\|\mathbf{H} x\|_{\mathbf{A}}}{\|x\|_{\mathbf{A}}} = \left\| \mathbf{A}^{\frac{1}{2}} \mathbf{H} \mathbf{A}^{-\frac{1}{2}} \right\|_2.$$

Throughout the paper, $x_0$ is the initial point when optimizing (1). For any convex function $g$, provided that there is no ambiguity with the reference function $g$, we denote $x^*$ as any minimizer of $g$, $g^* = g(x^*)$ and for any $\varepsilon_t > 0$, denote $g_{\varepsilon_t}(x) = g(x) + \frac{\varepsilon_t}{2} \|x - x_0\|_2^2$, which is strong convex and has unique minimizer $x_{\varepsilon_t}^*$ with $g_{\varepsilon_t}(x_{\varepsilon_t}^*) = g_{\varepsilon_t}^*$. Moreover, for any $x, y \in \mathbb{R}^d, \varepsilon_t \in \mathbb{R}$, we denote

$$\|x\|_y = \sqrt{x^\top \nabla^2 g(y) x}, \quad \|x\|_y^{\varepsilon_t} = \sqrt{x^\top \nabla^2 g_{\varepsilon_t}(y) x}.$$

In the Appendix, $g$ could be the objective $f$ or its scaling $af$. If $f$ satisfies Assumption 2, 3, then we know that $\sup \|\nabla^2 f(x)\|_2 \leq L$ and we denote $d_{\text{eff}} = \sup (\nabla^2 f(x))/L$. Denote $C$ as the absolute constant of the $\mathcal{O}(\cdot)$ term of Theorem 1.

Next, we present the structure of proofs and lemmas that appear in the main body of the paper. Figure 4 shows the relations between main theorems and key lemmas. An arrow from block $A$ to $B$ means that the proof of $B$ needs $A$. Some technical lemmas that are not essential are not listed in figure 4. Theorem 6 is the detailed version of the key part of Theorem 2. Theorem 4 is the formal version of 1. Lemma 5 and Lemma 6 are the formal versions of Lemma 1 and Lemma 2, respectively.

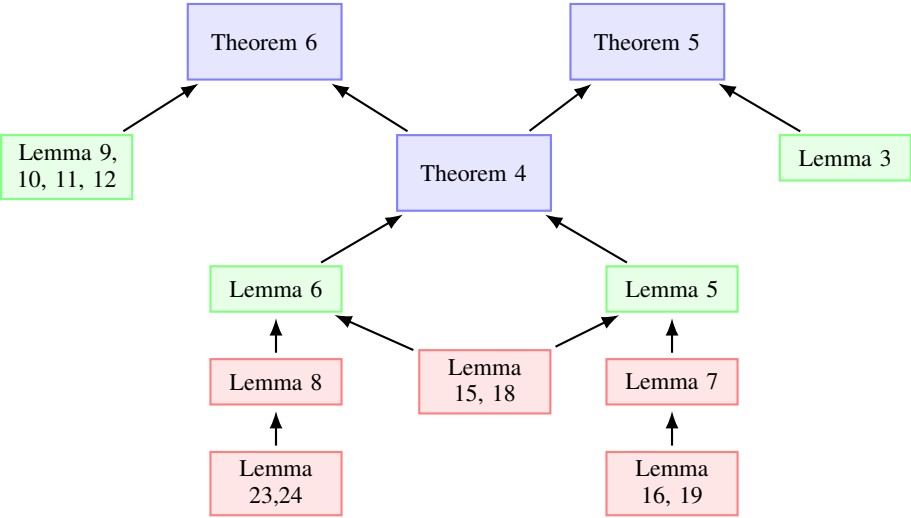

Figure 4: Relations between the main results in Appendix.

## B POSTPONED PROOFS IN SECTION 4

### B.1 PRELIMINARIES

Before proving the Hessian approximation results in Section 4, we present some preliminary results about the basic properties of SR1 update (Lemma 4) and a relaxation to our assumption (Theorem 3)

Lemma 4 states a basic property of the SR1 update. The matrix approximations $\mathbf{A} - \mathbf{B}_k$ exhibit monotonically decreasing eigenvalues. Given Weyls inequality (Lemma 21), $\{\lambda_i(\mathbf{A} - \mathbf{B}_k)\}_k$ monotonically decreases with respect to $k$, for all $1 \leq i \leq d$.

**Lemma 4** (Monotonically decreasing matrices). *If $\mathbf{A} \succeq \mathbf{B} \succeq 0$, then for any $\mathbf{u} \in \mathbb{R}^d$ such that $\mathbf{u}^\top (\mathbf{A} - \mathbf{B})\mathbf{u} \neq 0$, we have*

$$\mathbf{B} \preceq \mathbf{SR1}(\mathbf{A}, \mathbf{B}, \mathbf{u}) \preceq \mathbf{A}.$$

*Proof of Lemma 4.* For any $\mathbf{v} \in \mathbb{R}^d$, since $\mathbf{A} - \mathbf{B} \succeq 0$, we have

$$\mathbf{v}^\top \mathbf{SR1}(\mathbf{A}, \mathbf{B}, \mathbf{u})\mathbf{v} = \mathbf{v}^\top \mathbf{B}\mathbf{v} + \frac{(\mathbf{v}^\top (\mathbf{A} - \mathbf{B})\mathbf{u})^2}{\mathbf{u}^\top (\mathbf{A} - \mathbf{B})\mathbf{u}} \geq \mathbf{v}^\top \mathbf{B}\mathbf{v}.$$

On the other hand, by the Cauchy-Schwartz inequality, we have

$$\mathbf{v}^\top (\mathbf{A} - \mathbf{B})\mathbf{v} \cdot \mathbf{u}^\top (\mathbf{A} - \mathbf{B})\mathbf{u} \geq (\mathbf{v}^\top (\mathbf{A} - \mathbf{B})\mathbf{u})^2.$$

Hence,

$$\mathbf{v}^\top \mathbf{SR1}(\mathbf{A}, \mathbf{B}, \mathbf{u})\mathbf{v} \leq \mathbf{v}^\top \mathbf{B}\mathbf{v} + \frac{(\mathbf{u}^\top (\mathbf{A} - \mathbf{B})\mathbf{u})}{\mathbf{u}^\top (\mathbf{A} - \mathbf{B})\mathbf{u}} \cdot \mathbf{v}^\top (\mathbf{A} - \mathbf{B})\mathbf{v} \leq \mathbf{v}^\top \mathbf{A}\mathbf{v}.$$

The claimed result then follows from the above inequalities. $\qquad\square$

Theorem 3 below could be used to give a stronger version of Theorem 2: If a constant number of eigenvalues are large while the sum of the rest is bounded, then the trace decreases to a constant level within $\tilde{\mathcal{O}}(1)$ iterations.

**Theorem 3.** *Suppose that $\mathbf{B}_0$, $\mathbf{B}_k$ is produced by Algorithm 1, then for every $k \in \mathbb{N}^*$ and $0 < p < 1$, if $\sum_{i=k+1}^{d} \lambda_i(\mathbf{A}) \leq T_k$, then there exists $K_1 = \mathcal{O}\left(k \ln\left(\frac{\text{tr}\mathbf{A}}{T_k}\right) + \ln \frac{1}{p}\right)$, such that with probability at least $1 - p$, we have*

$$\text{tr}(\mathbf{A} - \mathbf{B}_{K_1}) \leq 2T_k.$$

*Proof of Theorem 3.* Denote $\mathbf{R}_t = \mathbf{A} - \mathbf{B}_t$ and $\text{tr}(\mathbf{R}_t) = b_t$ for simplicity. Let $T_k = \sum_{i=k+1}^{d} \lambda_i(\mathbf{R}_0)$. Define

$$\beta_t^{(j)} = \frac{\sum_{i=1}^{j} \lambda_i(\mathbf{R}_t)}{b_t}, \quad 1 \leq j \leq d.$$

We only need to condition on the process when $\beta_t^{(k)} \geq \frac{1}{2}, 0 \leq t \leq K$. Otherwise, for some $t$, we have $b_t \leq 2\sum_{i=k+1}^{d} \lambda_i(\mathbf{R}_t) \leq 2\sum_{i=k+1}^{d} \lambda_i(\mathbf{R}_0)$, then the proof is finished. Let

$$\mathcal{A}_t \stackrel{\text{def}}{=} \left\{ \text{tr}(\mathbf{R}_{t+1}) \leq \text{tr}(\mathbf{R}_t) - C_3 \frac{\text{tr}(\mathbf{R}_t^2)}{\text{tr}(\mathbf{R}_t)} \right\}, \quad 0 \leq t \leq K.$$

By Lemma 7 we have $\mathbb{P}(\mathcal{A}_t) \geq C_4$. By Cauchy-Schwartz inequality, we have

$$\text{tr}(\mathbf{R}_t^2) \geq \frac{1}{k}\left(\sum_{i=1}^{k} \lambda_i(\mathbf{R}_t)\right)^2 = \frac{(\beta_t^{(k)})^2 b_t^2}{k} \geq \frac{b_t^2}{4k}.$$

Therefore, if $\mathcal{A}_t$ is true, then

$$b_{t+1} \leq b_t - \frac{C_3}{4k}b_t.$$

Choose $K = \lceil \frac{8}{C_4} \max\left\{ \frac{k \ln \frac{b_0}{T_k}}{C_3}, \ln \frac{1}{p} \right\} \rceil$. Note that $\{b_t\}$ does not increase and if

$$|\{0 \leq t \leq K : \mathcal{A}_t \text{ is true}\}| \geq \frac{C_4 K}{2} \geq \frac{4k \ln \frac{b_0}{T_k}}{C_3},$$

(here $|A|$ means the cardinality of a set $A$) then

$$b_K \leq b_0 \left(1 - \frac{C_3}{4k}\right)^{\frac{4k}{C_3} \ln b_0 T_k} \leq T_k.$$

This implies $\mathrm{tr}(\mathbf{R}_K) \leq T_k$. Lemma 18 yields that

$$\mathbb{P}\left(|\{0 \leq t \leq K : \mathcal{A}_t \text{ is true}\}| \leq \frac{C_4 K}{2}\right) \leq e^{-\frac{C_4 K}{8}} \leq p.$$

Hence, with probability at least $1 - p$, we have $\mathrm{tr}(\mathbf{A} - \mathbf{B}_K) \leq 2T_k$. $\qquad\square$

Since the relationship between $\mathbf{A}$ and $\mathbf{B}_k$ is invariant under simultaneous scaling by a constant factor. We set $\mathrm{tr}(\mathbf{A} - \mathbf{B}_0) \leq 1$ without loss of generality in the remaining proof of Appendix B. Then the claimed upper bound in Theorem 4 can be obtained by rescaling a factor $\mathrm{tr}(\mathbf{A} - \mathbf{B}_0)$.

## B.2 FORMAL STATEMENT AND PROOFS

Below, we present the deferred proofs of Lemma 1 and Lemma 2. We begin by formally stating Lemma 1.

**Lemma 5** (Formal version of Lemma 1). *Suppose that* $\mathrm{tr}(\mathbf{A} - \mathbf{B}_0) \leq 1$, *then for every* $1 \leq k \leq d$, $m > 0$, $0 < p < 1$, *there exists* $K = \mathcal{O}\left(m \ln m + \ln \frac{1}{p}\right) \in \mathbb{N}$ *such that with probability at least* $1 - p$, $\lambda_k(\mathbf{A} - \mathbf{B}_K) \leq \frac{1}{\sqrt{km}}$.

*Proof of Lemma 5.* Denote $\mathbf{R}_t = \mathbf{A} - \mathbf{B}_t$ and $\mathrm{tr}(\mathbf{R}_t) = b_t$ for simplicity. Define

$$\beta_t^{(j)} = \frac{\sum_{i=1}^{j} \lambda_i(\mathbf{R}_t)}{b_t}, \quad 1 \leq j \leq d.$$

Since $b_t \leq 1$, we only need to condition on the process when $\beta_t^{(k)} \geq \sqrt{\frac{k}{m}}$, $0 \leq t \leq K$. Otherwise, for some $t$, we have $\lambda_k(\mathbf{R}_t) \leq \frac{1}{k} b_t \beta_t^{(k)} \leq \frac{1}{\sqrt{km}}$, then the proof is finished. Let

$$\mathcal{A}_t \stackrel{\text{def}}{=} \left\{\mathrm{tr}(\mathbf{R}_{t+1}) \leq \mathrm{tr}(\mathbf{R}_t) - C_3 \frac{\mathrm{tr}(\mathbf{R}_t^2)}{\mathrm{tr}(\mathbf{R}_t)}\right\}, \quad 0 \leq t \leq K.$$

By Lemma 7 we have $\mathbb{P}(\mathcal{A}_t) \geq C_4$. By Cauchy-Schwartz inequality, we have

$$\mathrm{tr}(\mathbf{R}_t^2) \geq \frac{1}{k}\left(\sum_{i=1}^{k} \lambda_i(\mathbf{R}_t)\right)^2 = \frac{(\beta_t^{(k)})^2 b_t^2}{k} \geq \frac{b_t^2}{m}.$$

Therefore, if $\mathcal{A}_t$ is true, then

$$b_{t+1} \leq b_t - \frac{C_3}{m} b_t.$$

Choose $K = \lceil \frac{8}{C_4} \max\left\{\frac{m \ln m}{4C_3}, \ln \frac{1}{p}\right\}\rceil$. Note that $\{b_t\}$ does not increase and if

$$|\{0 \leq t \leq K : \mathcal{A}_t \text{ is true}\}| \geq \frac{C_4 K}{2} \geq \frac{m \ln m}{C_3},$$

then

$$b_K \leq b_0 \left(1 - \frac{C_3}{m}\right)^{\frac{m}{C_3} \ln m} \leq \frac{1}{m}.$$

This implies $\lambda_k(\mathbf{R}_K) \leq \frac{1}{km} \leq \frac{1}{\sqrt{km}}$. Lemma 18 yields that

$$\mathbb{P}\left(|\{0 \leq t \leq K : \mathcal{A}_t \text{ is true}\}| \leq \frac{C_4 K}{2}\right) \leq e^{-\frac{C_4 K}{8}} \leq p.$$

Hence, with probability at least $1 - p$, we have $\lambda_k(\mathbf{A} - \mathbf{B}_K) \leq \frac{1}{\sqrt{km}}$. $\qquad\square$

In the following, we formally state Lemma 2, which proves the improved decay rate of the eigenvalues based on the eigengaps.

**Lemma 6** (Formal version of Lemma 2). *Suppose that* $\operatorname{tr}(\mathbf{A} - \mathbf{B}_0) \leq 1$, *for any* $1 \leq m < s < d$, *if* $\lambda_m(\mathbf{A} - \mathbf{B}_0) \leq \frac{1}{s+1}$, *then for every* $r > 0, p, u \in (0,1)$, *there exists* $K = \mathcal{O}\left(\frac{s^2 r}{um^2} \ln \frac{s}{u} + \ln \frac{1}{p}\right) \in \mathbb{N}$ *such that with probability at least* $1 - p$, *one of the following statements must hold:*

$$\lambda_m(\mathbf{A} - \mathbf{B}_K) \leq \frac{1}{r}, \tag{9}$$

$$\exists 1 \leq i \leq K, \lambda_m(\mathbf{A} - \mathbf{B}_i) \leq (1 + u)\lambda_{s+1}(\mathbf{A} - \mathbf{B}_i). \tag{10}$$

*Proof of Lemma 6.* We only need to condition on the process that satisfies for all $1 \leq k \leq K$, $\lambda_m(\mathbf{A} - \mathbf{B}_k) > (1 + u)\lambda_{s+1}(\mathbf{A} - \mathbf{B}_k)$, otherwise (10) holds, and we finish the proof. Then by Lemma 8, there exist constant $C_5, C_6 > 0$ such that for each $1 \leq k \leq K$, with probability at least $C_5$, we have

$$\operatorname{tr}(\mathbf{A} - \mathbf{B}_{k+1})_s \leq \operatorname{tr}(\mathbf{A} - \mathbf{B}_k)_s - \frac{umC_6}{s^2 \ln \frac{s}{u}} \left(\operatorname{tr}(\mathbf{A} - \mathbf{B}_k)_s\right)^2, \tag{11}$$

where $\operatorname{tr}(\mathbf{H})_s$ means the sum of the top $s$ eigenvalues of $\mathbf{H}$.

Choose $K = \lceil \frac{8}{C_5} \max\left\{\frac{s^2 r}{4um^2 C_6} \ln \frac{s}{u}, \ln \frac{1}{p}\right\}\rceil$ and let

$$\mathcal{B}_k \stackrel{\text{def}}{=} \left\{\operatorname{tr}(\mathbf{A} - \mathbf{B}_{k+1})_s \leq \operatorname{tr}(\mathbf{A} - \mathbf{B}_k)_s - \frac{umC_6}{s^2 \ln \frac{s}{u}} \left(\operatorname{tr}(\mathbf{A} - \mathbf{B}_k)_s\right)^2\right\}, \quad 1 \leq k \leq K.$$

Lemma 18 yields that

$$\mathbb{P}\left(\left|\{0 \leq t \leq K : \mathcal{A}_t \text{ is true}\}\right| \leq \frac{C_5 K}{2}\right) \leq e^{-\frac{C_5 K}{8}} \leq p.$$

Hence, with probability at least $1 - p$, we have

$$\left|\{0 \leq t \leq K : \mathcal{B}_t \text{ is true}\}\right| \geq \frac{C_5 K}{2} \geq \frac{s^2 r}{um^2 C_6} \ln \frac{s}{u}.$$

Note that $\operatorname{tr}(\mathbf{A} - \mathbf{B}_k)_s$ does not increase, as a result of Lemma 15, we have

$$\lambda_m(\mathbf{A} - \mathbf{B}_K) \leq \frac{1}{m}\operatorname{tr}(\mathbf{A} - \mathbf{B}_K)_s \leq \frac{1}{m} \cdot \frac{1}{\frac{C_5 K}{2} \frac{umC_6}{s^2 \ln \frac{s}{u}}} \leq \frac{1}{m} \cdot \frac{m}{r} \leq \frac{1}{r}.$$

$\square$

By leveraging Lemma 5 and Lemma 6, we can complete the proof of Theorem 1. Concretely , we prove the following generalized version of Theorem 1.

**Theorem 4** (General version of Theorem 1). *Suppose that* $0 \preceq \mathbf{B}_0 \preceq \mathbf{A}$, $\mathbf{B}_k$ *is produced by Algorithm 1, then for every* $\delta, p, u \in (0,1), r \geq 3$, *there exists* $K \in \mathbb{N}$ *satisfying*

$$K = \mathcal{O}\left(\frac{1}{\delta u}(1 + u)^{\frac{4}{\delta}} r^{1+\delta} \ln \frac{r}{u} + \ln \frac{1}{\delta p}\right),$$

*such that with probability at least* $1 - p$, *we have*

$$\|\mathbf{A} - \mathbf{B}_K\|_2 \leq \frac{\operatorname{tr}(\mathbf{A} - \mathbf{B}_0)}{r}. \tag{12}$$

*Specifically, choose* $\delta = u = \frac{1}{\ln r}$, *then* $K = \mathcal{O}\left(r(\ln r)^3 + \ln \frac{1}{p}\right)$.

*Proof of Theorem 4.* First, let us assume $C_0 = 1$. We use Lemma 5 to give an initial bound. Let $L_1$ be the constant term in Lemma 5. Set $m \to L_1\left(4(1 + u)^{\frac{4}{\delta}} r^{1+\delta} \ln \frac{r}{u} + \ln \frac{3r}{p}\right)$ in Lemma 5. Then for each $1 \leq k \leq r$, with probability at least $1 - \frac{p}{2r}$, we have

$$\lambda_k(\mathbf{A} - \mathbf{B}_m) \leq \frac{1}{2(1 + u)^{\frac{2}{\delta}}\sqrt{kr^{1+\delta}}}. \tag{13}$$

Thus, with probability at least $1 - \frac{p}{2r}$, (13) holds for all $1 \leq k \leq r$. Denote $t = \lceil \frac{\delta}{2} \rceil$, for all $1 \leq i \leq t, i \in \mathbb{N}$, define

$$r_i \stackrel{\text{def}}{=} (1+u)^{t+1-i} r, s_i \stackrel{\text{def}}{=} \lceil r^{1-\frac{(1+i)\delta}{2}} \rceil, m_i \stackrel{\text{def}}{=} s_{i+1}, Q_i \stackrel{\text{def}}{=} \frac{s_i^2 r_i}{um_i^2} \ln \frac{s_i}{u} + \ln \frac{2t}{p}.$$

We consider the following non-negative integer-valued random variables:

$$K_0 = m, K_i \stackrel{\text{def}}{=} \min \left\{ k \in \mathbb{N}, k \geq K_{i-1} : \lambda_{m_i}(\mathbf{A} - \mathbf{B}_{K_i}) \leq \frac{1}{r_{i+1}} \right\}, \quad 1 \leq i \leq t.$$

We claim that there exists a constant $C_7$, such that conditioned on (13), for each $1 \leq i \leq t$, with probability at least $1 - \frac{p}{2t}$, we have $K_i - K_{i-1} \leq C_7 Q_i$. The proof of this claim is as follows.

1. **Step 1:** First, Let's check that $\lambda_{m_i}(\mathbf{A} - \mathbf{B}_{K_{i-1}}) \leq \frac{1}{1+s_i}$. Since $K_{i-1} \geq m$, we have

$$\lambda_{m_i}(\mathbf{A} - \mathbf{B}_{K_{i-1}}) \leq \lambda_{m_i}(\mathbf{A} - \mathbf{B}_m) \leq \frac{1}{2\sqrt{m_i r^{1+\delta}}}.$$

It suffices to prove that
$$2\sqrt{m_i r^{1+\delta}} \geq 1 + s_i.$$
Taking the value of $m_i, s_i$ into the above inequality, we only need to prove
$$2r^{1-\frac{i}{4}\delta} \geq 1 + 1 + r^{1-\frac{i+1}{2}\delta}.$$
It holds since $\frac{i}{4} \leq \frac{1+i}{2}$ for all $i \in \mathbb{N}$.

2. **Step 2:** Utilize Lemma 6 to prove our claim. Let $C_7$ be the constant term in Lemma 6. Set
$$K \to C_7 Q_i, r \to r_i, s \to s_i, m \to m_i, p \to \frac{p}{2t}$$
in Lemma 6. Then with probability at least $1 - \frac{p}{2t}$, we have either
$$\lambda_{m_i}(\mathbf{A} - \mathbf{B}_{K_{i-1}+K}) \leq \frac{1}{r_i} \leq \frac{1}{r_{i+1}},$$
or
$$\exists 1 \leq i \leq K, \lambda_{m_i}(\mathbf{A} - \mathbf{B}_{K_{i-1}+i}) \leq (1+u)\lambda_{s_i+1}(\mathbf{A} - \mathbf{B}_{K_{i-1}+i}) \leq \frac{1+u}{r_i} = \frac{1}{r_{i+1}}.$$

In both cases, we have $\lambda_{m_i}(\mathbf{A} - \mathbf{B}_{K_{i-1}+K}) \leq \frac{1}{r_{i+1}}$. Therefore, $K_i \leq K_{i-1} + K$ and this leads to our claim.

From our claim we know that with probability at least $1 - \frac{p}{2} - \sum_{i=1}^{t} \frac{p}{2t} = 1 - p$, we have

$$K_t \leq K_0 + C_7 \sum_{i=1}^{t} Q_i \leq K_0 + tC_7 \max_{1 \leq i \leq t} \left\{ \frac{s_i^2 r_i}{um_i^2} \ln \frac{s_i}{u} + \ln \frac{2t}{p} \right\}$$

$$\leq K_0 + tC_7 \left( \frac{\lceil r^\delta \rceil r}{u} \ln \frac{r}{u} + \ln \frac{2t}{p} \right)$$

$$\leq K_0 + C_7 \lceil \frac{2}{\delta} \rceil \left( \frac{\lceil r^{1+\delta} \rceil}{u} \ln \frac{r}{u} + \ln \frac{2+\delta}{\delta p} \right)$$

Choose $C = \max\{L_1, C_7\}$ and then we finish the proof when $C_0 = 1$. For general cases, by rescaling then the proof is complete. □

**Corollary 1.** *Denote $T_k = \sum_{i=k+1}^{d} \lambda_i(\mathbf{A} - \mathbf{B}_0)$. Suppose that $0 \preceq \mathbf{B}_0 \preceq \mathbf{A}$, $\mathbf{B}_k$ is produced by Algorithm 1, then for every $p \in (0,1), r \geq 3, 0 \leq k \leq d$, there exists $K \in \mathbb{N}$ satisfying*
$$K = \mathcal{O} \left( k \ln \frac{T_0}{T_k} + r (\ln r)^3 + \ln \frac{1}{p} \right),$$
*such that with probability at least $1 - p$, we have*
$$\|\mathbf{A} - \mathbf{B}_K\|_2 \leq \frac{T_k}{r}. \tag{14}$$

*Proof.* By Theorem 3, after $K_1 = \mathcal{O}\left(k \ln \frac{T_0}{T_k} + \ln \frac{2}{p}\right)$ iterations, we have $\text{tr}(\mathbf{A} - \mathbf{B}_{K_1}) \leq 2T_k$ with probability at least $1 - \frac{p}{2}$. Restart from $K_1$, by Theorem 4, after $K_2 = \mathcal{O}\left(r \left(\ln r\right)^3 + \ln \frac{2}{p}\right)$ iterations, we have $\|\mathbf{A} - \mathbf{B}_{K_1+K_2}\|_2 \leq \frac{\text{tr}(\mathbf{A}-\mathbf{B}_{K_1})}{2r} \leq \frac{T_k}{r}$ with probability at least $1 - \frac{p}{2}$. Then taking $K = K_1 + K_2$ finishes the proof. $\qquad \square$

## C   POSTPONED PROOFS IN SECTION 5

### C.1   PROOF OF CONVERGENCE RATE FOR QUADRATIC FUNCTIONS

*Proof of Lemma 3.* First consider the case when $\epsilon_k = 0$. Since $f$ is quadratic, we have

$$f(x) = \frac{1}{2}(x - x^*)^\top \mathbf{A}(x - x^*) + c_0, \nabla f(x) = \mathbf{A}(x - x^*).$$

Hence, $x_{k+1} - x^* = x_k - x^* - (\mathbf{B}_k + \frac{1}{r_k}\mathbf{I}_d)^{-1}\mathbf{A}(x_k - x^*)$, and take norm on both sides we have

$$\|x_{k+1} - x^*\|_\mathbf{A} \leq \left\|\mathbf{I}_d - \left(\mathbf{B}_k + \frac{1}{r_k}\mathbf{I}_d\right)^{-1}\mathbf{A}\right\|_\mathbf{A} \|x_k - x^*\|_\mathbf{A}$$

$$= \left\|\mathbf{I}_d - \mathbf{A}^{\frac{1}{2}}\left(\mathbf{B}_k + \frac{1}{r_k}\mathbf{I}_d\right)^{-1}\mathbf{A}^{\frac{1}{2}}\right\|_2 \|x_k - x^*\|_\mathbf{A}.$$

Since $\frac{1}{r_k} \geq \|\mathbf{A} - \mathbf{B}_k\|_2$, we have $\mathbf{B}_k + \frac{1}{r_k}\mathbf{I}_d \succeq \mathbf{A}$. As a result, $\mathbf{A}^{\frac{1}{2}}(\mathbf{B}_k + \frac{1}{r_k}\mathbf{I}_d)^{-1}\mathbf{A}^{\frac{1}{2}} \preceq \mathbf{I}_d$. This indicates that

$$\left\|\mathbf{I}_d - \mathbf{A}^{\frac{1}{2}}(\mathbf{B}_k + \frac{1}{r_k}\mathbf{I}_d)^{-1}\mathbf{A}^{\frac{1}{2}}\right\|_2 = \lambda_{\max}\left(\mathbf{I}_d - \mathbf{A}^{\frac{1}{2}}(\mathbf{B}_k + \frac{1}{r_k}\mathbf{I}_d)^{-1}\mathbf{A}^{\frac{1}{2}}\right)$$

$$= 1 - \lambda_{\min}\left(\mathbf{A}^{\frac{1}{2}}(\mathbf{B}_k + \frac{1}{r_k}\mathbf{I}_d)^{-1}\mathbf{A}^{\frac{1}{2}}\right)$$

$$= 1 - \lambda_{\max}^{-1}\left(\mathbf{A}^{-\frac{1}{2}}(\mathbf{B}_k + \frac{1}{r_k}\mathbf{I}_d)\mathbf{A}^{-\frac{1}{2}}\right)$$

$$\leq 1 - \lambda_{\max}^{-1}\left(\mathbf{A}^{-\frac{1}{2}}(\mathbf{A} + \frac{1}{r_k}\mathbf{I}_d)\mathbf{A}^{-\frac{1}{2}}\right)$$

$$= \frac{1}{1 + \mu r_k}.$$

The inequality above follows from the fact that $\mathbf{B}_k \preceq \mathbf{A}$. Note that $f(x) - f^* = \frac{1}{2}\|x - x^*\|_\mathbf{A}^2$, then the proof is complete for $\epsilon_k = 0$. For any $\epsilon_k > 0$, this iteration can be seen as one step for quadratic $f_{\epsilon_k}$ with its Hessian $\mathbf{A} + \epsilon_k\mathbf{I}_d$ and Hessian approximation $\mathbf{B}_k + \epsilon_k\mathbf{I}_d$. It can be proved similarly as above. $\qquad \square$

**Theorem 5.** *Suppose that the update in Algorithm 2 has an initial approximation matrix $\mathbf{B}_0$ such that $0 \preceq \mathbf{B}_0 \preceq \mathbf{A}$ and $\text{tr}(\mathbf{A} - \mathbf{B}_0) \leq T_0$. Then for every $\frac{1}{4T_0} > \varepsilon > 0$, there exists absolute constants $C > 0$, $k_0 = \tilde{\mathcal{O}}(T_0^{-\frac{1}{2}}\varepsilon^{-\frac{1}{2}})$, such that if we set*

$$\epsilon_k = T_0\varepsilon, \quad \frac{1}{T_0 r_k} = \begin{cases} 1 & k \leq 8C\left(1 + \ln \frac{C}{p}\right), \\ \frac{8C(\ln k)^3}{k} & k > 8C\left(1 + \ln \frac{C}{p}\right), \end{cases}$$

*with probability at least $1 - p$, for all $k \geq k_0$, we have*

$$f_{T_0\varepsilon}(x_{k+1}) - f_{T_0\varepsilon}^* \leq (1 - \sqrt{T_0\varepsilon})^{2(k-k_0)}(f_{T_0\varepsilon}(x_0) - f_{T_0\varepsilon}^*). \tag{15}$$

*And for $k = 2k_0$, we also have*

$$f(x_{2k_0}) - f(x^*) \leq \varepsilon T_0 D^2. \tag{16}$$

*Proof of Theorem 5.* First suppose that $T_0 = 1$. From Theorem 1 we know that there exists a constant $C \geq 3$ such that for every $r > 0$, $\left\| \mathbf{A} - \mathbf{B}_{\lfloor Cr(\ln r)^3 + C \ln \frac{k^2}{p} \rfloor} \right\|_2 \leq \frac{1}{r}$ with probability at most $1 - \frac{p}{k^2}$. A straightforward calculation shows that $k > C \ln \frac{k^2}{p}$ when $k \geq 8C \left( 1 + \ln \frac{C}{p} \right)$. Hence, if $k \geq 8C \left( 1 + \ln \frac{C}{p} \right)$, then there exists a unique $r > 1$ such that $C \left( r (\ln r)^3 + \ln \frac{k^2}{p} \right) = k$. From Theorem 1 we have with probability at least $1 - \frac{p}{k^2}$, $\|\mathbf{A} - \mathbf{B}_k\|_2 \leq \frac{1}{r}$. Now we evaluate $r$ in the form of $k$. Since $r \leq k$, we have $r(\ln k)^3 + \ln \frac{k^2}{p} \geq \frac{k}{C}$. This means that

$$r \geq \frac{\frac{k}{C} - 2 \ln k - \ln \frac{1}{p}}{(\ln k)^3}.$$

By $k \geq 8C \left( 1 + \ln \frac{C}{p} \right)$ and performing a basic calculation we can see that

$$\frac{k}{C} - 2 \ln k - \ln \frac{1}{p} \geq \frac{k}{8C}.$$

Therefore, with probability at least $1 - \sum_{k \geq 2}^{k_0} \frac{p}{k^2} \geq 1 - p$, we have for all $k \geq 8C \left( 1 + \ln \frac{C}{p} \right)$, $\|\mathbf{A} - \mathbf{B}_k\|_2 \leq \frac{8C(\ln k)^3}{k}$, which implies $r_k \leq \frac{1}{\|\mathbf{A} - \mathbf{B}_k\|_2}$. Then Lemma 3 tells us that

$$\|x_{k+1} - x_\varepsilon^*\|_{\mathbf{A} + \varepsilon \mathbf{I}_d} \leq \frac{1}{1 + \varepsilon r_k} \|x_k - x_\varepsilon^*\|_{\mathbf{A} + \varepsilon \mathbf{I}_d}, \quad \forall k \geq 8C \left( 1 + \ln \frac{C}{p} \right). \tag{17}$$

Note that $\operatorname{tr}(\mathbf{A} - \mathbf{B}_0) \leq 1$, so $\mathbf{A} - \mathbf{B}_0 \preceq \mathbf{I}_d$, so for $k$ such that $r_k = 1$, we have $\mathbf{B}_k + (\frac{1}{r_k} + \varepsilon)\mathbf{I}_d \succeq \mathbf{A} + \varepsilon \mathbf{I}_d$ and $r_k \leq \frac{1}{\|\mathbf{A} - \mathbf{B}_k\|_2}$. Using Lemma 3 again we have

$$\|x_{k+1} - x_\varepsilon^*\|_{\mathbf{A} + \varepsilon \mathbf{I}_d} \leq \|x_k - x_\varepsilon^*\|_{\mathbf{A} + \varepsilon \mathbf{I}_d}, \quad \forall k \leq 8C \left( 1 + \ln \frac{C}{p} \right). \tag{18}$$

Combining (17), (18), and $f_\varepsilon(x_k) - f_\varepsilon^* = \frac{1}{2} \|x_k - x_\varepsilon^*\|_{\mathbf{A} + \varepsilon \mathbf{I}_d}^2$, recursively, we have

$$f_\varepsilon(x_{k+1}) - f_\varepsilon^* \leq \prod_{t = t_0}^{k} (1 + r_t \varepsilon)^{-2} (f_\varepsilon(x_0) - f^*), \quad t_0 = \lceil 8C \left( 1 + \ln \frac{C}{p} \right) \rceil. \tag{19}$$

Note that if $t = \tilde{\Omega}(\varepsilon^{-\frac{1}{2}})$, $r_t = \Omega(\varepsilon^{-\frac{1}{2}})$. As a result, (19) leads to (15) directly.

For general $T_0$, consider the function $g(x) = f(x)/T_0$. Then Algorithm 2 is equivalent to

$$x_{k+1} = x_k - \left( T_0 \mathbf{B}_k + T_0 \left( \frac{1}{T_0 r_k} + \varepsilon \right) \mathbf{I}_d \right)^{-1} (T_0 \nabla g + T_0 \varepsilon (x_k - x_0)).$$

Hence, using the above result to $g$ and notice that $g_\varepsilon = f_{T_0 \varepsilon}/T_0$, we know that (17), (18) still hold. Thereafter, we have

$$f_{T_0 \varepsilon}(x_{k+1}) - f_{T_0 \varepsilon}^* \leq \prod_{t = t_0}^{k} (1 + T_0 r_t \varepsilon)^{-2} (f_{T_0 \varepsilon}(x_0) - f^*). \tag{20}$$

If $t = \tilde{\Omega}(T_0^{-\frac{1}{2}} \varepsilon^{-\frac{1}{2}})$, then $T_0 r_t \varepsilon = \Omega(T_0^{\frac{1}{2}} \varepsilon^{\frac{1}{2}})$. As a result, (20) leads to (15) directly.

Choose $k_0 = \tilde{\Theta}(T_0^{-\frac{1}{2}} \varepsilon^{-\frac{1}{2}})$ such that $k_0 \geq \frac{1}{2\sqrt{T_0 \varepsilon}} \ln \frac{f_{T_0 \varepsilon}(x_0) - f_{T_0 \varepsilon}^*}{\varepsilon T D^2}$, then (15) implies that $f_{T_0 \varepsilon}(x_{2k_0}) - f_{T_0 \varepsilon}^* \leq \frac{\varepsilon T_0 D^2}{2}$. Hence,

$$f(x_{2k_0}) - f^* \leq f_{T_0 \varepsilon}(x_{2k_0}) - f_{T_0 \varepsilon}^* + \frac{\varepsilon T_0 \|x^* - x_0\|^2}{2} \leq \varepsilon T_0 D^2.$$

Then the proof is complete, showing that the convergence rate is $\tilde{O}(T_0 D^2/k^2)$. $\qquad \square$

---

**Algorithm 3** Phase 1 of Algorithm 2

---

1: **Requires:** Initial point $x_0 \in \mathbb{R}^d$, stepsize $\frac{1}{L}$, $l_2$ regularizer $\epsilon_0 > 0$.
2: **for** $k = 0, 1, 2 \ldots S_1 - 1$ **do**
3: $\quad$ $x_{k+1} = x_k - \frac{1}{L+\epsilon_0}(\nabla f(x_k) + \epsilon_k(x_k - x_0))$
4: **end for**

---

From Lemma 3, we can also derive a superlinear convergence rate for $\mu$ strong convex quadratic functions as stated in the following corollary.

**Corollary 2.** *Suppose that the choice of parameters in Algorithm 2 is the same as in Theorem 5 except for $\epsilon_k$. If $\mathbf{A} \succeq \mu \mathbf{I}_d$ and we choose $\epsilon_k = 0$, then for all $k \geq t_0 = \lceil 8C \left(1 + \ln \frac{C}{p}\right) \rceil$, we have*

$$f(x_{k+1}) - f^* \leq \prod_{t=t_0}^{k} \left(1 + \frac{k\mu}{8CT_0(\ln k)^3}\right)^{-2} (f(x_0) - f^*) \tag{21}$$

*with probability at least $1 - p$.*

### C.2 PROOF OF THEOREM 2

It may be confusing if we directly state how to choose the parameters. Therefore, We will first break down the process of Algorithm 2 into 3 phases, add requirement on the parameters in each phase step by step, give convergence analysis respectively and at last combine the result in different phases.

In Algorithm 2, we require $\{n_k\}$ to satisfy either $n_k = n_{k-1}$ or $n_k = k$. This induces another sub-sequence $\{S_t\} \subset [N]$, where $S_t$ is the $t$-th integer such that $n_{S_t} = S_t$. Hence, for $k \in [S_t, S_{t+1})$, we have $n_k = S_t$.

Divide the algorithm into 3 phases:

- **Phase 1:** (Gradient descent) $k \in [0, S_1)$;
- **Phase 2:** (Hessian approximation) $k \in [S_1, S_2)$;
- **Phase 3:** (Quasi-Newton iteration) $k \in [S_2, N]$.

The first and second phases are for preprocessing. They aim to give a rough estimation for the minimizer of $f_{\varepsilon_0}(x)$ and the Hessian $\nabla^2 f(x^*_{\varepsilon_0})$, which only cost a constant number of iterations. The last phase is the key step, exhibiting $\tilde{\mathcal{O}}(1/k^2)$ convergence rate under proper choice of parameters and preprocessing.

#### C.2.1 ANALYSIS FOR PHASE 1

**Parameter requirement for phase 1:** For $k \in [0, S_1)$, set $n_k = \gamma_k = d_k = 0$, $\epsilon_k = \varepsilon_0$, $r_k = \frac{1}{L}$.

Algorithm 3 shows the first phase of Algorithm 2 under the above requirement. It is actually minimizing a $\varepsilon_0$ strong convex function $f_{\varepsilon_0}(x) = f(x) + \frac{\varepsilon_0}{2}\|x - x_0\|_2^2$. As a consequence, it exhibits a linear convergence rate.

**Proposition 1.** *Under Assumption 1, 2, in Algorithm 3 we have*

$$\|x_{S_1} - x^*_{\varepsilon_0}\|_2 \leq 2e^{-\frac{\varepsilon_0 S_1}{L+2\varepsilon_0}} D. \tag{22}$$

*Proof.* Let $g(x) = f(x) + \frac{\varepsilon_0}{2}\|x - x_0\|_2^2$. Then phase one is equivalent to iterating as

$$x_{k+1} = x_k - \frac{1}{L_g}\nabla g(x_k),$$

where $L_g = L + \varepsilon_0 \geq \sup \|\nabla^2 g(x)\|_2$. It is well-known that for a $m$ strong convex function which is $L_1$ gradient Lipschitz, if the stepsize $\alpha \in (0, \frac{2}{m+L_1})$, then we have

$$\|x_k - x^*\|_2^2 \leq \left(1 - \alpha \frac{2mL_1}{m+L_1}\right)^k \|x_0 - x^*\|_2^2.$$

---

**Algorithm 4** Phase 2 of Algorithm 2

---

1: **Requires:** Initial point $x_{S_1} \in \mathbb{R}^d$, matrix $\mathbf{B}_{S_1} \in \mathbb{R}^{d \times d}$, distribution $\mathcal{D}$.
2: **for** $k = S_1, S_1 + 1, \dots S_2 - 1$ **do**
3:     $x_{k+1} = x_k$
4:     Sample a vector $\mathbf{s}_k \sim \mathcal{D}$ which satisfies (4)
5:     Compute $\mathbf{B}_{k+1} = \mathbf{SR1}(\nabla^2 f(x_k), \mathbf{B}_k, \mathbf{s}_k)$
6: **end for**

---

In our case, $m = \varepsilon_0, L_1 = L_g, \alpha = \frac{1}{L_g}$. Hence,

$$\|x_k - x^*\|_2 \leq \left(1 - \frac{2\varepsilon_0}{\varepsilon_0 + L_g}\right)^{k/2} \|x_0 - x^*\|_2 \leq e^{-\frac{k\varepsilon_0}{L + 2\varepsilon_0}} D.$$

Then the proof is complete. $\qquad\square$

Proposition 1 will help to satisfy the condition 2 in Theorem 6.

### C.2.2 ANALYSIS FOR PHASE 2

**Parameter requirement for phase 2:** For $k \in [S_1, S_2)$, set $n_k = S_1, \epsilon_k = r_k = 0, \gamma_k = d_k = 1$.

Algorithm 4 shows the second phase of Algorithm 2 under the above requirement. It is simply doing Hessian approximation to $\nabla^2 f(x_{S_2})$. Applying Theorem 1 we could evaluate the Hessian approximation quality, which will help to verify the condition 2 in Theorem 6.

**Proposition 2.** *Under Assumption 2, in Algorithm 4, if $0 \preceq \mathbf{B}_{S_1} \preceq \nabla^2 f(x_{S_1})$, then there exists an absolute constant $C$, for every $r > 3, 1 > p > 0$ such that $S_2 - S_1 \geq C\left(r(\ln r)^3 + \ln \frac{1}{p}\right)$, with probability at least $1 - p$, we have*

$$\|\mathbf{B}_{S_2} - \nabla^2 f(x_{S_2})\|_2 \leq \frac{d_{\mathrm{eff}} L}{r} \tag{23}$$

*Proof.* Let $\mathbf{A} = \nabla^2 f(x_{S_1})$ and $\mathbf{B}_0$ refers to $\mathbf{B}_{S_1}$ in Theorem 1 and apply this theorem we directly finish the result. $\qquad\square$

### C.2.3 ANALYSIS FOR PHASE 3

Let $N_t = S_{t+1} - S_t$. We begin to consider the last phase, the most important one, in which $\epsilon_k$ is relatively small ($\epsilon_k = \mathcal{O}(1/M^2 D^2)$). We come up with the first requirement as follows:

**Parameter requirement for phase 3:** For all $t \geq 2$ and $k \in [S_t, S_{t+1})$, there exist $\eta_t, \varepsilon_t, \alpha_t, \beta_t$ such that $\frac{1}{r_k} = L\left(\frac{1}{\eta_t \alpha_t} + \frac{\varepsilon_t(1-\alpha_t)}{\alpha_t}\right), \epsilon_k = L\varepsilon_t$, and $\gamma_k = \alpha_t^{-1}, d_k = 1$ for $k \in [S_t, S_{t+1} - 1)$, $\gamma_k = \alpha_t^{-1}, d_k = (1 - \beta_t)^2$ for $k = S_{t+1} - 1$. The specific values of $\eta_t, \varepsilon_t, \alpha_t, \beta_t$ should satisfy conditions in Theorem 6.

From the above requirement, it can be checked that phase 3 are in the form of Algorithm 5. Note that we write in the form of double loop. However, this is just for the convenience of proof. Our proposed Algorithm 2 **does not need it**. The following theorem provides suitable conditions for parameters in phase 3 and its convergence result.

**Theorem 6.** *Under Assumption 1, 2, 3, suppose that in Algorithm 5, the following conditions are satisfied:*

*1. For the initial point and Hessian approximation, we have*

$$\varepsilon_2 \leq \frac{1}{121 M^2 D^2 L}, \quad \mathbf{B}_{S_2} \preceq \nabla^2 f(x_{S_2});$$

$$\left\|x_{S_2} - x_{L\varepsilon_2}^*\right\|_{x_{L\varepsilon_2}^*}^{L\varepsilon_2} \leq \frac{D\sqrt{L\varepsilon_2}}{33\sqrt{2}}, \quad \left\|\mathbf{B}_{S_2} - \nabla^2 f(x_{S_2})\right\|_2 \leq 2L\min\{MD\sqrt{L}, \frac{1}{MD\sqrt{L}}\}\sqrt{\varepsilon_2}.$$

---

**Algorithm 5** Phase 3 of Algorithm 2

---

1: **Requires:**
    Initial point $x_{S_2} \in \mathbb{R}^d$, $\{\varepsilon_t\} > 0$, initial matrix $\mathbf{B}_{S_2} \in \mathbb{R}^{d \times d}$, correction parameters $\{\beta_t\}$
    Distribution $\mathcal{D}$, stepsize $\{\eta_t\}$, factor $\{\alpha_t\}$.
2: **for** $t = 2, 3, \dots$ **do**
3:    **for** $k = 0, 1, \dots, N_t - 1$ **do**
4:       $x_{k+1+S_t} = x_{k+S_t} - \alpha_t(\mathbf{B}_{k+S_t} + (\frac{L}{\eta_t} + L\varepsilon_t)\mathbf{I}_d)^{-1}\nabla f_{L\varepsilon_t}(x_{k+S_t})$
5:       Sample $\mathbf{s}_{k+S_t} \sim \mathcal{D}$ which satisfies (4)
6:       Update: $\mathbf{B}_{k+S_t+1} = \mathbf{SR1}(\nabla^2 f(x_{S_t}), \mathbf{B}_{k+S_t}, \mathbf{s}_{k+S_t})$
7:    **end for**
8:    Correct Hessian approximation: $\mathbf{B}_{S_{t+1}} = (1 - \beta_t)^2 \mathbf{B}_{S_{t+1}}$
9: **end for**

---

2. *Denote* $c^* = \frac{1}{4096}$, *for parameters, we have for all* $t \geq 2$,

$$\varepsilon_t = \left(\frac{1}{1 + c^*}\right)^{t-2} \varepsilon_2, \quad \alpha_t = \left(1 - \frac{3MD\sqrt{2L\varepsilon_t}}{32}\right)^2;$$

$$\beta_t = \frac{MD\sqrt{L\varepsilon_t}}{4}, \quad \frac{3\sqrt{\varepsilon_t}}{2MD\sqrt{L}} \leq \frac{1}{\eta_t} \leq \frac{2\sqrt{2\varepsilon_t}}{MD\sqrt{L}}.$$

3. *When proceeding the inner loop as stated in Algorithm 6, we guarantee that*

$$\left\|x_{S_t} - x^*_{L\varepsilon_t}\right\|^{L\varepsilon_t}_{x^*_{L\varepsilon_t}} \leq \frac{21D}{320}\sqrt{\varepsilon_t}, \quad \left\|\mathbf{B}_{S_t} - \nabla^2 f(x_{S_{t-1}})\right\|_2 \leq \frac{19L}{40}\min\{MD\sqrt{L}, \frac{1}{MD\sqrt{L}}\}\sqrt{\varepsilon_t}.$$

*Then for every* $\varepsilon > 0$, *to get a solution* $z$ *such that* $f(z) - f^* \leq \varepsilon$, *with high probability, we only need at most* $\tilde{\mathcal{O}}(d_{\text{eff}} LMD^2 \varepsilon^{-\frac{1}{2}})$ *iterations in the 3rd phase of Algorithm 2.*

*Proof of Theorem 6.* For simplicity, we first consider the function $f$ such that $L = 1$. Then for general $f$, the result can be derived from $f/L$ by scaling.

For notation convenience, denote $x_t$ as $x_{S_t}$ and $\tilde{\mathbf{G}}_t = \mathbf{B}_{S_t}$, $\mathbf{H}_t = (1 - \beta_{t-1})^2 \mathbf{B}_{S_t}$. We aim to give a uniform bound to the error measure in the form of $\varepsilon_t$ in each sub-problem solving process. The constants $R_i, w$ that appear in the proof can be seen in Lemma 11. According to Lemma 11, if for every $t \geq 2$, all conditions in this lemma hold, then we can prove the convergence by induction. From the description of Lemma 11, we only need to prove that:

1. For $t = 2$, (45),(46),(47) and $\mathbf{H_2} \preceq \nabla^2 f(x_{S_2})$ hold.

2. For every $t \geq 3$, (48) and $\left\|\tilde{\mathbf{G}}_t - \nabla^2 f(x_{t-1})\right\|_2 \leq R_4\sqrt{\varepsilon_t}$ hold.

The above two statements can be directly verified from our conditions in this theorem. Next, we give an upper bound to the number of iterations in each inner loop. Take $q = \frac{1}{16\sqrt{2}}$ in Lemma 10 and we can check that our choice of parameters satisfies its conditions by using Lemma 12. Therefore, for every $k \in [S_t, S_{t+1})$, we have

$$\left\|x_{k+1} - x^*_{\varepsilon_t}\right\|^{\varepsilon_t}_{x^*_{\varepsilon_t}} \leq \left(1 - \frac{qMD}{2}\sqrt{\varepsilon_t}\right)\left\|x_k - x^*_{\varepsilon_t}\right\|^{\varepsilon_t}_{x^*_{\varepsilon_t}}, \quad x_0 = x_{t-1}.$$

Note (52),(53) in the proof of Lemma 11 also give a bound to $\left\|x_{t-1} - x^*_{\varepsilon_t}\right\|^{\varepsilon_t}_{x_{\varepsilon_t}}$, that is

$$\left\|x_{t-1} - x^*_{\varepsilon_t}\right\|^{\varepsilon_t}_{x_{\varepsilon_t}} \leq \frac{1}{1 - MR_1\sqrt{\varepsilon_t}}\left(w\sqrt{\varepsilon_t} + \frac{R_1\sqrt{\varepsilon_{t-1}}}{1 - Mw\sqrt{\varepsilon_t}}\right) \leq \frac{D}{4}\sqrt{\varepsilon_t}.$$

Hence, for $k - S_t \geq \frac{2\ln 4}{qMD\sqrt{\varepsilon_t}}$, we have

$$\left\|x_k - x^*_{\varepsilon_t}\right\|^{\varepsilon_t}_{x^*_{\varepsilon_t}} \leq \left(1 - \frac{qMD}{2}\sqrt{\varepsilon_t}\right)^k \frac{D}{4}\sqrt{\varepsilon_t} \leq \frac{21D}{320}\sqrt{\varepsilon_t}.$$

Table 1: Parameter settings for Algorithm 2 in different phases.

| Parameter | Phase 1: $k \in [0, S_1)$ | Phase 2: $k \in [S_1, S_2)$ | Phase 3: $k \in [S_t, S_{t+1}), t \geq 2$ |
|---|---|---|---|
| $n_k$ | 0 | $S_1$ | $S_t$ |
| $\epsilon_k$ | $(11MD)^{-2}$ | 0 | $L\varepsilon_t$ |
| $r_k$ | $\frac{1}{L}$ | 0 | $\frac{\eta_t \alpha_t}{L(1+\eta_t \varepsilon_t (1-\alpha_t))}$ |
| $\gamma_k$ | 0 | 1 | $\alpha_t^{-1}$ |
| $d_k$ | 0 | 1 | 1 for $k < S_{t+1} - 1$, $(1-\beta_t)^2$ for $k = S_{t+1} - 1$ |

Then (48) holds. By Theorem 1, with probability at least $1 - p$, we have

$$\left\| \mathbf{G}_k - \nabla^2 f(x_{t-1}) \right\|_2 \leq \frac{19}{40MD} \sqrt{\varepsilon_t}, \quad k - S_t \geq C d_{\text{eff}} \left( \frac{40MD}{19\sqrt{\varepsilon_t}} \left( \ln \frac{40CMD}{19\sqrt{\varepsilon_t}} \right)^3 + \ln \frac{1}{p} \right).$$

Thus, we only need to iterate $N_t \sim \tilde{\Theta} \left( \frac{MD + M^{-1}D^{-1}}{\sqrt{\varepsilon_t}} \right)$ times in each inner loop. Now we choose $m = \lceil 4096 \log_2 \frac{1}{M^2 \varepsilon} \rceil$, then we have

$$\varepsilon_m \leq \rho^m \varepsilon_0 \leq 2M^2 \varepsilon \cdot \frac{1}{2M^2 \|x^* - x_0\|_2^2} \leq \frac{\varepsilon}{2 \|x^* - x_0\|_2^2}.$$

Similar to the proof in Lemma 9, we have

$$f(x_m) - f(x_{\varepsilon_m}^*) \leq \frac{1}{(1 - M \|x_m - x_{\varepsilon_m}^*\|_{x_{\varepsilon_m}^*}^{\varepsilon_m})^2} \left( \|x_m - x_{\varepsilon_m}^*\|_{x_{\varepsilon_m}^*}^{\varepsilon_m} \right)^2$$

$$\leq \frac{10}{9} \cdot \frac{\|x_0 - x^*\|_2^2}{1000} \varepsilon_m \leq \frac{\varepsilon}{2}.$$

This implies

$$f(x_m) - f(x^*) \leq f(x_m) - f(x_{\varepsilon_m}^*) + f(x_{\varepsilon_m}^*) - f(x^*) \leq \frac{\varepsilon}{2} + \frac{\varepsilon_m}{2} \|x^*\|_2^2 \leq \varepsilon.$$

Without the loss of generality, we can set $MD \geq 1$, otherwise we can choose our $M = D^{-1}$. Then to obtain the solution $x_m$, we only need at most $m \cdot \tilde{\mathcal{O}} \left( \frac{d_{\text{eff}} MD}{\sqrt{\varepsilon_m}} \right) = \tilde{\mathcal{O}} \left( \frac{d_{\text{eff}} MD^2}{\sqrt{\varepsilon}} \right)$ iterations in total. Then we finish our proof for $L = 1$.

For general $L$, consider Algorithm 5 as a process to minimize $g = f/L$. For conditions in Theorem 6, note that the quantity $MD\sqrt{L}$ remain the same after scaling, so the condition on $\eta_t, \varepsilon_t, \alpha_t, \beta_t$ remains the same as well. Note also that $\nabla^2 g = \nabla^2 f/L$ and $\mathbf{B}_k' = \mathbf{B}_k/L$ where $\mathbf{B}_k$ is the Hessian approximation for $\nabla^2 f$, $\mathbf{B}_k'$ is the Hessian approximation for $\nabla^2 g$ using initial $\mathbf{B}_0' = \mathbf{B}_0/L$, so the condition for Hessian approximation is consistent after scaling. Note also that $\sqrt{z^\top (\nabla^2 f + Lc)z} = \sqrt{L}\sqrt{z^\top (\nabla^2 g + c)z}$ and the $x_{Lc}^*$ of $f$ is the same as the $x_c^*$ of $g$, so the conditions related to $\|x - y\|_z^c$ is consistent after scaling.

Therefore, by our requirement on parameters, phase 3 equivalently performs the optimization on $g$ using parameters consistent with the case when $L = 1$. Hence, to find $x_m$ such that $g(x_m) - g^* \leq \varepsilon$, we need at most $\tilde{\mathcal{O}} \left( \frac{d_{\text{eff}} M\sqrt{L}D^2}{\sqrt{\varepsilon}} \right)$ iterations (the self-concordance coefficient for $g$ is $M\sqrt{L}$) . This means to find $x_m$ such that $f(x_m) - f^* \leq L\varepsilon$, we need at most $\tilde{\mathcal{O}} \left( \frac{d_{\text{eff}} MLD^2}{\sqrt{L\varepsilon}} \right)$ iterations. Then the proof is finished. $\qquad \square$

### C.2.4 FINAL PROOF

First, we summarize the previous results and present the explicit form of parameters in Table 1, 2.

Combining the results in 3 phases, we can easily give a proof for Theorem 2.

Table 2: One example for the value of $\varepsilon_t, \alpha_t, \beta_t, \eta_t, S_t, t \geq 2$.

| Parameter | Values |
|---|---|
| $\varepsilon_t$ | $\left(\frac{4096}{4097}\right)^{t-2} \frac{1}{121M^2D^2L}$ |
| $\alpha_t$ | $\left(1 - \frac{3MD\sqrt{2L\varepsilon_t}}{32}\right)^2$ |
| $\beta_t$ | $\frac{MD\sqrt{L\varepsilon_t}}{4}$ |
| $\eta_t$ | $\frac{3\sqrt{\varepsilon_t}}{2MD\sqrt{L}} \leq \frac{1}{\eta_t} \leq \frac{2\sqrt{2\varepsilon_t}}{MD\sqrt{L}}$ |
| $S_t$ | $S_1 = \frac{L+2\varepsilon_0}{\varepsilon_0} \ln\left(\frac{66\sqrt{2}D}{\varepsilon_0^{1/2}}\right)$ |
| | $S_2 = S_1 + C\left(r(\ln r)^3 + \ln\frac{1}{p}\right)$, where $r = 22d_{\text{eff}}\max\{1, M^2D^2L\}$ |
| | $S_{t+1} - S_t = Cd_{\text{eff}}\left(\frac{40MD\sqrt{L}}{19\sqrt{\varepsilon_t}}\left(\ln\frac{40CMD\sqrt{L}}{19\sqrt{\varepsilon_t}}\right)^3 + \ln\frac{1}{p}\right)$ |

*Proof of Theorem 2.* Suppose that the parameters satisfy the requirement in each phase, fix a failure probability $p$, set $\varepsilon_0 = \frac{1}{121M^2D^2}$, $S_1 = \frac{L+2\varepsilon_0}{\varepsilon_0}\ln\frac{66\sqrt{2}D}{\varepsilon_0^{0.5}} = \tilde{\mathcal{O}}(M^2D^2L)$, $S_2 = S_1 + C\left(r(\ln r)^3 + \ln\frac{1}{p}\right)$, where $r = 22d_{\text{eff}}\max\{1, M^2D^2L\} = \tilde{\mathcal{O}}(d_{\text{eff}}M^2D^2L)$. Set

$$N_t = S_{t+1} - S_t = Cd_{\text{eff}}\left(\frac{40MD\sqrt{L}}{19\sqrt{\varepsilon_t}}\left(\ln\frac{40CMD\sqrt{L}}{19\sqrt{\varepsilon_t}}\right)^3 + \ln\frac{1}{p}\right), \quad t \geq 2.$$

Then by Proposition 1, for $\varepsilon \geq \frac{1}{M^2}$, we need at most $\tilde{\mathcal{O}}(M^2D^2L)$ iterations. By Proposition 2 and Theorem 6, for $\varepsilon \leq \frac{1}{M^2}$, we need at most $\tilde{\mathcal{O}}(d_{\text{eff}}M^2D^2L + d_{\text{eff}}LMD^2\varepsilon^{-\frac{1}{2}}) = \tilde{\mathcal{O}}(d_{\text{eff}}LMD^2\varepsilon^{-\frac{1}{2}})$ iterations to let $f - f^* \leq \varepsilon$ with probability at least $1 - \tilde{\mathcal{O}}(d_{\text{eff}}LMD^2\varepsilon^{-\frac{1}{2}})p$. Combining two cases, we finish the proof. $\square$

### C.3 Analysis of Computational Complexity

We now give a computational complexity analysis for Algorithm 2 in the parameter scheme described in Table 1, 2. We first demonstrate the computational cost for a single iteration. This efficiency stems from the low-rank nature of the Hessian approximation $\mathbf{B}_k$. According to Sherman-Morrison-Woodbury formula:

$$\left(\mathbf{A} + \mathbf{U}\mathbf{C}\mathbf{V}^\top\right)^{-1} = \mathbf{A}^{-1} - \mathbf{A}^{-1}\mathbf{U}\left(\mathbf{C}^{-1} + \mathbf{V}^\top\mathbf{A}^{-1}\mathbf{U}\right)^{-1}\mathbf{V}^\top\mathbf{A}^{-1}$$

Note that in the scheme, our Hessian approximation, $\mathbf{B}_k$, starts from $\mathbf{B}_{S_1} = 0$. Since each iteration involves only a rank-one update, the rank of $\mathbf{B}_k$ is at most $k$. If we store the update vectors $\mathbf{u}_i$ from each step, then $\mathbf{B}_k = \sum_{i=1}^{k}\mathbf{u}_i\mathbf{u}_i^\top = \mathbf{U}_k\mathbf{U}_k^\top$, where $\mathbf{U}_k = (\mathbf{u}_1, \mathbf{u}_2, \cdots, \mathbf{u}_k)$. For any $a_k > 0$, by setting $\mathbf{A} = a_k\mathbf{I}_d, C = \mathbf{I}_k$, and $\mathbf{V} = \mathbf{U}$ in the SMW formula, we derive:

$$(a_k\mathbf{I}_d + \mathbf{U}_k\mathbf{U}_k^\top)^{-1} = a_k^{-1}\mathbf{I}_d - a_k^{-1}\mathbf{U}_k(\mathbf{I}_k + a_k^{-1}\mathbf{U}_k^\top\mathbf{U}_k)^{-1}\mathbf{U}_k^\top \cdot a_k^{-1}\mathbf{I}_d.$$

Consequently, for any vector $\mathbf{w}$, the term $(a_k\mathbf{I}_d + \mathbf{U}_k\mathbf{U}_k^\top)^{-1}\mathbf{w}$ can be computed as:

$$(a_k\mathbf{I}_d + \mathbf{U}_k\mathbf{U}_k^\top)^{-1}\mathbf{w} = a_k^{-1}\mathbf{w} - a_k^{-1}\mathbf{U}_k(a_k\mathbf{I}_k + \mathbf{U}_k^\top\mathbf{U}_k)^{-1}\mathbf{U}_k^\top\mathbf{w}.$$

Let's break down the computational costs:

- Calculating $\mathbf{w}_1 = \mathbf{U}_k^\top\mathbf{w}$ costs $\mathcal{O}(kd)$.
- Assuming $\mathbf{U}_{k-1}^\top\mathbf{U}_{k-1}$ is already available, then $\mathbf{U}_k^\top\mathbf{U}_k$ can be formed by:

$$\mathbf{U}_k^\top\mathbf{U}_k = \begin{bmatrix} \mathbf{U}_{k-1}^\top\mathbf{U}_{k-1} & \mathbf{U}_{k-1}^\top\mathbf{u}_k \\ \mathbf{u}_k^\top\mathbf{U}_{k-1} & \mathbf{u}_k^\top\mathbf{u}_k \end{bmatrix}.$$

This only requires computing $\mathbf{U}_{k-1}^\top\mathbf{u}_k$ and $\mathbf{u}_k^\top\mathbf{u}_k$, both costing $\mathcal{O}(kd)$.

- Inverting the $k \times k$ matrix $(a_k \mathbf{I}_k + \mathbf{U}_k^\top \mathbf{U}_k)$ costs $\mathcal{O}(k^3)$.
- Calculating $\mathbf{w}_2 = (a_k \mathbf{I}_k + \mathbf{U}_k^\top \mathbf{U}_k)^{-1} \mathbf{w}_1$ costs $\mathcal{O}(k^2)$.
- Calculating $\mathbf{w}_3 = \mathbf{U}_k \mathbf{w}_2$ costs $\mathcal{O}(kd)$.
- Finally, computing $\mathbf{w}_4 = a_k^{-1} \mathbf{w} - \mathbf{w}_3$ (the final result) costs $\mathcal{O}(d)$.

Therefore, the total computational complexity in the $k$-th iteration is $\mathcal{O}(k^3 + kd)$.

However, if for $k \in [S_t, S_{t+1})$, $a_k$ are the same, then for $k \in (S_t, S_{t+1})$, we have

$$
(a_k \mathbf{I}_k + \mathbf{U}_k^\top \mathbf{U}_k)^{-1} = \begin{bmatrix} a_{k-1} \mathbf{I}_{k-1} + \mathbf{U}_{k-1}^\top \mathbf{U}_{k-1} & \mathbf{U}_{k-1}^\top \mathbf{u}_k \\ \mathbf{u}_k^\top \mathbf{U}_{k-1} & a_k + \mathbf{u}_k^\top \mathbf{u}_k \end{bmatrix}^{-1}
$$
$$
= \begin{bmatrix} \mathbf{X}^{-1} + \frac{\mathbf{X}^{-1} \mathbf{b} \mathbf{b}^\top \mathbf{X}^{-1}}{c - \mathbf{b}^\top \mathbf{X}^{-1} \mathbf{b}} & -\frac{\mathbf{X}^{-1} \mathbf{b}}{c - \mathbf{b}^\top \mathbf{X}^{-1} \mathbf{b}} \\ -\frac{\mathbf{b}^\top \mathbf{X}^{-1}}{c - \mathbf{b}^\top \mathbf{X}^{-1} \mathbf{b}} & \frac{1}{c - \mathbf{b}^\top \mathbf{X}^{-1} \mathbf{b}} \end{bmatrix},
$$

where $\mathbf{X} = a_{k-1} \mathbf{I}_{k-1} + \mathbf{U}_{k-1}^\top \mathbf{U}_{k-1}, \mathbf{b} = \mathbf{U}_{k-1}^\top \mathbf{u}_k, c = a_k + \mathbf{u}_k^\top \mathbf{u}_k$. Since $\mathbf{X}^{-1}$ has already been computed in the $(k-1)$ -th iteration, we only need to compute $\mathbf{X}^{-1} \mathbf{b}$, which only costs $\mathcal{O}(k^2)$. Hence, the total computational complexity in the $k$-th iteration is $\mathcal{O}(k^2 + kd)$ except for $k = S_t$. Note that in our choice of $S_t$ (see the proof of Theorem 2 in C.2.4), satisfies $S_t - S_{t-1}$ grows at least linearly, so $S_t \leq k$ only for $t \leq \mathcal{O}(\ln k)$. The total computational cost in the first $k$ iterations is at most

$$
\sum_{l=1}^{k} \mathcal{O}(l^2 + ld) + \mathcal{O}(\ln k) \mathcal{O}(k^3 + kd) = \tilde{\mathcal{O}}(k^3 + k^2 d).
$$

## D    USEFUL LEMMAS IN APPENDIX B

Lemma 7 depicts the decrease of trace during the SR1 update.

**Lemma 7.** *Denote* $\mathbf{R}_k = \mathbf{A} - \mathbf{B}_k$*, then there exist constants* $C_3, C_4 > 0$ *such that:*

$$
\mathbb{P}\left(\operatorname{tr}(\mathbf{R}_{k+1}) \leq \operatorname{tr}(\mathbf{R}_k) - C_3 \frac{\operatorname{tr}(\mathbf{R}_k^2)}{\operatorname{tr}(\mathbf{R}_k)}\right) \geq C_4. \tag{24}
$$

*Proof.* Denote $\mathbf{R}_k = \mathbf{A} - \mathbf{B}_k$, then by (3) we have

$$
\mathbf{R}_{k+1} = \mathbf{R}_k - \frac{\mathbf{R}_k \mathbf{s}_k \mathbf{s}_k^\top \mathbf{R}_k}{\mathbf{s}_k^\top \mathbf{R}_k \mathbf{s}_k}. \tag{25}
$$

Taking the trace on both sides of (25), we have

$$
\operatorname{tr}(\mathbf{R}_{k+1}) = \operatorname{tr}(\mathbf{R}_k) - \frac{\mathbf{s}_k^\top \mathbf{R}_k^2 \mathbf{s}_k}{\mathbf{s}_k^\top \mathbf{R}_k \mathbf{s}_k}. \tag{26}
$$

From (4) we can see that

$$
\mathbb{E}\left[\mathbf{s}_k^\top \mathbf{R}_k \mathbf{s}_k\right] = \operatorname{tr}(\mathbf{R}_k), \quad \mathbb{E}\left[\mathbf{s}_k^\top \mathbf{R}_k^2 \mathbf{s}_k\right] = \operatorname{tr}(\mathbf{R}_k^2) \tag{27}
$$

By Markov inequality we have

$$
\mathbb{P}\left(\mathbf{s}_k^\top \mathbf{R}_k \mathbf{s}_k \leq 8(3 + C_1) \mathbb{E}\left[\mathbf{s}_k^\top \mathbf{R}_k \mathbf{s}_k\right]\right) \geq 1 - \frac{1}{8(3 + C_1)}. \tag{28}
$$

By Lemma 19 and Lemma 16 we have

$$
\mathbb{P}\left(\mathbf{s}_k^\top \mathbf{R}_k^2 \mathbf{s}_k \geq \frac{1}{2} \mathbb{E}\left[\mathbf{s}_k^\top \mathbf{R}_k^2 \mathbf{s}_k\right]\right) \overset{(55)}{\geq} \frac{\mathbb{E}\left[\mathbf{s}_k^\top \mathbf{R}_k^2 \mathbf{s}_k\right]}{4 \mathbb{E}\left[(\mathbf{s}_k^\top \mathbf{R}_k^2 \mathbf{s}_k)^2\right]}
$$
$$
\overset{(59)}{\geq} \frac{\operatorname{tr}(\mathbf{R}_k^2)^2}{4(3 + C_1) \operatorname{tr}(\mathbf{R}_k^2)^2}
$$
$$
\geq \frac{1}{4(3 + C_1)}. \tag{29}
$$

Combining (28) and (29) we have

$$\mathbb{P}\left(\operatorname{tr}(\mathbf{R}_{k+1}) \leq \operatorname{tr}(\mathbf{R}_k) - \frac{\operatorname{tr}(\mathbf{R}_k^2)}{16(3+C_1)\operatorname{tr}(\mathbf{R}_k)}\right) \geq \frac{1}{8(3+C_1)}. \tag{30}$$

$\square$

Lemma 8 is the key lemma to prove Lemma 6, which constructs a rational fraction to compare its roots with the eigenvalues . We can show that the top eigenvalues after one iteration are smaller than these roots respectively with a certain probability and quantify the amount of top eigenvalues' reduction by the difference value between the previous step's eigenvalues and these roots in each iteration.

**Lemma 8.** *Under the condition of Lemma 6, if (10) does not hold, then there exist constants $C_5, C_6 > 0$ such that for all $0 \leq k \leq K$, with probability at least $C_5$, we have*

$$\operatorname{tr}(\mathbf{A} - \mathbf{B}_{k+1})_s \leq \operatorname{tr}(\mathbf{A} - \mathbf{B}_k)_s - \frac{umC_6}{s^2 \ln \frac{s}{u}}(\operatorname{tr}(\mathbf{A} - \mathbf{B}_k)_s)^2, \tag{31}$$

*where $\operatorname{tr}(\mathbf{H})_s$ means the sum of top $s$ eigenvalues of a matrix $\mathbf{H}$.*

*Proof.* For simplicity of notation, denote $d_i = \lambda_i(\mathbf{A} - \mathbf{B}_k), \lambda_i = \lambda_i(\mathbf{A} - \mathbf{B}_{k+1})$. We first assume that $d_i$ are distinct from each other. Let

$$\mathbf{A} - \mathbf{B}_k = \mathbf{U}\mathbf{D}\mathbf{U}^\top$$

be the orthogonal decomposition of symmetric matrix $\mathbf{A} - \mathbf{B}_k$. Denote $\mathbf{U} = (u_{ij})_{d \times d}, \mathbf{s}_k = (s_1, \cdots, s_d)^\top$, $\mathbf{U}\mathbf{s}_k = \mathbf{v}$ and $\mathbf{v} = (v_1, \cdots, v_d)^\top$, assume that $v_i \neq 0$ for all $i$, then we can rewrite (3) as:

$$\mathbf{A} - \mathbf{B}_{k+1} = \mathbf{U}\left(\mathbf{D} - \frac{\mathbf{D}\mathbf{v}\mathbf{v}^\top\mathbf{D}}{\mathbf{v}^\top\mathbf{D}\mathbf{v}}\right)\mathbf{U}^\top.$$

By Lemma 24, we know that $\lambda_i$ are the roots of

$$q(x) = \sum_{i=1}^{d} \frac{d_i^2 v_i^2}{d_i - x} - \sum_{i=1}^{d} d_i v_i^2.$$

Now we begin to progress our proof in the following steps.

1. **Step 1:** First, we find a **special point** in $(d_m, d_{s+1})$. We claim that the set

$$Q \stackrel{\text{def}}{=} \left\{x \in S \stackrel{\text{def}}{=} \left[\frac{2d_m}{u+2}, \frac{3d_m}{u+3}\right] : \sum_{i=m}^{s+1} \frac{d_i^2}{|x - d_i|} \leq \frac{120}{u}\ln\frac{30(1+s)^2}{u}\right\} \neq \varnothing.$$

The proof of this claim is as follows. Let $T = \bigcup_{i=m}^{s+1}\left[d_i - \frac{u}{30}d_i^2, d_i + \frac{u}{30}d_i^2\right]$, then the integral

$$\int_{S\backslash T} \sum_{i=m}^{s+1} \frac{d_i^2}{|x - d_i|}dx \leq 2\sum_{i=m}^{s+1} d_i^2 \int_{\frac{ud_i^2}{30}}^{1} \frac{1}{x}dx$$

$$= 2\sum_{i=m}^{s+1} -d_i^2 \ln\frac{u}{30}d_i^2 = 2\sum_{i=m}^{s+1} d_i^2 \ln\frac{30}{u} + 4\sum_{i=m}^{s+1} d_i^2 \ln\frac{1}{d_i}$$

$$\stackrel{(a)}{\leq} 2(s-m)d_m^2 \ln\frac{30}{u} + 4d_m\sum_{i=m}^{s+1} d_i \ln\frac{1}{d_i}$$

$$\stackrel{(b)}{\leq} 2(s-m)d_m^2 \ln\frac{30}{u} + 4d_m(s-m+1)d_m \ln\frac{1}{d_m}$$

$$\leq 2s\left(\ln\frac{30}{u}s - 2\ln d_m\right)d_m^2. \tag{32}$$

Here $(a)$ holds because $d_i \le d_m$ and $(b)$ holds because $d_i \ln \frac{1}{d_i} \le d_m \ln \frac{1}{d_m}$ by Lemma 13. Meanwhile, we have

$$|S\backslash T| \ge \frac{3d_m}{u+3} - \frac{2d_m}{u+2} - \frac{u}{15}\sum_{i=m}^{s+1} d_i^2 \ge \frac{ud_m}{(u+3)(u+2)} - \frac{ud_m}{15} \ge \frac{ud_m}{60}. \tag{33}$$

Combining (32)(33) we know that there exists $z \in S\backslash T$ such that

$$\sum_{i=m}^{s} \frac{d_i^2}{|z-d_i|} \le \frac{120s}{u}\left(\ln\frac{30}{u}d_m + 2d_m\ln\frac{1}{d_m}\right) \le \frac{120}{u}\left(\ln 30 - \ln u + 2\ln(1+s)\right). \tag{34}$$

The last inequality in (34) can be derived from Lemma 13 and the condition that $d_m \le \frac{1}{s+1} \le \frac{1}{e}$. Then we finish the proof of our claim.

2. **Step 2:** Next, we construct a rational fraction $r(x)$ in order to compare with $q(x)$ as follows. Let $m \le j \le s$ satisfy $d_j < z < d_{j+1}$, denote $M = 8(3 + C_1)$ and define

$$r(x) \stackrel{\text{def}}{=} \sum_{i=1}^{j} \frac{d_i^2 v_i^2}{(d_i - x)} - \left(\sum_{i=j+1}^{d} \frac{d_i^2 v_i^2}{z - d_i} + \frac{8M}{u}\sum_{i=1}^{d} d_i + \sum_{i=m+1}^{j} \frac{d_i^2 v_i^2}{d_i - z}\right). \tag{35}$$

Using Lemma 22, we can know that $r(x) = 0$ has $j$ solutions $\mu_i, 1 \le i \le j$ such that

$$d_1 > \mu_1 > d_2 > \cdots > d_j > \mu_j,$$

and we have the following equation by (63)

$$\sum_{i=1}^{j}(d_i - \mu_i) = \frac{\sum_{i=1}^{j} d_i^2 v_i^2}{M_1}, \quad M_1 = \sum_{i=j+1}^{d} \frac{d_i^2 v_i^2}{z - d_i} + \frac{8M}{u}\sum_{i=1}^{d} d_i + \sum_{i=m+1}^{j} \frac{d_i^2 v_i^2}{d_i - z}. \tag{36}$$

Note that for $i \le m$, $d_i - z \ge d_i - \frac{3}{u+3}d_m \ge \frac{u}{u+3}d_i$, therefore,

$$r(z) \le \sum_{i=1}^{m}\left(1 + \frac{3}{u}\right) d_i v_i^2 + \sum_{i=m+1}^{j} \frac{d_i^2 v_i^2}{d_i - z} - M_1. \tag{37}$$

3. **Step 3:** Now we compare $\lambda_i$ with $\mu_i$. Since $v_i = \sum_{l=1}^{d} u_{il}s_l$, we have $\mathbb{E}\left[v_i^2\right] = 1$ and

$$\mathbb{E}\left[v_i^4\right] = \mathbb{E}\left[\left(\sum_{p=1}^{d} u_{ip}s_p\right)^4\right] \stackrel{(c)}{=} \mathbb{E}\left[\sum_{p=1}^{d} u_{ip}^4 s_p^4 + 6\sum_{1\le p<q\le d} u_{ip}^2 u_{iq}^2 s_p^2 s_q^2\right]$$

$$\le (3 + C_1)\left(\sum_{p=1}^{d} u_{ip}^2\right)^2 = 3 + C_1.$$

The equality $(c)$ holds for the same reason as (60). For each $m \le l \le j$, we define random variables

$$X \stackrel{\text{def}}{=} \sum_{i=1}^{d} d_i v_i^2, Y \stackrel{\text{def}}{=} \sum_{i=1}^{j} d_i^2 v_i^2, Z \stackrel{\text{def}}{=} \sum_{i=j+1}^{d} \frac{d_i^2 v_i^2}{z - d_i} + \sum_{i=m+1}^{j} \frac{d_i^2 v_i^2}{d_i - z}.$$

By Lemma 20 and Markov inequality, we have

$$\mathbb{P}\left(\mathcal{A} \stackrel{\text{def}}{=} \left\{Y \ge \frac{1}{2}\mathbb{E}\left[Y\right], X \le 2M\mathbb{E}\left[X\right], Z \le 2M\mathbb{E}\left[Z\right]\right\}\right) \ge \frac{1}{M}. \tag{38}$$

In the rest part of this step we condition on $\mathcal{A}$, then by (37), we have

$$r(z) \le \left(1 + \frac{3}{u}\right) X + \sum_{i=m+1}^{j} \frac{d_i^2 v_i^2}{d_i - z} - \frac{8M}{u}\mathbb{E}\left[X\right] - \sum_{i=m+1}^{j} \frac{d_i^2 v_i^2}{d_i - z} \le 0.$$

Hence, $\mu_j \geq z$, and

$$q(\mu_l) = \sum_{i=1}^{j} \frac{d_i^2 v_i^2}{(d_i - \mu_l)} - \sum_{i=j+1}^{d} \frac{d_i^2 v_i^2}{\mu_l - d_i} - X$$

$$\overset{(35)}{=} r(\mu_l) + \left( \sum_{i=j+1}^{d} \frac{d_i^2 v_i^2}{z - d_i} + \frac{8M}{u} \sum_{i=1}^{d} d_i + \sum_{i=m+1}^{j} \frac{d_i^2 v_i^2}{d_i - z} \right) - \sum_{i=j+1}^{d} \frac{d_i^2 v_i^2}{\mu_l - d_i} - X$$

$$\overset{\mu_l \geq z}{\geq} r(\mu_l) + \left( \sum_{i=j+1}^{d} \frac{d_i^2 v_i^2}{\mu_l - d_i} + \frac{8M}{u} \sum_{i=1}^{d} d_i \right) - \sum_{i=j+1}^{d} \frac{d_i^2 v_i^2}{\mu_l - d_i} - X$$

$$\geq \frac{8M}{u} \sum_{i=1}^{d} d_i - X \geq \frac{8M}{u} \mathbb{E}[X] - X \geq 0.$$

This implies $\lambda_l \leq \mu_l$. Therefore, $d_l - \lambda_l \geq d_l - \mu_l$. Since $Z \leq 2M\mathbb{E}[Z]$, $Y \geq \frac{1}{2}\mathbb{E}[Y]$, by (36), we can bound $\sum_{i=1}^{j}(d_i - \mu_i)$ as follows

$$M_1 \leq 2M\mathbb{E}[Z] + \frac{8M}{u}\mathbb{E}[X]$$

$$\overset{(34)}{\leq} 2M\left( \frac{120}{u} \left( \frac{1}{30} + \ln 30 - \ln u + 2\ln(1+s) \right) \right) + M\left( \sum_{i=s+2}^{d} \frac{d_i^2}{z - d_i} \right)$$

$$\overset{(d)}{\leq} 2M\left( \frac{120}{u} \left( \frac{1}{30} + \ln 30 - \ln u + 2\ln(1+s) \right) \right) + M\left( 1 + \frac{2}{u} \right)\left( \sum_{i=s+2}^{d} d_i \right)$$

$$\leq 2M\left( \frac{120}{u} \ln \frac{32(1+s)^2}{u} + 1 + \frac{2}{u} \right). \tag{39}$$

The inequality $(d)$ holds due to the fact that

$$z - d_i \geq \frac{2}{u+2} d_m - d_i \geq \frac{2(1+u)}{u+2} d_i - d_i \geq \frac{u}{u+2} d_i.$$

Take $C_6 = \frac{1}{3000M}$, by (39) and some numerical calculation we have $M_1 \leq \frac{1}{2uC_6} \ln \frac{s}{u}$, thus,

$$\sum_{i=1}^{j}(d_i - \lambda_i) \geq \sum_{i=1}^{j}(d_i - \mu_i) \geq \frac{Y}{M_1} \geq \frac{\sum_{i=1}^{j} d_i^2}{2M_1} \geq \frac{uC_6}{\ln \frac{s}{u}} \sum_{i=1}^{j} d_i^2.$$

Note that $m \leq j \leq s$, so we have

$$\sum_{i=1}^{j} d_i^2 \geq \frac{j}{s} \sum_{i=1}^{s} d_i^2 \geq \frac{j}{s^2} \left( \sum_{i=1}^{s} d_i \right)^2 \geq \frac{m}{s^2} \left( \sum_{i=1}^{s} d_i \right)^2.$$

$$\operatorname{tr}(\mathbf{A} - \mathbf{B}_{k+1})_s \leq \operatorname{tr}(\mathbf{A} - \mathbf{B}_k)_s - \frac{umC_6}{s^2 \ln \frac{s}{u}} \left( \operatorname{tr}(\mathbf{A} - \mathbf{B}_k)_s \right)^2. \tag{40}$$

Combining (38) and (40), we choose $C_5 = \frac{1}{M}$ and then we finish the proof.

For the case when $v_i$ may be zero and $d_i$ may not be distinct from each other, by Weyl's inequality (Lemma 21), the spectrum of Hermitian matrices is stable under perturbation. Hence, the conclusion is true for the general case. $\qquad\square$

# E    USEFUL LEMMAS IN APPENDIX C.2

We need several additional lemmas to help proving Theorem 6 as listed below. **All the Lemmas in this section assume** $L = 1$. (Though some do not use it.) Lemma 9 measures the deviation between the exact solutions of the proximate inner loops. Lemma 10 proves a linear convergence rate under proper conditions. Lemma 11 and Lemma 12 further measure the deviation between the approximated solutions of proximate inner loops.

**Lemma 9.** *Suppose that $\varepsilon_t \leq \varepsilon_{t-1} \leq \frac{1}{2M^2\|x^*-x_0\|_2^2}$, then*

$$\left\|x_{\varepsilon_t}^* - x^*\right\|_{x^*} \leq \sqrt{2\varepsilon_t}\|x^* - x_0\|_2, \tag{41}$$

$$\left\|x_{\varepsilon_{t-1}}^* - x_{\varepsilon_t}^*\right\|_{x_{\varepsilon_t}^*}^{\varepsilon_t^*} \leq \sqrt{2(\varepsilon_{t-1} - \varepsilon_t)}\|x^* - x_0\|_2. \tag{42}$$

*Proof of Lemma 9.* Denote $r = \left\|x_{\varepsilon_t}^* - x^*\right\|_{x^*}$, then by self-concordancy we have $\nabla^2 f\left(x^* + t\left(x_{\varepsilon_t}^* - x^*\right)\right) \succeq (1 - Mrt)^2 \nabla^2 f\left(x^*\right)$ for all $t \leq \frac{1}{Mr}$. Therefore,

$$f\left(x_{\varepsilon_t}^*\right) - f\left(x^*\right) = \left(\int_0^1 \nabla f\left(x^* + t\left(x_{\varepsilon_t}^* - x^*\right)\right) dt\right)^\top \left(x_{\varepsilon_t}^* - x^*\right)$$

$$= \left(\int_0^1 \left(\int_0^1 \nabla^2 f\left(x^* + st\left(x_{\varepsilon_t}^* - x^*\right)\right) ds\right) t\left(x_{\varepsilon_t}^* - x^*\right) dt\right)^\top \left(x_{\varepsilon_t}^* - x^*\right)$$

$$= \left(x_{\varepsilon_t}^* - x^*\right)^\top \left(\int_0^1 \left(\int_0^t \nabla^2 f\left(x^* + s\left(x_{\varepsilon_t}^* - x^*\right)\right) ds\right) dt\right) \left(x_{\varepsilon_t}^* - x^*\right)$$

$$= \left(x_{\varepsilon_t}^* - x^*\right)^\top \left(\int_0^1 (1 - t) \nabla^2 f\left(x^* + t\left(x_{\varepsilon_t}^* - x^*\right)\right) dt\right) \left(x_{\varepsilon_t}^* - x^*\right)$$

$$\geq \left(\int_0^{\min\{\frac{1}{Mr}, 1\}} (1 - t)(1 - Mrt)^2 dt\right) \left(x_{\varepsilon_t}^* - x^*\right)^\top \nabla^2 f\left(x^*\right) \left(x_{\varepsilon_t}^* - x^*\right)$$

$$= r^2 \cdot \begin{cases} \frac{1}{3Mr} - \frac{1}{12M^2r^2} & Mr \geq 1 \\ \frac{1}{12}M^2r^2 - \frac{1}{3}Mr + \frac{1}{2} & Mr < 1 \end{cases}.$$

On the other hand,

$$f\left(x_{\varepsilon_t}^*\right) - f\left(x^*\right) = f\left(x_{\varepsilon_t}^*\right) - f_{\varepsilon_t}\left(x^*\right) + \frac{\varepsilon_t}{2}\|x^* - x_0\|_2^2$$

$$\leq f_{\varepsilon_t}\left(x_{\varepsilon_t}^*\right) - f_{\varepsilon_t}\left(x^*\right) + \frac{\varepsilon_t}{2}\|x^* - x_0\|_2^2$$

$$\leq \frac{\varepsilon_t}{2}\|x^* - x_0\|_2^2.$$

Combining these two inequalities we have:

$$\min\left\{\frac{r^2}{4}, \frac{r}{4M}\right\} \leq \frac{\varepsilon_t}{2}\|x^* - x_0\|_2^2.$$

Hence, either $r \leq \sqrt{2\varepsilon_t}\|x^* - x_0\|_2$ or $r \leq 2M\varepsilon_t\|x^* - x_0\|_2^2$. Since $\sqrt{2\varepsilon_t} \leq \frac{1}{M\|x^*-x_0\|_2}$, we must have $r \leq \sqrt{2\varepsilon_t}\|x^* - x_0\|_2$. For the second conclusion in Lemma 9, note that $f_{\varepsilon_t}$ is also self-concordant with constant $M$, and $f_{\varepsilon_{t-1}} = f_{\varepsilon_t} + \frac{\varepsilon_{t-1}-\varepsilon_t}{2}\|x^* - x_0\|_2^2$, so we can use the same argument on $f_{\varepsilon_{t-1}}$ compared with $f_{\varepsilon_t}$, then the result leads to (42). $\square$

We extract the inner loop for solving sub-problems in Algorithm 5 to Algorithm 6 as shown above.

**Lemma 10.** *If we use Algorithm 6 to optimize the regularized objective function $f_{\varepsilon_t}$, while satisfying the following conditions:*

> *1. There exists a constant $q$ such that $M\left\|x_1 - x_{\varepsilon_t}^*\right\|_{x_{\varepsilon_t}^*}^{\varepsilon_t} \leq qMD\sqrt{\varepsilon_t} \leq \frac{1}{30}$.*

---

**Algorithm 6** Inner loop of Algorithm 5

---

1: **Requires:** Initial point $x_0 \in \mathbb{R}^d$, regularizer $\varepsilon_t$, matrix $\mathbf{G}_0 \in \mathbb{R}^{d \times d}$, distribution $\mathcal{D}$, parameter sequence $\{\eta_k\}$, stepsize $0 < \alpha_t < 1$.

2: **for** $k = 0, 1, 2 \ldots$ **do**

3: $\qquad x_{k+1} = x_k - \alpha_t \left( \mathbf{G}_k + \left( \frac{1}{\eta_k} + \varepsilon_t \right) \mathbf{I}_d \right)^{-1} \nabla f_{\varepsilon_t}(x_k)$

4: $\qquad$ Sample a random vector $\mathbf{s}_k \sim \mathcal{D}$ and compute $\mathbf{G}_{k+1} = \mathbf{SR1}(\nabla^2 f(x_1), \mathbf{G}_k, \mathbf{s}_k)$.

5: **end for**

---

$\quad$ 2. $\mathbf{G}_k + \left( \frac{1}{\eta_k} + \varepsilon_t \right) \mathbf{I}_d \succeq \nabla^2 f_{\varepsilon_t}(x_1), \alpha = (1 - 3qMD\sqrt{\varepsilon_t})^2.$

*Then if for every $k \geq 1$, we have $\frac{1}{\eta_k} \leq \frac{\sqrt{\varepsilon_t}}{8qMD}$, the following inequality holds:*

$$\left\| x_{k+1} - x_{\varepsilon_t}^* \right\|_{x_{\varepsilon_t}^*}^{\varepsilon_t} \leq \left( 1 - \frac{qMD}{2} \sqrt{\varepsilon_t} \right) \left\| x_k - x_{\varepsilon_t}^* \right\|_{x_{\varepsilon_t}^*}^{\varepsilon_t}. \tag{43}$$

*Proof of Lemma 10.* For simplicity, in this proof we replace $\|z\|_{x_{\varepsilon_t}^*}^{\varepsilon_t}$ by $\|z\|_*$. We denote $v_k = \|x_k - x_{\varepsilon_t}^*\|_*$, $\mathbf{J}_k = \int_0^1 \nabla^2 f_{\varepsilon_t} \left( x_{\varepsilon_t}^* + t \left( x_k - x_{\varepsilon_t}^* \right) \right) dt$. We use induction to prove that $\|x_{k+1} - x_{\varepsilon_t}^*\|_* \leq (1 - qMD\sqrt{\varepsilon_t}/2) \|x_k - x_{\varepsilon_t}^*\|_*.$

Suppose that we already have $v_{s+1} \leq v_s$ for all $s \leq k - 1$. Then we have

$$x_{k+1} - x_{\varepsilon_t}^* = \left( \mathbf{I}_d - \alpha_t \left( \mathbf{G}_k + \frac{1}{\eta_k} \mathbf{I}_d + \varepsilon_t^{-1} \mathbf{J}_k \right) \right) \left( x_k - x_{\varepsilon_t}^* \right).$$

Since $v_k \leq v_0$ and $Mv_0 \leq qMD\sqrt{\varepsilon_t}$, we have

$$\|x_k - x_1\|_{x_1} \leq \frac{1}{1 - Mv_0} \|x_k - x_1\|_* \leq \frac{1}{1 - Mv_0} \left( \|x_k - x_{\varepsilon_t}^*\|_* + \|x_1 - x_{\varepsilon_t}^*\|_* \right) \leq \frac{2v_0}{1 - Mv_0}.$$

Then for every $t \in [0, 1]$ we have

$$\nabla^2 f_{\varepsilon_t} \left( x_{\varepsilon_t}^* + t \left( x_k - x_{\varepsilon_t}^* \right) \right) \preceq \frac{1}{\left( 1 - M \left( t \|x_k - x_1\|_{x_1} + (1 - t) \|x_{\varepsilon_t}^* - x_1\|_{x_1} \right) \right)^2} \nabla^2 f_{\varepsilon_t}(x_1).$$

This implies

$$\nabla^2 f_{\varepsilon_t} \left( x_{\varepsilon_t}^* + t \left( x_k - x_{\varepsilon_t}^* \right) \right) \preceq \frac{1}{\left( 1 - \frac{2Mv_0}{1 - Mv_0} \right)^2} \nabla^2 f_{\varepsilon_t}(x_0).$$

Take integral for $t$ over $[0, 1]$ we have

$$\mathbf{J}_k \preceq \frac{1}{\left( 1 - \frac{2Mv_0}{1 - Mv_0} \right)^2} \nabla^2 f_{\varepsilon_t}(x_0) \preceq \frac{1}{\left( 1 - 3qMD\sqrt{\varepsilon_t} \right)^2} \nabla^2 f_{\varepsilon_t}(x_0).$$

By the same reason we have

$$\mathbf{J}_k \succeq \left( 1 - 3qMD\sqrt{\varepsilon_t} \right)^2 \nabla^2 f_{\varepsilon_t}(x_0).$$

Since $\mathbf{G}_k + \frac{1}{\eta_k} \mathbf{I}_d \succeq \nabla^2 f(x_0)$ and $\alpha_t \leq \left( 1 - 3qMD\sqrt{\varepsilon_t} \right)^2$, we can see that $\mathbf{G}_k + \frac{1}{\eta_k} \mathbf{I}_d + \varepsilon_t \mathbf{I}_d \succeq \alpha_t \mathbf{J}_k$, as a result,

$$\lambda_{\min} \left( \mathbf{I}_d - \alpha_t \left( \mathbf{G}_k + \frac{1}{\eta_k} \mathbf{I}_d + \varepsilon_t \mathbf{I}_d \right)^{-1} \mathbf{J}_k \right) \geq 0$$

Also by SR1 update we have $\mathbf{G}_k + \varepsilon_t \mathbf{I}_d \preceq \nabla^2 f_{\varepsilon_t}(x_0) \preceq \frac{1}{(1 - 3qMD\sqrt{\varepsilon_t})^2} \mathbf{J}_k$, and by our assumption, $\frac{1}{\eta_k} \leq \frac{\sqrt{\varepsilon_t}}{8qMD}$, so we can deduce that

$$\lambda_{\max}\left(\mathbf{I}_d - \alpha_t \left(\mathbf{G}_k + \frac{1}{\eta_k}\mathbf{I}_d + \varepsilon_t \mathbf{I}_d\right)^{-1} \mathbf{J}_k\right) \leq 1 - \alpha_t \lambda_{\max}^{-1}\left(\frac{1}{\eta_k}\mathbf{J}_k^{-1} + \frac{1}{(1 - 3qMD\sqrt{\varepsilon_t})^2}\mathbf{I}_d\right)$$

$$\leq 1 - \frac{\alpha_t}{\frac{2}{\eta_k \varepsilon_t} + \frac{1}{(1 - 3qMD\sqrt{\varepsilon_t})^2}}$$

$$\leq 1 - \frac{\left(1 - 3qMD\sqrt{\varepsilon_t}\right)^2}{\frac{1}{4qMD\sqrt{\varepsilon_t}} + \frac{1}{(1 - 3qMD\sqrt{\varepsilon_t})^2}}$$

$$\leq 1 - 2qMD\left(1 - 3qMD\sqrt{\varepsilon_t}\right)^2 \sqrt{\varepsilon_t}.$$

Hence,
$$\|x_{k+1} - x_{\varepsilon_t}^*\|_{\mathbf{J}_k} \leq (1 - 1.62qMD\sqrt{\varepsilon_t}) \|x_k - x_{\varepsilon_t}^*\|_{\mathbf{J}_k}.$$

Since $M\|x_k - x_{\varepsilon_t}^*\|_* \leq qMD\sqrt{\varepsilon_t}$, we have

$$\nabla^2 f_{\varepsilon_t}\left(x_{\varepsilon_t}^* + t\left(x_k - x_{\varepsilon_t}^*\right)\right) \preceq \frac{1}{(1 - tMv_k)^2} \nabla^2 f_{\varepsilon_t}\left(x_{\varepsilon_t}^*\right).$$

This could imply $\|z\|_{\mathbf{J}_k} \in \left[\sqrt{1 - Mr_k}\|z\|_*, \frac{1}{\sqrt{1 - Mr_k}}\|z\|_*\right]$. At last, we can derive that

$$\|x_{k+1} - x_{\varepsilon_t}^*\|_* \leq \frac{1 - 1.62qMD\sqrt{\varepsilon_t}}{1 - qMD\sqrt{\varepsilon_t}} \|x_k - x_{\varepsilon_t}^*\|_* \leq (1 - qMD\sqrt{\varepsilon_t}/2) \|x_k - x_{\varepsilon_t}^*\|_*.$$

Then we finish the proof by induction. $\qquad\square$

**Lemma 11.** *Under the condition of Theorem 6 and follow the same notation. Denote constants*

$$R_0 = \frac{D}{4}, R_1 = \frac{21D}{320}, R_2 = \frac{3MD}{2}, R_3 = 2\sqrt{2}MD, R_4 = \frac{19MD}{40}, c^* = \frac{1}{4096}$$

*if*

$$\varepsilon_t < \varepsilon_{t-1} \leq (1 + c^*)\varepsilon_t \leq \frac{1}{121M^2D^2},$$

*and the approximate solution $x_{t-2}, x_{t-1}$ and matrix $\mathbf{H}_{t-2}, \tilde{\mathbf{G}}_{t-1}$ satisfy $\mathbf{H}_{t-2} \preceq \nabla^2 f(x_{t-2})$ and*

$$\|x_{t-1} - x_{t-2}\|_{x_{t-1}} \leq R_0\sqrt{\varepsilon_{t-1}}, \tag{44}$$

$$\left\|x_{t-1} - x_{\varepsilon_{t-1}}^*\right\|_{x_{\varepsilon_{t-1}}^*}^{\varepsilon_{t-1}} \leq R_1\sqrt{\varepsilon_{t-1}}, \tag{45}$$

$$\left\|\tilde{\mathbf{G}}_{t-1} - \nabla^2 f(x_{t-1})\right\|_2 \leq R_2\sqrt{\varepsilon_{t-1}}. \tag{46}$$

*Choose $\beta_{t-1} = MR_0\sqrt{\varepsilon_{t-1}}$, then the correction approximate matrix $\mathbf{H}_{t-1}$ satisfies $\mathbf{H}_{t-1} \preceq \nabla^2 f(x_{t-1})$ and*

$$\left\|\mathbf{H}_{t-1} - \nabla^2 f(x_{t-1})\right\|_2 \leq R_3\sqrt{\varepsilon_t}. \tag{47}$$

*Next, if the approximate solution $x_t$ satisfies*

$$\left\|x_t - x_{\varepsilon_t}^*\right\|_{x_{\varepsilon_t}^*}^{\varepsilon_t} \leq R_1\sqrt{\varepsilon_t}, \tag{48}$$

*then*

$$\|x_t - x_{t-1}\|_{x_t} \leq R_0\sqrt{\varepsilon_t}. \tag{49}$$

*Moreover, if $\left\|\tilde{\mathbf{G}}_t - \nabla^2 f(x_{t-1})\right\|_2 \leq R_4\sqrt{\varepsilon_t}$, then we have*

$$\left\|\tilde{\mathbf{G}}_t - \nabla^2 f(x_t)\right\|_2 \leq R_2\sqrt{\varepsilon_t}. \tag{50}$$

*Proof.* First, we prove $\mathbf{H}_{t-1} \preceq \nabla^2 f(x_{t-1})$ and (47). Since $\mathbf{H}_{t-1} = (1-\beta_{t-1})^2 \tilde{\mathbf{G}}_{t-1}$, we have

$$\left\|\mathbf{H}_{t-1} - \nabla^2 f(x_{t-1})\right\|_2 \le 2\beta_{t-1}\left\|\tilde{\mathbf{G}}_{t-1}\right\|_2 + \left\|\tilde{\mathbf{G}}_{t-1} - \nabla^2 f(x_{t-1})\right\|_2$$

$$\le (2MR_0 + R_2)\sqrt{\varepsilon_{t-1}} \overset{(66)}{\le} R_3\sqrt{\varepsilon_t}.$$

By the property of SR1 update we have $\tilde{\mathbf{G}}_{t-1} \preceq \nabla^2 f(x_{t-2})$. Using the self-concordancy we have

$$\mathbf{H}_{t-1} = (1-\beta_{t-1})^2 \tilde{\mathbf{G}}_{t-1} \preceq (1-\beta_{t-1})^2 \nabla^2 f(x_{t-2})$$

$$\preceq \frac{(1-\beta_{t-1})^2}{(1 - M\|x_{t-1} - x_{t-2}\|_{x_{t-1}})^2}\nabla^2 f(x_{t-1})$$

$$\overset{(44)}{\preceq} \nabla^2 f(x_{t-1}).$$

Second, let us bound $\|x_t - x_{t-1}\|_{x_t}$ using the self-concordance property. We have

$$\|x_t - x_{t-1}\|_{x_t}^{\varepsilon_t} \le \frac{1}{1 - M\|x_t - x_{\varepsilon_t}^*\|_{x_{\varepsilon_t}^*}^{\varepsilon_t}}\|x_t - x_{t-1}\|_{x_{\varepsilon_t}^*}^{\varepsilon_t}. \tag{51}$$

We need to bound $\|x_t - x_{t-1}\|_{x_{\varepsilon_t}^*}^{\varepsilon_t}$, and we have

$$\|x_t - x_{t-1}\|_{x_{\varepsilon_t}^*}^{\varepsilon_t} \le \left\|x_t - x_{\varepsilon_t}^*\right\|_{x_{\varepsilon_t}^*}^{\varepsilon_t} + \left\|x_{\varepsilon_t}^* - x_{\varepsilon_{t-1}}^*\right\|_{x_{\varepsilon_t}^*}^{\varepsilon_t} + \left\|x_{\varepsilon_{t-1}}^* - x_{t-1}\right\|_{x_{\varepsilon_t}^*}^{\varepsilon_t}. \tag{52}$$

By (48) and Lemma 9, we have

$$\left\|x_t - x_{\varepsilon_t}^*\right\|_{x_{\varepsilon_t}^*}^{\varepsilon_t} \le R_1\sqrt{\varepsilon_t}, \quad \left\|x_{\varepsilon_t}^* - x_{\varepsilon_{t-1}}^*\right\|_{x_{\varepsilon_t}^*}^{\varepsilon_t} \le \sqrt{2(\varepsilon_{t-1} - \varepsilon_t)}\|x^* - x_0\|_2.$$

Denote $w = \sqrt{\frac{2(\varepsilon_{t-1}-\varepsilon_t)}{\varepsilon_t}}\|x^* - x_0\|_2$, then the last term in (52) can be bounded by (45) and Lemma 9:

$$\left\|x_{\varepsilon_{t-1}}^* - x_{t-1}\right\|_{x_{\varepsilon_t}^*}^{\varepsilon_t} \le \frac{\left\|x_{\varepsilon_{t-1}}^* - x_{t-1}\right\|_{x_{\varepsilon_{t-1}}^*}^{\varepsilon_t}}{1 - M\left\|x_{\varepsilon_{t-1}}^* - x_{\varepsilon_t}^*\right\|_{x_{\varepsilon_t}^*}^{\varepsilon_t}} \le \frac{R_1\sqrt{\varepsilon_{t-1}}}{1 - Mw\sqrt{\varepsilon_t}}$$

Hence, $\|x_t - x_{t-1}\|_{x_{\varepsilon_t}^*}^{\varepsilon_t} \le R_1\sqrt{\varepsilon_t} + w\sqrt{\varepsilon_t} + \frac{R_1\sqrt{\varepsilon_{t-1}}}{1 - Mw\sqrt{\varepsilon_t}}$, combining with (51) we have

$$\|x_t - x_{t-1}\|_{x_t}^{\varepsilon_t} \le \frac{1}{1 - MR_1\sqrt{\varepsilon_t}}\left(R_1\sqrt{\varepsilon_t} + w\sqrt{\varepsilon_t} + \frac{R_1\sqrt{\varepsilon_{t-1}}}{1 - Mw\sqrt{\varepsilon_t}}\right). \tag{53}$$

This leads to (49) by (67) in Lemma 25.

Finally, we use (47), (49) to prove (50). We have

$$\left\|\tilde{\mathbf{G}}_t - \nabla^2 f(x_t)\right\|_2 \le \left\|\tilde{\mathbf{G}}_t - \nabla^2 f(x_{t-1})\right\|_2 + \left\|\nabla^2 f(x_{t-1}) - \nabla^2 f(x_t)\right\|_2$$

$$\le R_4\sqrt{\varepsilon_t} + \left(\frac{1 - (1 - M\|x_t - x_{t-1}\|_{x_t})^4}{(1 - M\|x_t - x_{t-1}\|_{x_t})^2}\right)\left\|\nabla^2 f(x_t)\right\|_2$$

$$\overset{(49)}{\le} R_4\sqrt{\varepsilon_t} + \frac{1 - (1 - MR_0\sqrt{\varepsilon_t})^4}{(1 - MR_0\sqrt{\varepsilon_t})^2} \overset{(68)}{\le} R_2\sqrt{\varepsilon_t}.$$

$\square$

**Lemma 12.** *Suppose that $\varepsilon_t \le \varepsilon_{t-1} \le \frac{1}{M^2 D^2}$ and we already have*

$$\left\|x_{\varepsilon_{t-1}}^* - x_{\varepsilon_t}^*\right\|_{x_{\varepsilon_t}^*}^{\varepsilon_t} \le \frac{D}{32\sqrt{2}}\sqrt{\varepsilon_t}, \quad \left\|x_{t-1} - x_{\varepsilon_{t-1}}^*\right\|_{x_{\varepsilon_{t-1}}^*}^{\varepsilon_{t-1}} \le \frac{D}{33\sqrt{2}}\sqrt{\varepsilon_t}.$$

*Then we have*

$$\left\|x_{t-1} - x_{\varepsilon_t}^*\right\|_{x_{\varepsilon_t}^*}^{\varepsilon_t} \le \frac{D}{16\sqrt{2}}\sqrt{\varepsilon_t}. \tag{54}$$

*Proof.* Note that $f_{\varepsilon_t}(x)$ is self-concordant with constant $M$, so we have

$$\left\| x_{t-1} - x^*_{\varepsilon_{t-1}} \right\|^{\varepsilon_t}_{x^*_{\varepsilon_t}} \leq \frac{1}{1 - M \left\| x^*_{\varepsilon_{t-1}} - x^*_{\varepsilon_t} \right\|^{\varepsilon_t}_{x^*_{\varepsilon_{t-1}}}} \left\| x_{t-1} - x^*_{\varepsilon_{t-1}} \right\|^{\varepsilon_t}_{x^*_{\varepsilon_{t-1}}}$$

$$\leq \frac{1}{1 - M \frac{D\sqrt{\varepsilon_t}}{32\sqrt{2}}} \left\| x_{t-1} - x^*_{\varepsilon_{t-1}} \right\|^{\varepsilon_t}_{x^*_{\varepsilon_{t-1}}}$$

$$\leq \frac{32\sqrt{2}}{32\sqrt{2} - 1} \left\| x_{t-1} - x^*_{\varepsilon_{t-1}} \right\|^{\varepsilon_{t-1}}_{x^*_{\varepsilon_{t-1}}}$$

$$\leq \frac{32\sqrt{2}}{32\sqrt{2} - 1} \cdot \frac{D}{33\sqrt{2}} \sqrt{\varepsilon_t} \leq \frac{D}{32\sqrt{2}} \sqrt{\varepsilon_t}.$$

Hence,

$$\left\| x_{t-1} - x^*_{\varepsilon_t} \right\|^{\varepsilon_t}_{x^*_{\varepsilon_t}} \leq \left\| x^*_{\varepsilon_{t-1}} - x^*_{\varepsilon_t} \right\|^{\varepsilon_t}_{x^*_{\varepsilon_t}} + \left\| x_{t-1} - x^*_{\varepsilon_{t-1}} \right\|^{\varepsilon_t}_{x^*_{\varepsilon_t}} \leq \frac{D}{16\sqrt{2}} \sqrt{\varepsilon_t}.$$

$\square$

## F TECHNICAL LEMMAS

In this section, we present technical lemmas that are used in the previous proofs. Among these lemmas, Lemma 16, Lemma 17, and Lemma 21 are well-known and can be found in classical textbooks. As such, we do not provide their proofs.

**Lemma 13.** *The function $h(t) = t \ln t$ decreases in the interval $\left( 0, \frac{1}{e} \right]$.*

*Proof.* This simply follows from the $h(t)$'s derivative: $h'(t) = 1 + \ln(t) \leq 0$, for $t \leq \frac{1}{e}$. $\square$

**Lemma 14.** *Let $\{a_n\}_{n \geq 0}$ be a sequence of real positive numbers and $c > 0$ such that $a_{n+1} \leq a_n - c a_n^2$, then for all $n \in \mathbb{N}$, we have $a_n \leq \frac{a_0}{cn + a_0}$.*

*Proof.* Since $a_{n+1} \leq a_n - c a_n^2$, we have $\frac{1}{a_{n+1}} \geq \frac{1}{a_n} + \frac{c}{1 - c a_n}$. Hence, we get

$$\frac{1}{a_n} \geq \sum_{i=0}^{n-1} \frac{c}{1 - c a_i} + \frac{1}{a_0} \geq nc + \frac{1}{a_0}.$$

This implies $a_n \leq \frac{a_0}{cn + a_0}$. $\square$

**Lemma 15.** *Let $\{a_n\}_{n \geq 0}$ be a sequence of real positive numbers that do not increase. Let $c > 0$ be a constant. For every $n \in \mathbb{N}$, denote $A_n = \left\{ k \in \mathbb{N}, k \leq n : a_{k+1} \leq a_k - c a_k^2 \right\}$, then we have*

$$a_n \leq \frac{a_0}{a_0 + c|A_n|},$$

*where $|A_n|$ denotes the number of elements in the set $A_n$.*

*Proof.* Construct the sequence $\{a_{n_k}\}$ by ordering the elements of $\bigcup_{i=0}^{+\infty} A_i$ according to their subscripts in increasing order. Denote $m = |A_n|$, then $a_{n_m} \geq a_n \geq a_{n_{m+1}}$. By Lemma 14, $a_{n_m} \leq \frac{a_0}{a_0 + cm}$. Therefore, $a_n \leq a_{n_m} \leq \frac{a_0}{a_0 + cm}$. $\square$

**Lemma 16** (PaleyZygmund inequality). *Let $X \geq 0$ be a nonnegative random variable. Then for all $0 < \theta < 1$, we have*

$$\mathbb{P}\left( X \geq \theta \mathbb{E}[X] \right) \geq \frac{(1 - \theta)^2 \mathbb{E}[X]^2}{\mathbb{E}[X^2]}. \tag{55}$$

**Lemma 17** (Chernoff bound for Bernoulli variables)**.** *Let* $X_1, \cdots, X_n \overset{i.i.d}{\sim} \text{Bernoulli}(1, p)$, *then for every* $0 < \delta < 1$, *we have*

$$\mathbb{P}\left(\sum_{i=1}^n X_i \le (1 - \delta)np\right) \le e^{-\frac{\delta^2 np}{2}}. \tag{56}$$

Lemma 18 is our main tool for proving high-probability bounds. This lemma extends classical Chernoff bounds to dependent processes by requiring only a one-sided lower bound on conditional success probabilities. This allows deriving exponential concentration inequalities similar to the independent case, making it particularly useful for analyzing adaptive algorithms and sequential decision processes where independence assumptions fail but some probabilistic structure remains. The result provides a powerful tool for establishing high-probability guarantees in dependent settings.

**Lemma 18** (Coupling)**.** *Consider a random process* $X_k, k \in \mathbb{N}^*$, *where* $X_k$ *is taken in* $\{0, 1\}$. *Denote* $\mathcal{F}_k$ *as the* $\sigma$-*algebra generated by* $X_1, \cdots, X_k$. *Suppose that for all* $k \ge 1$, *we have* $\mathbb{P}(X_k = 1 | \mathcal{F}_{k-1}) \ge p$, *then for any* $k \ge 0, t \ge 0$, *we have*

$$\mathbb{P}(X_1 + \cdots + X_k \ge t) \ge \mathbb{P}(Y_1 + \cdots + Y_k \ge t),$$

*where* $Y_1, \cdots, Y_k \overset{i.i.d}{\sim} \text{Bernoulli}(1, p)$. *Moreover, for every* $0 < \delta < 1, n \in \mathbb{N}^*$, *we have*

$$\mathbb{P}\left(\sum_{i=1}^n X_i \le (1 - \delta)np\right) \le e^{-\frac{\delta^2 np}{2}}. \tag{57}$$

*Proof.* We construct an auxiliary process $\{Z_k\}_{k \in \mathbb{N}^*}$ with $Z_k \in \{0, 1\}$ as follows:

Since $X_1, \ldots, X_k$ take on finitely many values, each event in $\mathcal{F}_k$ can be expressed as a union of atomic events. For each atomic event $A \in \mathcal{F}_{k-1}$ where $\mathbb{P}(X_k = 1 \mid A) = q_A \ge p$, we define $Z_k|_A$ to be an independent Bernoulli random variable with parameter $\frac{p}{q_A}$, i.e., $Z_k|_A \sim \text{Bernoulli}\left(1, \frac{p}{q_A}\right)$, independent of $X_k|_A$. By repeating this construction for all atomic events in $\mathcal{F}_{k-1}$, we obtain a well-defined random variable $Z_k \in \{0, 1\}$ satisfying:

$$\mathbb{P}(X_k Z_k = 1 \mid \mathcal{F}_{k-1}) = p.$$

Since $X_k \ge X_k Z_k$, it suffices to prove that

$$\mathbb{P}(X_1 Z_1 + \cdots + X_k Z_k = t) = \mathbb{P}(Y_1 + \cdots + Y_k = t). \tag{58}$$

Now we can prove (58) by induction. Suppose that (58) holds for $k - 1$, then we have

$$\begin{aligned}
&\mathbb{P}(X_1 Z_1 + \cdots X_k Z_k = t) \\
&= \mathbb{P}(X_k Z_k = 0 | X_1 Z_1 + \cdots X_{k-1} Z_{k-1} = t)\mathbb{P}(X_1 Z_1 + \cdots X_{k-1} Z_{k-1} = t) \\
&\quad + \mathbb{P}(X_k Z_k = 1 | X_1 Z_1 + \cdots X_{k-1} Z_{k-1} = t - 1)\mathbb{P}(X_1 Z_1 + \cdots X_{k-1} Z_{k-1} = t - 1) \\
&= (1 - p)\mathbb{P}(X_1 Z_1 + \cdots X_{k-1} Z_{k-1} = t) + p\mathbb{P}(X_1 Z_1 + \cdots X_{k-1} Z_{k-1} = t - 1) \\
&= (1 - p)\mathbb{P}(Y_1 + \cdots Y_{k-1} = t) + p\mathbb{P}(Y_1 + \cdots Y_{k-1} = t - 1) \\
&= \mathbb{P}(Y_1 + \cdots + Y_k = t).
\end{aligned}$$

Hence (58) holds for all $k, t$. This implies (58) and implies (57) by (58) and (56). $\qquad \square$

**Lemma 19.** *Let* $\mathbf{A} \in \mathbb{R}^{d \times d}, \mathbf{A} \succeq 0$ *and* $X$ *be a random variable that satisfies (4). Let* $X_i \overset{i.i.d}{\sim} X$ *and* $\mathbf{X} = (X_1, \cdots, X_d)^\top$. *Then we have*

$$\mathbb{E}\left[(\mathbf{X}^\top \mathbf{A} \mathbf{X})^2\right] \le (3 + C_1)\text{tr}(\mathbf{A})^2. \tag{59}$$

*Proof.* The left side of (59) is actually

$$\mathbb{E}\left[\left(\sum_{i,j} A_{ij} X_i X_j\right)^2\right] = \sum_{i,j,k,l} A_{ij} A_{kl} \mathbb{E}\left[X_i X_j X_k X_l\right].$$

Denote

$$A_1 = \{(i,j,k,l) : i = j = k = l\},$$

$$A_2 = \{(i,j,k,l) : i = j, k = l, i \neq k \text{ or its permutation}\}.$$

Then the term $\mathbb{E}\left[X_i X_j X_k X_l\right]$ satisfies:

$$\mathbb{E}\left[X_i X_j X_k X_l\right] = \begin{cases} C_1 & (i,j,k,l) \in A_1 \\ 1 & (i,j,k,l) \in A_2 \\ 0 & (i,j,k,l) \in (A_1 \cup A_2)^c \end{cases} \tag{60}$$

Hence,

$$\mathbb{E}\left[\left(\sum_{i,j} A_{ij} X_i X_j\right)^2\right] = C_1 \sum_i A_{ii}^2 + \sum_{1 \leq i < j \leq d} (4A_{ij}^2 + 2A_{ii}A_{jj})$$

$$\leq (2 + C_1)\|\mathbf{A}\|_F^2 + \mathrm{tr}(\mathbf{A})^2 = (3 + C_1)\mathrm{tr}(\mathbf{A})^2.$$

$\square$

**Lemma 20.** *Let $X_1, \cdots, X_n$ be $n$ random variables such that $\mathbb{E}\left[X_i^2\right] = 1$ and $\sup_{1 \leq i \leq n} \mathbb{E}\left[X_i^4\right] = M$. Then for any $a_1, \cdots, a_n \geq 0$, we have*

$$\mathbb{E}\left[\left(\sum_{i=1}^n a_i X_i^2\right)^2\right] \leq M\left(\sum_{i=1}^n a_i\right)^2. \tag{61}$$

*This implies*

$$\mathbb{P}\left(\sum_{i=1}^n a_i X_i^2 \geq \frac{1}{2}\sum_{i=1}^n a_i\right) \geq \frac{1}{4M}. \tag{62}$$

*Proof.* By AM-GM inequality, $\mathbb{E}\left[X_i^2 X_j^2\right] \leq \frac{1}{2}\mathbb{E}\left[X_i^4 + X_j^4\right] \leq M$. Then the left-hand side of the equation (61) is

$$\sum_{i=1}^n a_i^2 \mathbb{E}\left[X_i^4\right] + \sum_{1 \leq i < j \leq n} a_i a_j \mathbb{E}\left[X_i^2 X_j^2\right] \leq M\left(\sum_{i=1}^n a_i\right)^2$$

Hence, (61) holds and implies (62) by Lemma 16. $\square$

**Lemma 21** (Weyl's Inequality for Hermitian Matrices). *Let $\mathbf{A}$ and $\mathbf{B}$ be $n \times n$ Hermitian matrices. Denote their eigenvalues in non-increasing order as:*

$$\lambda_1(\mathbf{A}) \geq \lambda_2(\mathbf{A}) \geq \cdots \geq \lambda_n(\mathbf{A}),$$

$$\lambda_1(\mathbf{B}) \geq \lambda_2(\mathbf{B}) \geq \cdots \geq \lambda_n(\mathbf{B}),$$

*Then for any $i, j \geq 1$ with $i + j - 1 \leq n$, we have:*

$$\lambda_{i+j-1}(\mathbf{A} + \mathbf{B}) \leq \lambda_i(\mathbf{A}) + \lambda_j(\mathbf{B}),$$

*and for any $i, j \geq 1$ with $i + j - 1 \geq n$, we have*

$$\lambda_i(\mathbf{A}) + \lambda_j(\mathbf{B}) \leq \lambda_{i+j-n}(\mathbf{A} + \mathbf{B}).$$

**Remark.** *If $\mathbf{B} = -\mathbf{u}\mathbf{u}^\top$ and $\mathbf{u} \neq 0$, then $\lambda_n(\mathbf{B}) < 0$ and $\lambda_i(\mathbf{B}) = 0, i < n$. In this case, $\lambda_{i+1}(\mathbf{A}) \leq \lambda_i(\mathbf{A} - \mathbf{B}) \leq \lambda_i(\mathbf{A})$. Moreover, we have $|\lambda_i(\mathbf{A}) - \lambda_i(\mathbf{A} + \mathbf{B})| \leq \|\mathbf{B}\|_2$. Hence, the spectrum of Hermitian matrices is stable under perturbation.*

**Lemma 22.** *Consider a rational fraction*

$$f(x) = \sum_{i=1}^{m} \frac{a_i}{b_i - x}, \quad a_i > 0, b_1 > b_2 > \cdots > b_m \geq 0.$$

*For any $t > 0$, the equation*

$$f(x) = t$$

*has $m$ solutions $x_1 > x_2 > \cdots > x_m$ such that $x_i \in (b_{i+1}, b_i)$ for $1 \leq i \leq m - 1$ and $x_m \in (-\infty, b_m)$. Moreover,*

$$\sum_{i=1}^{m} x_i = \sum_{i=1}^{m} b_i - \frac{1}{t} \sum_{i=1}^{m} a_i. \tag{63}$$

*Proof.* Let us consider each $(b_{i+1}, b_i)$, note that

$$f'(x) = \sum_{i=1}^{m} \frac{a_i}{(b_i - x)^2} > 0,$$

and $\lim_{x \to b_{i+1}^+} f(x) = -\infty$, $\lim_{x \to b_i^-} f(x) = +\infty$. Hence, by Intermediate Value Theorem, for any $t \geq 0$, the equation $f(x) = t$ has a unique root in $(b_{i+1}, b_i)$. Similarly, it has a unique root in $(-\infty, b_m)$. Multiplying $\prod_{i=1}^{m}(b_i - x)$ to the equation, they can be also seen as the roots of a polynomial

$$p(x) = \sum_{i=1}^{m} a_i \prod_{j \neq i}(b_j - x) - t \prod_{i=1}^{m}(b_i - x).$$

And (63) follows directly by Vieta's formula. $\qquad\square$

The next two lemmas ( Lemma 23, Lemma 24) are well-known results in numerical algebra. Their proofs can be found in any numerical algebra textbook. The famous Divide-and-Conquer algorithm for solving eigenvalues of tridiagonal Hermitian matrices is based on the following theory.

**Lemma 23.** *Let $\mathbf{D} = diag(d_1, \cdots, d_n)$ be a diagonal matrix such that $d_1 > d_2 > \cdots > d_n$, assume that $\rho \neq 0$, $\mathbf{u} \in \mathbb{R}^n$ and each coordinate of $\mathbf{u} \in \mathbb{R}^n$ is non-zero. If $\mathbf{v} \in \mathbb{R}^n$ and $\lambda \in \mathbb{R}$ satisfy*

$$(\mathbf{D} + \rho \mathbf{u}\mathbf{u}^\top)\mathbf{v} = \lambda \mathbf{v},$$

*then $\mathbf{v}^\top \mathbf{u} \neq 0$, and $\mathbf{D} - \lambda \mathbf{I}_n$ is invertible.*

*Proof.* If $\mathbf{v}^\top \mathbf{u} = 0$, then $\mathbf{D}\mathbf{v} = \lambda \mathbf{v}, \mathbf{v} \neq 0$, hence $\lambda$ is the eigenvalue of $\mathbf{D}$. Note that $\mathbf{D}$'s diagonals are different from each other, so there exists $i$ such that $d_i = \lambda$ and $\mathbf{v} = \alpha \mathbf{e}_i, \alpha \neq 0$. Therefore, we have $0 = \mathbf{v}^\top \mathbf{u} = \alpha \mathbf{u}^\top \mathbf{e}_i$, contradicting the condition that each coordinate of $\mathbf{u}$ is non-zero.

Besides, if $\mathbf{D} - \lambda \mathbf{I}_n$ is singular, then there exists $i$ such that $\mathbf{e}_i^\top (\mathbf{D} - \lambda \mathbf{I}_n) = 0$ and thus we have

$$0 = \mathbf{e}_i^\top (\mathbf{D} - \lambda \mathbf{I}_n)\mathbf{v} = -\rho \mathbf{u}^\top \mathbf{v} \mathbf{e}_i^\top \mathbf{u},$$

but $\rho \mathbf{u}^\top \mathbf{v} \neq 0$, so $\mathbf{e}_i^\top \mathbf{u} = 0$, a contradiction. $\qquad\square$

**Lemma 24.** *Let $\mathbf{D} = diag(d_1, \cdots, d_n)$ be a diagonal matrix such that $d_1 > d_2 > \cdots > d_n$, $\mathbf{u} \in \mathbb{R}^n$, and suppose that each coordinate of $\mathbf{u}$ is non-zero and $\rho > 0$. Denote $\lambda_1 \geq \lambda_2 \geq \cdots \geq \lambda_n$ as the eigenvalues of $\mathbf{D} - \rho \mathbf{u}\mathbf{u}^\top$. Then $\lambda_i$ are distinct and they are exactly the roots of*

$$1 - \rho \left( \frac{u_1^2}{d_1 - \lambda_i} + \cdots + \frac{u_n^2}{d_n - \lambda_i} \right) = 0, \tag{64}$$

*where $u_i$ denotes the $i$-th component of $\mathbf{u}$.*

*Proof.* First, consider the case when $d_i$ are distinct from each other. Denote $\mathbf{v}_i$ as the unit eigenvector with respect to $\lambda_i$. Then we have

$$(\mathbf{D} - \rho\mathbf{u}\mathbf{u}^\top)\mathbf{v}_i = \lambda_i\mathbf{v}_i.$$

By Lemma 23, $\mathbf{D} - \lambda_i\mathbf{I}_n$ is invertible and $\mathbf{u}^\top\mathbf{v}_i \neq 0$. Therefore,

$$\mathbf{v}_i = \rho\mathbf{u}^\top\mathbf{v}_i(\mathbf{D} - \lambda_i\mathbf{I}_n)^{-1}\mathbf{u}.$$

Left multiply both sides by $\mathbf{u}^\top$, note that $\mathbf{u}^\top\mathbf{v}_i \neq 0$, we have

$$1 = \rho\mathbf{u}^\top(\mathbf{D} - \lambda_i\mathbf{I}_n)^{-1}\mathbf{u},$$

this is equivalent to

$$1 - \rho\left(\frac{u_1^2}{d_1 - \lambda_i} + \cdots + \frac{u_n^2}{d_n - \lambda_i}\right) = 0.$$

By Lemma 22, the above equation has exactly $n$ roots, each belonging to $(d_2, d_1), (d_3, d_2), \cdots, (-\infty, d_n)$ respectively. Thus the proof is complete. $\qquad\square$

**Lemma 25.** *Under the condition of Lemma 11, we have the following inequalities:*

$$w \overset{def}{=} \sqrt{\frac{2(\varepsilon_{t-1} - \varepsilon_t)}{\varepsilon_t}}\|x^* - x_0\|_2 \leq \frac{D}{32}. \tag{65}$$

$$(2MR_0 + R_2)\sqrt{\varepsilon_{t-1}} \leq R_3\sqrt{\varepsilon_t}. \tag{66}$$

$$\frac{1}{1 - MR_1\sqrt{\varepsilon_t}}\left(R_1\sqrt{\varepsilon_t} + w\sqrt{\varepsilon_t} + \frac{R_1\sqrt{\varepsilon_{t-1}}}{1 - Mw\sqrt{\varepsilon_t}}\right) \leq R_0\sqrt{\varepsilon_t}. \tag{67}$$

$$R_4\sqrt{\varepsilon_t} + \frac{1 - (1 - MR_0\sqrt{\varepsilon_t})^4}{(1 - MR_0\sqrt{\varepsilon_t})^2} \leq R_2\sqrt{\varepsilon_t}. \tag{68}$$

*Proof.* Since $\varepsilon_t < \varepsilon_{t-1} \leq (1 + c^*)\varepsilon_t \leq \frac{1}{121M^2D^2}$, we have

$$\sqrt{\frac{2(\varepsilon_{t-1} - \varepsilon_t)}{\varepsilon_t}}\|x^* - x_0\|_2 \leq \sqrt{2c^*}\|x^* - x_0\|_2 \leq \frac{D}{32\sqrt{2}},$$

this implies (65). The inequality (66) is equivalent to

$$\left(2M \cdot \frac{1}{4M} + \frac{3}{2}\right)\sqrt{\varepsilon_{t-1}} \leq 2\sqrt{2\varepsilon_t},$$

this is obvious because we have $\varepsilon_{t-1} \leq 2\varepsilon_t$ by $c^* \leq 1$.

For (67), notice that $w \leq \frac{D}{32}$ and $R_1 \leq \frac{21D}{320}$, we only need to prove

$$\frac{320}{299}\left(\frac{D}{32}\sqrt{\varepsilon_t} + \frac{21D}{320}\sqrt{\varepsilon_t} + \frac{32}{31} \cdot \frac{21D}{320}\sqrt{\varepsilon_{t-1}}\right) \leq \frac{D}{4}\sqrt{\varepsilon_t}.$$

This can be done by numerical calculation.

Finally, let us prove (68). Since $R_0 \leq \frac{D}{4}$, $R_4 = \frac{19MD}{40}$, we only need to prove

$$\frac{19}{40}\sqrt{\varepsilon_t} + \frac{1 - (1 - 0.25\sqrt{\varepsilon_t})^4}{(1 - 0.25\sqrt{\varepsilon_t})^2} \leq \frac{3}{2}\sqrt{\varepsilon_t}.$$

We can see that for $x \in \left(0, \frac{1}{44}\right)$, $(1 - x)^2 \geq 1 - 2x$, as a result,

$$\frac{1}{(1 - x)^2} - (1 - x)^2 \leq \frac{1}{1 - 2x} - (1 - 2x) = \frac{2x}{1 - 2x} + 2x \leq \frac{45}{11}x.$$

Take $x = 0.25\sqrt{\varepsilon_t}$ and then we finish the proof.

$\qquad\square$

## G   LIMITATIONS

Our results do not yet characterize the accelerated convergence resulting from faster eigenvalue decay beyond a merely bounded trace. Although AGD and SR1 share the same worst-case convergence rate, there exist classes of problems where SR1 converges more rapidly. This is also supported by our experimental findings, where SR1-based methods demonstrate faster convergence on problems with rapidly decaying Hessian eigenvalues. We hypothesize that this acceleration arises from the larger eigengap induced by the faster eigenvalue decay. We think this can be proved by obtaining a faster decay rate of the Hessian approximation's trace and a more delicate analysis on the regularized SR1 method that fully utilizes the benign property of self-concordant functions. We leave this for future work.

## H   LLM USAGE

In the preparation of this paper, we employed Deepseek (a large language model) solely for the purpose of refining language expression and correcting grammatical errors in the manuscript. The LLM was not involved in any aspect of research ideation, data analysis, interpretation of results, or substantive content generation. All intellectual contributions, including the formulation of research questions, methodological design, empirical investigation, and critical discussion, originate exclusively from the human authors. The use of the LLM was strictly limited to enhancing the clarity, coherence, and grammatical accuracy of the text, and it did not contribute to the scholarly or creative substance of the work.

