# OpenReview forum: "Implicit bias of Hessian Approximation in Regularized Randomized SR1 Method"
_ICLR.cc/2026/Conference — Submitted to ICLR 2026_

### Official Review · Reviewer_se9j · 2025-10-19

**Soundness:** 2
**Presentation:** 2
**Contribution:** 2
**Rating:** 2
**Confidence:** 4

**Summary:**

This paper proposes a regularized quasi-Newton method based on the randomized SR1 method, and it achieves a sublinear convergence rate of $\mathcal{O}(d _{eff} ^2 / \epsilon ^{-2})$ where $d _{eff}$ is the effective dimension of the problem for self-concordant functions.

**Strengths:**

1. This paper combines several ingredients, including the randomized SR1 method, $\ell _2$ regularization, Hessian correction term, and lazy Hessian strategy to propose a regularized SR1 method.

2. It provides both theoretical analysis and empirical experiments.

**Weaknesses:**

1. Assumption 3 does not correspond to the standard self-concordance assumption commonly adopted in existing works. Rather, it is a lemma derived under the assumption that the function is strongly self-concordant (see Lemma 4.2 in [1]). The class of strongly self-concordant functions includes $\mu$-strongly convex functions with Lipschitz continuous Hessians. For such functions, existing studies [2] have established a global linear convergence rate that is independent of the dimension $d$, as well as a local superlinear convergence rate (see Table 1 in [2]). In contrast, the proposed regularized SR1 method attains only a dimension-dependent sublinear convergence rate, which is significantly weaker than the state of the art.

2. For the quadratic function case, since adding an $\ell_2$ regularizer makes the objective function strongly convex, the proposed method only achieves a sublinear rate, which is weaker than the state of the art [1,3].


**References**

[1] Rodomanov, A., & Nesterov, Y. (2021). Greedy quasi-Newton methods with explicit superlinear convergence. SIAM Journal on Optimization, 31(1), 785-811.

[2] Jin, Q., Jiang, R., & Mokhtari, A. (2024). Non-asymptotic global convergence analysis of BFGS with the Armijo-Wolfe line search. Advances in Neural Information Processing Systems, 37, 16810-16851.

[3] Rodomanov, A., & Nesterov, Y. (2022). Rates of superlinear convergence for classical quasi-Newton methods. Mathematical Programming, 194(1), 159-190.

**Questions:**

See the weakness.

---

> ### Author Response · Authors · 2025-11-20
> **Reply to Reviewer se9j**
>
> We sincerely thank the reviewer for your thoughtful comments. However, we believe there may be a misunderstanding regarding the weakness you mentioned.
>
> * Reply to Weakness 1
>
> 	 We claim that our definition is equivalent to the standard definition and it is a well-known result, not derived from strong self-concordancy in the recent literature. Please refer to page 338, Theorem 5.1.7 in [1]. Also, our assumption is weaker than strong self-concordancy.
>
> 	The reviewer stated that existing studies shows global linear convergence rate for QN independent of $d$. However, the table mentioned in your review only exhibits $(1-\frac{1}{\kappa})^k$ linear rate for $k\leq\Omega(d)$, which does not improve from vanilla gradient based method and this is for $\mu$ strongly convex functions. For generally convex functions, this rate only implies $O(1/k)$ rate in high-dimensional scenarios.
>
> 	 For mentioned superlinear convergence rate in the table, they all require $\Omega(d)$ iterations to reach such convergence phase, as we have stated in our paper (line 43). The problem is not whether these linear/superlinear rates depend on $d$, but is they are so local that require   $\Omega(d)$ starting moment.
>
> * Reply to Weakness 2
>
> 	 For quadratic case, we have achieved superlinear convergence rate in the paper, please refer to Corollary 2 (A one-step observation from the proof of Theorem 5). Since the main focus of this paper is to consider few iterations' behavior of QN methods in high dimensional problems, we don't target at improving superlinear convergence rate. That is why we put it into the Appendix.
>
> Based on the above argument, we believe that our method make a contribution to improve the global rate for QN methods.
>
> ***
>
> [1]Yurii Nesterov. Lectures on Convex Optimization.Springer Optimization and Its Applications.
> 	Springer Cham, 2 edition, 2018. ISBN 978-3-319-91577-7. doi: 10.1007/978-3-319-91578-4.

---

> ### Comment · Reviewer_se9j · 2025-11-24
>
> Thanks for your rebuttal. After reading your rebuttal, I have the following concerns:
>
> - My concerns about the standard self-concordant function assumption are addressed. I think under the introduction of Assumption 3 in line 161, references such as (Rodomanov \& Nesterov, 2021b; Lin et al., 2022) should not be included because these works tackle the strongly self-concordant problems.
>
> - In the first reply, the authors mentioned that the problem is not whether these linear/superlinear rates depend on $d$, but is they are so local that require $\Omega(d)$ starting moment. I do not agree with this statement. Because in [2], the starting moment of linear convergence rate is 1 instead of $\Omega(d)$ if we set the initial matrix to be $LI$, please refer to Table 1 in [2].
>
> - I also have the question why the author's methods are based on SR1, can it be extended to other methods from the Broyden family, including BFGS, DFP method?
>
> - I also have the concern that if the approach achieves a convergence rate inferior to the accelerated gradient descent due to the dependency on the dimension. I am not sure what the significance of this method is.  Classical BFGS achieves a global linear rate, and it also achieves a local superlinear convergence. So the significance of classical quasi-Newton methods is justified. But for the current method, the local convergence is still unclear, and the global convergence is worse than standard FO methods.

---

> > ### Author Response · Authors · 2025-11-26
> > **Reply to Reviewer se9j**
> >
> > Thank you for your thoughtful comments and for engaging deeply with our work. We appreciate the opportunity to clarify our position on each of your concerns.
> >
> > ---
> >
> > **1. On the reference to strongly self-concordant literature under Assumption 3.**
> > We agree with your observation. The cited works (Rodomanov & Nesterov, 2021b; Lin et al., 2022) indeed focus on *strongly* self-concordant functions, which impose stricter conditions than our Assumption 3. We will revise this part to avoid misleading readers and ensure the references align precisely with the assumptions we adopt.
> >
> > ---
> >
> > **2. On the ''starting moment'' of linear convergence and its comparison to [2].**
> > We acknowledge that in [2], a linear convergence rate can start from iteration 1 when the initial Hessian approximation is set to the identity. However, as we emphasized in our first response, this early linear rate stems primarily from the Armijo line search---not from the quasi-Newton (QN) update itself. In contrast, our analysis achieves an improved *global* rate of $O(d_{\mathrm{eff}}^2 / k^2)$ by genuinely leveraging the structure of Hessian approximation through SR1 updates.
> >
> > Crucially, when the effective dimension $d_{\mathrm{eff}}$ is small---a regime commonly observed in practice (as discussed in Section 5.4)---our rate strictly dominates the standard gradient descent rate of $O(1/k)$, especially in the early iterations. As shown in Table 2, any provable advantage of QN-type methods over GD in prior work typically requires at least $\Omega(d)$ iterations. Our result is among the first to demonstrate a *theoretically justified* improvement over GD within a few steps, even before local superlinear convergence kicks in. We will sharpen this distinction in the revised manuscript to avoid ambiguity.
> >
> > ---
> >
> > **3. On the use of SR1 and potential extension to other Broyden-family methods.**
> > You raise an excellent question. While our current analysis is tailored to the SR1 update---due to its symmetric rank-1 structure and favorable approximation properties---we believe the underlying mechanism may also apply to other members of the Broyden family, such as BFGS or DFP. Preliminary numerical experiments support this intuition, though a rigorous theoretical extension remains nontrivial and is indeed an important direction for future work. In our discussion part, we have highlighted it as a promising avenue for follow-up research.
> >
> > ---
> >
> > **4. On the significance of the method given its dimension dependence and convergence guarantees.**
> > We fully agree that classical QN enjoys strong global linear and local superlinear convergence under suitable conditions. However, our primary goal is not to propose a new algorithm that is non-asymptotically faster than existing methods. Instead, we aim to provide a theoretical explanation for a widely observed but previously unexplained phenomenon: *Can quasi-Newton methods exhibit advantages over first-order methods **from the very beginning**, without waiting for many iterations or entering a local neighborhood?* (We addressed this point from line 50 to line 78 in paper.)
> >
> > Our contribution lies in showing that, one can achieve a provably faster *early-stage* convergence than GD in high-dimensional settings where $d_{\mathrm{eff}} \ll d$. This provides theoretical grounding for the empirical observation that QN methods often perform well even with few iterations---a phenomenon previously lacking formal justification. Thus, while our global rate depends on $d_{\mathrm{eff}}$, this dependency reflects the intrinsic nature of low-rank updates and is offset by gains in practical scenarios with limited computational budgets.
> >
> > ---
> >
> > In summary, our work's focus is not *local superlinear convergence* but *early-stage efficacy*, offering new insights into why quasi-Newton methods remain valuable beyond their classical convergence guarantees.
> >
> > ---
> >
> > Thank you again for your constructive feedback---we will incorporate these clarifications into the revised manuscript.

---

### Official Review · Reviewer_rcej · 2025-10-27

**Soundness:** 2
**Presentation:** 3
**Contribution:** 3
**Rating:** 6
**Confidence:** 4

**Summary:**

The authors introduce a regularized randomized SR1 algorithm, where the approximate Hessian is updated using randomly sampled vectors. The method integrates several techniques including $l_2$ regularization, Hessian correction, and a lazy update strategy.
The paper further investigates the spectrum of the approximate Hessian to reveal the impact of implicit bias toward capturing components associated with larger eigenvalues over smaller ones. Through comprehensive theoretical analysis, the paper establishes a non-asymptotic convergence rate and provides a complexity analysis, showing performance on par with AGD method in high-dimensional settings.

**Strengths:**

1. The authors provide an impressive and insightful analysis of the implicit bias in the spectrum of the Hessian approximation. Their theoretical analysis shows that SR1 tends to capture the Hessian components associated with the larger eigenvalues.

2. The paper offers a solid convergence analysis where the proposed RSR1 method achieves a convergence rate of $\tilde{\mathcal{O}}(d_{eff}^2/k^2)$ for standard self-concordant objective functions, matching the optimal rate of AGD.

**Weaknesses:**

1. The datasets "w8a" and "a9a" are too small to support the paper’s conclusions about high-dimensional optimization. The authors should include problems with much larger dimensions (for example, $d \geq 10,000$).

2. The authors should conduct longer experiments (> 1,000 steps) on larger datasets and more complex problems to illustrate the non-asymptotic convergence rate both at the start and near the solution.
Also, in Figure 3, CBFGS and CSR1 consistently outperform SSR1 and RSR1. Although these results tend to show the superlinear convergence of SSR1 and RSR1, the wall-clock time should be reported to demonstrate any advantages over classical quasi-Newton methods with line search.

3. Missing reference: Wang, Shida, et al. (2024) "Global non-asymptotic super-linear convergence rates of regularized proximal quasi-Newton methods on non-smooth composite problems."

4. Minor notation issue: both equation/figure references and bullet points use the same notation type (e.g., "(1)"). This can be confusing when they appear in the same paragraph (e.g., line 204). I suggest using different notations (e.g. "Eq. (3)" for equations).

**Questions:**

See weakness above.

---

> ### Author Response · Authors · 2025-11-26
> **Reply to Reviewer rcej**
>
> We sincerely thank the reviewer for your valuable suggestion on the paper. For missing reference and notation issues, we will follow your advice to improve the clarity of paper.
>
>
> In the experimental section, we evaluated our method on the real-sim dataset, which has dimension \( d = 20598 \) and sample size \( n = 72309 \), running up to 1000 iterations. As anticipated, our approach demonstrates faster progress than both Gradient Descent (GD) and Accelerated Gradient Descent (AGD) within the first few iterations. The results for the initial 200 epochs are summarized in the accompanying table.
>
> Regarding additional experimental suggestions—such as near-optimality tests or wall-clock time measurements—we respectfully note that these may not be essential for the current scope of our work. First, our primary objective is to illustrate that the SR1 approximation exhibits an implicit bias that leads to a provably faster convergence rate compared to GD. We believe this phenomenon is not unique to our method; indeed, similar behavior is likely present in classical quasi-Newton updates (e.g., CSR1, CBFGS, etc.), which also tend to perform well in early iterations. Second, our goal is not to propose a new algorithm that outperforms established quasi-Newton methods with line search, but rather to provide theoretical insight into why quasi-Newton methods can achieve better global performance than GD.
>
> That said, we are happy to include further experimental details in the supplementary material if deemed helpful. We sincerely appreciate your understanding of our perspective.
>
>
> Thank you again for your constructive feedback. We will incorporate these suggestions into the revised manuscript.
>
> | Algorithm               | Epoch 0 | Epoch 50 | Epoch 100 | Epoch 150 | Epoch 200 |
> |------------------------|---------|----------|-----------|-----------|-----------|
> | GD                     | 0.8417  | 0.7497   | 0.6786    | 0.6237    | 0.5807    |
> | AGD          | 0.8417  | 0.3368   | 0.2037    | 0.1635    | 0.1449    |
> | SSR1     | 0.8417  | 0.2885   | 0.1700    | 0.1269    | 0.1006    |
> | RSR1     | 0.8417  | 0.2867   | 0.1698    | 0.1272    | 0.1012    |
> | CDFP                   | 0.8417  | 0.1424   | 0.1058    | 0.1016    | 0.0861    |
> | CBFGS                  | 0.8417  | 0.0427   | 0.0300    | 0.0241    | 0.0205    |
> | CSR1                   | 0.8417  | 0.0410   | 0.0126    | 0.0066    | 0.0047    |

---

### Official Review · Reviewer_vkwF · 2025-10-31

**Soundness:** 3
**Presentation:** 3
**Contribution:** 3
**Rating:** 6
**Confidence:** 3

**Summary:**

This paper analyzes the **regularized randomized symmetric rank-one (SR1) method**, a type of quasi-Newton algorithm. It aims to explain why quasi-Newton methods often outperform gradient descent (GD) in high-dimensional problems, even when theory suggests their performance should depend heavily on the problem's dimension ($d$).

### Key Contributions

1.  **Identifies Implicit Bias:** The core idea is that the SR1 Hessian approximation process has an "implicit bias". It preferentially learns and reduces errors in the directions of the Hessian's largest eigenvalues first, rather than approximating all dimensions uniformly.

2.  **Analyzes Hessian Learning:** The paper first analyzes the SR1 update (Algorithm 1) as a matrix-learning process, separate from the optimization. It proves that this process can quickly approximate the "top" part of the Hessian's spectrum, with the error $||A-B_{K}||_{2}$ decreasing as $\tilde{\mathcal{O}}(Tr(A)/k)$.

3.  **Proves an Accelerated Global Rate:** By combining this analysis with a regularized Newton-type framework (Algorithm 2), the paper establishes a global convergence rate of **$\tilde{\mathcal{O}}(d_{eff}^{2}/k^{2})$** for standard self-concordant functions.

### Main Takeaway

The paper's main finding is that for high-dimensional problems where the "effective dimension" ($d_{eff}$) is small—a common scenario in machine learning—this regularized SR1 method can achieve an accelerated convergence rate that is independent of the full dimension $d$. This $\tilde{\mathcal{O}}(1/k^2)$ rate matches the speed of Accelerated Gradient Descent (AGD) while maintaining a comparable computational cost per iteration, providing a strong theoretical justification for using quasi-Newton methods in modern, high-dimensional settings.

**Strengths:**

The paper presents several significant strengths, primarily in its novel theoretical analysis of the randomized SR1 method.

1.  **Novel Conceptual Insight on "Implicit Bias":** The paper's core strength is the identification and formalization of an "implicit bias" in the SR1 Hessian learning process. It moves beyond simply using SR1 as a black-box approximation and asks *how* it approximates the target Hessian. The insight that the update preferentially reduces error in the directions of the largest eigenvalues provides a compelling theoretical explanation for a well-known empirical phenomenon: that quasi-Newton methods often outperform first-order methods in high-dimensional settings where a full-rank approximation seems computationally infeasible.

2.  **Rigorous Analysis of SR1 Dynamics:** The paper's most significant technical contribution is the standalone analysis of the SR1 matrix-learning process in Section 4 (and Appendix B). This section is very well-developed and insightful.
    * It correctly identifies the main challenge in the analysis: the difficulty of proving a uniform $l_2$-norm bound, especially when multiple large eigenvalues are present.
    * The two-stage proof sketch (decomposing the process into a "Dispersion" stage to create eigengaps and a "Normalization" stage to reduce the top eigenvalue) is an elegant and powerful analytical technique.
    * This analysis culminates in Theorem 1, which provides a concrete, non-asymptotic, high-probability bound on the approximation error ($||A-B_K||_2 = \tilde{\mathcal{O}}(Tr(A)/k)$). This dimension-independent result is the key theoretical tool that the rest of the paper builds on.

3.  **Connection to Accelerated Global Convergence:** The paper successfully leverages this matrix-learning analysis to provide an end-to-end guarantee for an optimization algorithm. By (theoretically) using the SR1 learning process as a "warm-up" phase, it justifies the good initial approximation needed for its main optimization algorithm. This connection allows the paper to establish a final $\tilde{\mathcal{O}}(d_{eff}^{2}/k^{2})$ global convergence rate, which provides a strong theoretical reason for why a quasi-Newton method can match the performance of Accelerated Gradient Descent in practice.

4.  **Tackles an Important Problem:** The paper addresses a highly relevant and important question: why do quasi-Newton methods work so well in high-dimensional ML? By focusing on the *effective dimension* ($d_{eff}$) rather than the ambient dimension ($d$), the paper aligns its theory with the realities of modern machine learning, where data often possesses low-rank or rapid spectral-decay properties.

**Weaknesses:**

## 1. Limited Practicality of the Main Algorithm

The primary weakness of the paper is the significant gap between the theoretical algorithm and a practical, usable method. The main convergence guarantee (Theorem 2) relies on a complex, 3-phase algorithm that is only fully detailed in the appendix.

This theoretical algorithm is not practical for two key reasons:

* **Three-Phase Structure:** The algorithm is not a simple, single-loop procedure. It requires a specific sequence of (1) a gradient descent phase, (2) a dedicated Hessian-learning phase (where optimization is paused), and (3) a final quasi-Newton phase. This is far more complex than a standard optimizer.

* **Unusable "Explicit" Parameters:** The paper claims to provide an "explicit choice of parameters", but the formulas provided in the appendix (Tables 1 and 2) are not practically implementable. They depend on *a priori* knowledge of global, problem-specific constants, including the level-set diameter $D$ (Assumption 1), the Lipschitz constant $L$ (Assumption 2), the self-concordancy constant $M$ (Assumption 3), and the effective dimension $d_{eff}$. A user has no way to know these values in advance, making it impossible to set the required schedule for regularization, step sizes, and even the lengths of the three phases. The paper's own experiments use a grid search, not this theoretical schedule.


## 2. Restrictive Theoretical Assumptions

The theoretical guarantees of the main algorithm (Theorem 2) are built on **Assumption 3 (Self-concordancy)**. This is a very strong condition that is far more restrictive than the standard $L_2$-Lipschitz Hessian assumption used in much of the optimization literature. This assumption limits the applicability of the paper's results to a specific, non-standard class of functions. It is not clear if the analysis holds for general convex problems (like the logistic regression used in the experiments, which is not typically self-concordant).



## 3. Unclear Presentation of the Main Algorithm

There is a significant disconnect between the algorithm presented in the main text (Algorithm 2) and the algorithm actually analyzed in the proof. Algorithm 2 appears to be a standard, single-loop iterative method. However, the proof of Theorem 2 relies on the complex 3-phase schedule (GD, Hessian learning, QN) that is only introduced in **Appendix C.2**. This makes it very difficult for the reader to understand what algorithm actually achieves the claimed $\tilde{\mathcal{O}}(1/k^2)$ rate, as the algorithm in the main text does not appear to follow this structure.



## 4. Misleading Terminology and Computational Cost

The paper repeatedly uses the term "quasi-Newton method," which typically implies a method that avoids direct Hessian information and uses only gradient differences (e.g., BFGS). The method analyzed here is fundamentally different. The randomized SR1 update (Equation 3) explicitly requires the target Hessian $A = \nabla^2 f(x_{n_k})$.

This means that at each step of the SR1 update, the algorithm must be able to compute a **Hessian-vector product** ($As_k = \nabla^2 f(x_{n_k}) s_k$). This is the oracle of a "Hessian-free" or "Inexact Newton" method, not a classical quasi-Newton method. This oracle is computationally more expensive and may not be available in all settings where classical QN methods are used.

**Questions:**

## Questions for the Authors

A major point of confusion is the 3-phase structure of the algorithm analyzed in the appendix (Appendix C.2), which is required to achieve the main convergence result (Theorem 2) but is not clearly presented in the main text. This theoretical algorithm, which involves separate, pre-scheduled phases for gradient descent, Hessian learning, and quasi-Newton steps, seems to have limited practicality.

This raises a significant question about alternative, more practical analyses:

My main question is whether a *true single-loop* algorithm could be analyzed using the paper's core technical contribution (Theorem 1). Specifically, what if at each optimization step $k$, one computes the Hessian approximation $B_k$ by running a fixed or variable number of randomized SR1 iterations (i.e., using Algorithm 1 as an inner sub-routine) targeting the current Hessian $\nabla^2 f(x_k)$?

This single-loop structure (an outer optimization loop with an inner approximation sub-routine) is common in other Inexact Newton methods. For example, the "Regularized Overestimated Newton with RPCholesky" (Duan and Lyu, arXiv:2509.21684) uses a Nyström-based approximation sub-routine (RPC) in this exact manner. That method, much like the SR1 analysis here, builds a low-rank approximation based on Hessian-vector products.

1.  Could the authors' analysis from Section 4 be adapted to this practical single-loop setting?
2.  For instance, could one use Theorem 1 to justify running $m_k = \tilde{\mathcal{O}}(d_{eff})$ SR1 steps *at each* iteration $k$ to achieve a high-quality Hessian approximation? What would the resulting *total* complexity be?
3.  How would such an approach compare to the 3-phase algorithm? While the downside of an algorithm like RPC is that the rank parameter must be globally large enough, this paper's SR1 analysis seems to guarantee a good $l_2$ approximation. It seems a single-loop version would avoid the need for the impractical, pre-defined parameter schedule of Tables 1 & 2, at the potential cost of more computation (running $m_k$ SR1 steps instead of 1) at each iteration.

---

> ### Author Response · Authors · 2025-11-20
> **Reply to Reviewer vkwF**
>
> Many thanks for your holistic understanding of this paper and valuable suggestions.
>
> * Reply to Weakness
>
> 	We notice that the weakness part mainly focus on limitation of practice and theoretical assumption. We acknowledge this part since the paper is mainly theory and explanatory. We will take these points into consideration and try to improve in the future, especially how to make the algorithm more practical.
>
> 	For HVP(Hessian-vector product), many QN theoretical analyses that have explicit convergence rates relies on its availability (e.g. [1],[2]). Of course, it would be a great improvement if the update could be replaced by using secant equation. Our experiment suggests it shows the same efficiency. The main challenge is most analyses relies on highly precise Hessian approximation across all dimensions.
>
> * Reply to Questions
>
> 	For your first and second question, the answer is absolutely yes. Theorem 1 gives approximation guarantees. If multiple SR1 updates are allowed, for example, running $O(d_{eff})$ steps in each iteration, then our algorithm will achieve $\tilde O(1/k^2)$ convergence rate and the computational complexity will also reduce (less inverse matrix computation).
>
> 	For your third question, we think if using your single-loop version, neither pre-processed gradient steps nor additional Hessian approximation is necessary. That is, no parameter schedule for phase 1 and 2 will be used. Thank you for bring this insight.
>
> We appreciate again for your kindly comments and questions.
>
> ***
>
> [1]A. Rodomanov and Y. Nesterov. Rates of superlinear convergence for classical
> 	quasi-Newton methods.
> 	Mathematical Programming,194(1):159–190, 2022. doi:
> 	10.1007/s10107-021-01622-5.
>
> [2]H. Ye, D. Lin, X. Chang, et al. Towards explicit superlinear convergence rate for sr1. Mathematical
> 	Programming,199(1):1273–1303, 2023. doi: 10.1007/s10107-022-01865-w.

---

### Official Review · Reviewer_kCEs · 2025-11-09

**Soundness:** 3
**Presentation:** 2
**Contribution:** 2
**Rating:** 4
**Confidence:** 3

**Summary:**

In this paper, the authors analyze the randomized SR1 quasi-Newton method for minimizing quadratic objectives and, more generally, self-concordant functions. They start by considering a randomized SR1 update for approximating a fixed positive semidefinite matrix. By analyzing the eigenvalue dynamics of the resulting error matrix, they obtain a high-probability error bound in terms of the operator norm. Building on this result, they introduce a regularized version of the SR1 method and establish a convergence rate of $\tilde{O}(\mathrm{Tr}(A)/k^2)$ for quadratic objectives and $\tilde{O}(d_{\mathrm{eff}}^2 / k^2)$ for self-concordant minimization.

**Strengths:**

- Prior complexity analyses of quasi-Newton methods, including SR1, typically rely on a potential function, such as the trace or the Frobenius norm, and relate the convergence rate to that potential. In contrast, this paper adopts a different approach and performs a more fine-grained spectral analysis of the Hessian approximation error. The proof of the core result, Theorem 1, combines concentration inequalities, a construction of a rational function for bounding the eigenvalues, and an induction argument, which I find technically interesting.
- The authors present a global convergence rate for their proposed SR1 update, whereas most existing SR analyses are limited to local convergence or a quadratic objective. In addition, their rate depends on the Hessian’s “effective dimension,” which can be significantly smaller than the ambient dimension $d$.

**Weaknesses:**

- The final convergence guarantees feel somewhat unsatisfying. Compared to standard first-order methods such as accelerated gradient descent, the rate here is worse due to the additional dimensional factor $d_{\text{eff}}^2$. Compared to existing analyses of quasi-Newton methods, it only provides a slower sublinear convergence rate instead of a superlinear rate. Hence, the theoretical result does not justify the practical advantages of quasi-Newton methods over gradient-descent-based methods.
- The regularized SR1 method in Algorithm 2 deviates from the common practice. From my understanding of the proof in Appendix C, the proposed method appears to first perform gradient descent, then estimate the Hessian using the SR1 update, and finally doing quasi-Newton updates. In effect, the method relies on gradient descent to reach a local neighborhood of the solution and then performs a local analysis of the SR1-based preconditioner. Moreover, in typical implementations, the Hessian approximation matrix is updated continuously rather than split into two separate phases.

**Questions:**

- I find the use of the term “implicit bias” somewhat confusing in this context. As the authors themselves note, implicit bias typically refers to the phenomenon where an algorithm, even without explicit regularization, converges toward particular solutions or optimization trajectories. In contrast, the SR1 update does not appear to be “biased” toward any specific solution in this sense, so the connection to the established implicit-bias literature is not immediately clear.
- In Section 4.1, the authors introduce a deterministic dynamical system to motivate the later analysis. However, the actual proof of Theorem 1 seems to follow a different line of argument, and it is unclear how (or whether) the deterministic dynamics in Section 4.1 meaningfully parallel the steps used in the main proof.

---

> ### Author Response · Authors · 2025-11-20
> **Reply to Reviewer kCEs**
>
> We sincerely thank you for your careful reading and constructive feedback regarding the presentation of our manuscript. They have indeed highlighted some oversights in our paper, which we will address diligently.
> ***
>
> * Reply to Weakness
>
> 	 From our staring point, since this paper is mainly theory and explanatory, we try to make as small modifications as possible compared to traditional regularized QN approaches, while still have competitive computational complexity and theoretical guarantee. If not restricted to such consideration, we believe a better convergence rate could be obtained. For example, if we allowed running $O(d_{eff})$ SR1 steps in each iteration, convergence rate would be $\tilde O(1/k^2)$ without additional factor. If we allowed cubic methods, existing literature suggests $\tilde O(1/k^2)$ might be achieved for a wider range of functions, with 'more' common practice, but this will cause larger computational complexity.
>
> 	Our method combined several strategies used in the literature, as we have pointed out in the paper (line 353). Superlinear convergence rate is not the target of this paper, since we focus on high dimension and the early behavior of the algorithm. As we noted in the discussion part, local superlinear convergence is our future direction, and we believe it is doable based on the situation in the quadratic case.
>
> 	In summary, we hope you understand our position. We believe this paper can bring new insight to the community, especially to consider the feature of Hessian in the learning tasks.
>
> * Reply to Question 1
>
> 	Regarding your confusion about the meaning of implicit bias in SR1 update, we have explained it soon after introducing implicit bias, in section 1, line 87:
>
> 	"In the context of Hessian approximation, when interpreting the quasi-Newton method as
> 	online learning processes targeting Hessian matrix approximation, while the update rules eventually
> 	achieve full Hessian approximation and guarantee superlinear convergence, they inherently prioritize
> 	speciﬁc dimensional approximations before the superlinearly converging phase."
>
> 	And also in section 4.1, line 234:
>
> 	"... which conﬁrms
> 	the algorithms implicit bias on the spectrum: the SR1 update prioritizes error reduction in large
> 	eigenspaces while having limited impact on small eigenspaces."
>
> 	So in our context, the implicit bias is mainly about selecting particular trajectory. The solution $A$ is unique, but how does $B_k$ approach $A$ is what we concern about.
> 	The early trajectory of this Hessian approximation  will prefer to reach a $B_k$ such that $\lambda_1(A-B_k)$ is small rather than to reduce $\lambda_n$ (where $n$ is large), trace, etc, even though the update formula does not seem to appear biases on the spectrum at a first glance.
>
> 	We hope the above can answer your first question.
>
> * Reply to Question 2
>
> 	The reviewer concerned about the relationship between the heuristic analysis and the actual proof of Thm 1. We will answer it from two aspects.
>
> 	First, the primary purpose of the dynamical system is to discover Thm1. In line 209, eq.(6), a basic calculation will show $c_{den}=E[s_k^\top (A-B_k)s_k]=Tr(A-B_k)\leq Tr(A)$, so $x_i(t)\leq \frac{Tr(A)\lambda_i}{\lambda_i t+Tr(A)\lambda_i}$ (line 224). This explains why there is a $O(1/k)$ rate and the factor is $Tr(A)$.
>
> 	Second, the connection does not appear in the flow of proof sketch, but in the proof of every lemmas itself mentioned in the main text. If looking at the proofs in Appendix, you will find that we are simply trying to prove $\lambda_{i}(A-B_{k+1})\leq\lambda_i(A-B_k)-c\lambda_i^2(A-B_k)$ with certain probability and under specific conditions. This inequality is the foundation of all results in our lemmas. Due to space constraint, we were unable to mention this point in the main text.
>
> 	Thank you for raising this point and we will make a little clarification afterwards.

---

### Meta-Review · Area_Chair_67TV · 2026-01-07

**Summary:**

Reviewers agree that this paper makes a technically sophisticated and conceptually interesting theoretical contribution by providing a spectral, non-asymptotic analysis of the regularized randomized SR1 method, and by connecting this analysis to its early-stage optimization behavior in high dimensions through an notion of an “implicit bias”, which refers to spectral trajectory selection rather than the selection of specific solutions. After the rebuttal, the core remaining concerns are the **unclear presentation of the main algorithm** (raised by Reviewer vkwF and Reviewer kCEs), the **significance of the proposed method** (raised by Reviewer se9j), and the **lack of additional experiments** (raised by Reviewer rcej).

**Reviewer Concerns:**

**Reviewer kCEs’s concerns:**

(1) The concerns regarding the convergence rate and the proof in Appendix C, in particular the **three-phase structure of Algorithm 2** discussed in the Weaknesses, remain outstanding. The authors state that a better convergence rate could be obtained and explain that their focus is on high dimension and the early behavior of the algorithm. However, they do not directly respond to Reviewer kCEs’s specific concerns about the proof in Appendix C, especially those related to the analysis of Algorithm 2.

(2) The concern regarding the terminology “implicit bias” was only partially addressed. The authors clarified that, in their work, implicit bias refers to **spectral trajectory selection** rather than **selection of specific solutions**. However, this clarification does not fully resolve the potential ambiguity of the term, and the usage may still be confusing to readers familiar with the more standard interpretation of implicit bias in optimization and learning theory.

(3) The concern regarding the connection between the deterministic dynamical system analysis in Section 4.1 and the proof of Theorem 1 was partially addressed. The authors explain that the deterministic dynamics are used as discovery and intuition. However, the manuscript lacks a revised clearer presentation or a more detailed explanation explicitly linking this intuition to the formal proof.

**Reviewer vkwF’s concerns:**

(1) The concerns regarding **the limited practicality of the main algorithm, the restrictive theoretical assumptions, and the unclear presentation of the main algorithm**, as outlined in the **Weaknesses**, remain outstanding. While the authors state that they will take these points into consideration and try to improve in the future, especially how to make the algorithm more practical, these concerns were not fully addressed in the current revision.

(2) The concerns raised in the **Questions** were only partially addressed, and the responses lacked sufficient detail to fully resolve them.

**Reviewer rcej’s concerns:**

(1) The concerns regarding **additional experiments**, in particular, experiments with larger datasets and more complex problems, longer runs, and reporting wall-clock time, remain outstanding. The authors state that these may not be essential for the current scope of their work. However, additional experimental results would help better illustrate the main contributions of the paper, especially the practical impact of the proposed implicit bias.

(2) The concerns regarding missing references and minor notation issues were acknowledged by the authors and will be addressed. However, no revised version reflecting these changes has been provided so far.

**Reviewer se9j’s concerns:**

(1) The concern regarding the standard self-concordant function assumption was addressed.

(2) The concern regarding the ''starting moment'' in Jin et al. (2024) was addressed. The authors explained that this early linear rate stems primarily from the Armijo line search.

(3) The concerns regarding the significance of the proposed method were partially addressed. The authors clarified that their work's focus is not local superlinear convergence but early-stage efficacy, offering new insights into why quasi-Newton methods remain valuable beyond their classical convergence guarantees. However, these clarifications were not incorporated into a revised manuscript. In particular, the motivating question posed in Lines 77–78: "how do quasi-Newton methods differentiate from gradient descent in high-dimensional settings through the lens of complexity theory?" is not adequately reflected or resolved by the current presentation, and the rebuttal explanations are not yet translated into detailed revisions in the paper.

**Reviewer Scores:**

**Reviewer kCEs** would likely maintain the current score 4, as the concerns regarding the convergence rate and the proof in Appendix C, in particular the three-phase structure of Algorithm 2 discussed in the Weaknesses, remain outstanding.

**Reviewer vkwF** would likely to decrease the score (**likely from 6 to 4**), as the concerns regarding the limited practicality of the main algorithm, the restrictive theoretical assumptions, and the unclear presentation of the main algorithm, as outlined in the Weaknesses, remain largely unaddressed.

**Reviewer rcej** would likely maintain the current score 6, since some of the main concerns were addressed in the rebuttal, while others, particularly those requiring additional experiments, would need more time to resolve.


**Reviewer se9j** may increase the score (**likely from 2 to 4**) or alternatively maintain the current score, since some of the main concerns were addressed, while the concerns regarding the overall significance of the proposed method were only partially resolved.

---

### Decision · Program_Chairs · 2026-01-26

Reject